# Tracer and observationally-derived constraints on diapycnal diffusivities in an ocean state estimate

David S. Trossman[1,2], Caitlin B. Whalen[3], Thomas W. N. Haine[4], Amy F. Waterhouse[5], An T. Nguyen[6], Arash Bigdeli[7], Matthew Mazloff[5], and Patrick Heimbach[6,8]

[1]Department of Oceanography & Coastal Sciences, Louisiana State University, Baton Rouge, USA
[2]Center for Computation & Technology, Louisiana State University, Baton Rouge, USA
[3]Applied Physics Laboratory, University of Washington, Seattle, USA
[4]Department of Earth & Planetary Sciences, Johns Hopkins University, Baltimore, USA
[5]Scripps Institution of Oceanography, University of California at San Diego, San Diego, USA
[6]Oden Institute for Computational Engineering & Sciences, University of Texas at Austin, Austin, USA
[7]EP Analytics, Inc., Austin, USA
[8]Jackson School of Geosciences & Institute for Geophysics, University of Texas at Austin, Austin, USA

**Correspondence:** David S. Trossman (dtrossman@lsu.edu)

**Abstract.** Use of an ocean parameter and state estimation framework–such as the Estimating the Circulation & Climate of the Ocean (ECCO) framework–could provide an opportunity to learn about the spatial distribution of the diapycnal diffusivity parameter ($\kappa_\rho$) that observations alone cannot due to gaps in coverage. However, we show that the inclusion of misfits to observed physical variables–such as in situ temperature, salinity, and pressure–currently accounted for in ECCO is not sufficient, as $\kappa_\rho$ from ECCO does not agree closely with any observationally-derived product. These observationally-derived $\kappa_\rho$ products were inferred from microstructure measurements, derived from Argo and CTD data using a strain-based parameterization of finescale hydrographic structure, or calculated from climatological and seafloor data using a parameterization of tidal mixing. The $\kappa_\rho$ products are in close agreement with one another, but have both measurement and structural uncertainties, whereas tracers can have relatively small measurement uncertainties. With the ultimate goal being to jointly improve the ECCO state estimate and representation of $\kappa_\rho$ in ECCO, we investigate whether adjustments in $\kappa_\rho$ due to inclusion of misfits to a tracer–dissolved oxygen concentrations from an annual climatology–would be similar to those due to inclusion of misfits to observationally-derived $\kappa_\rho$ products. We do this by performing sensitivity analyses with ECCO. We compare multiple adjoint sensitivity calculations: one configuration that uses misfits to observationally-derived $\kappa_\rho$ and the other uses misfits to observed dissolved oxygen concentrations. We show that adjoint sensitivities of dissolved oxygen concentration misfits to the state estimate's control space typically direct $\kappa_\rho$ to improve relative to the observationally-derived values. These results suggest that the inclusion of oxygen in ECCO's misfits will improve $\kappa_\rho$ in ECCO, particularly in (sub)tropical regions.

## 1 Introduction

We consider the challenges with using observational data in the context of a parameter and state estimation framework to infer the global distribution of ocean mixing. Ocean models must parameterize the unresolved, turbulent diffusion of oceanic tracers.

Ocean mixing is typically conceptualized in terms of diffusion along and across isopycnal surfaces. Subgrid-scale transport of isopycnal thickness (or bolus)–which is effectively an advective contribution to tracer budgets–also must be parameterized. Ocean models often represent these unresolved processes with three parameters: the across-isopycnal mixing parameter (diapycnal diffusivity; *Munk and Wunsch*, 1998), the along-isopycnal mixing parameter (Redi coefficient; *Redi*, 1982), and the eddy isopycnal thickness transport parameter (Gent-McWilliams coefficient; *Gent and McWilliams*, 1990). Diapycnal mixing is an

essential component in explaining the observed oceanic stratification (*Munk and Wunsch*, 1998; *Gnanadesikan*, 1999; *Scott and Marotzke*, 2002). Changes in the background diapycnal diffusivity (*Dalan et al.*, 2005; *Krasting et al.*, 2018; *Hieronymus et al.*, 2019; *Sinha et al.*, 2020), Redi coefficient (*Gnanadesikan et al.*, 2015; *Ehlert et al.*, 2017), and Gent-McWilliams coefficient (*Danabasoglu and McWilliams*, 1995) are known to have a profound influence on climate simulations through alterations in the response to surface flux perturbations and changes in ventilation rates.

The spatiotemporal variabilities suggested in previous studies of the Redi coefficient (*Abernathey et al.*, 2013; *Bates et al.*, 2014; *Forget et al.*, 2015b; *Cole et al.*, 2015; *Busecke and Abernathey*, 2019; *Groeskamp et al.*, 2020) and Gent-McWilliams coefficient (*Forget et al.*, 2015b; *Katsumata*, 2016; *Bachman et al.*, 2020) fields are virtually absent in ocean models. There is also a dearth of independent observations with which to assess their observationally-derived values (*Cole et al.*, 2015; *Katsumata*, 2016; *Roach et al.*, 2018; *Groeskamp et al.*, 2020), and these values cannot be easily compared with those in

models. For instance, it is unclear how to compare Redi coefficients derived from observations with those from models because they are expected to vary with horizontal resolution. Also, the formulations of the perpendicular and parallel components of the eddy advection tensor relative to isopycnal surfaces are not the same in many models as in the observationally-derived Gent-McWilliams coefficient product (*Katsumata*, 2016). To gain deeper insight into the issues with model-representation of ocean mixing, we focus on the diapycnal diffusivity field–$\kappa_\rho$ hereafter–in this study.

Parameterizations for $\kappa_\rho$ (*Gaspar et al.*, 1990; *Large et al.*, 1994; *Reichl and Hallberg*, 2018) have allowed for a spatiotemporally-varying $\kappa_\rho$ field, but assessing the performance of these parameterizations has been challenging due to a profound lack of observations. Until recently, the only available observational information about $\kappa_\rho$ came from tracer release experiments (*Ledwell and Watson*, 1991; *Polzin et al.*, 1997; *Messias et al.*, 2008) and microstructure (i.e., the scales over which molecular viscosity and diffusion are important) measurements of velocity shear (e.g., *Waterhouse et al.* (2014)) or temperature variability (e.g.,

*Gregg* (1987)). These data are infrequently sampled and cover a relatively small portion of the ocean, but are independent observations with which to compare the more recent global mixing data products calculated from Argo (*Whalen et al.*, 2015), CTD (*Kunze*, 2017), and climatological and seafloor (*de Lavergne et al.*, 2020) observations. While our understanding of the global distribution of $\kappa_\rho$ has been transformed by the use of theories to derive $\kappa_\rho$ from limited observations (*MacKinnon et al.*, 2017; *Whalen et al.*, 2020), none of the observationally-derived $\kappa_\rho$ products have been used to date in ocean models to

assess whether corresponding simulations would be improved over globally uniform values, or those based on theory. (Here, by "constrain," we were referring to using new data to change the level of agreement between the model and an observational product–not necessarily to achieve a perfect match.)

Currently, the only information about $\kappa_\rho$ comes from temperature, salinity, and pressure observations in ocean parameter and state estimation or data assimilation systems. If these observations were collected at every location and depth of the ocean, there

could be sufficient information to accurately derive $\kappa_\rho$ (*Groeskamp et al.*, 2017), but there are spatiotemporal gaps. This work explores the use of a parameter and state estimation framework to invert for global fields of $\kappa_\rho$ using incomplete observations and theories.

    We use the Estimating the Circulation & Climate of the Ocean (ECCO) parameter and state estimation framework to evaluate how near-global, observationally-derived $\kappa_\rho$ can be used to inform ocean models. The aim of the ECCO framework is to
reconstruct the recent history of the ocean (the "state estimate") by filling in the gaps between incomplete observations, which are often sparse and aliased, through dynamical techniques. The state estimate is related to a reanalysis product (*Heimbach et al.*, 2019), but the state estimation framework overcomes some serious shortcomings (see the Appendix) by requiring dynamical and kinematical consistency (*Stammer et al.*, 2016) of the estimated state throughout its full period of estimation (here, 1992 to 2015). The version 4, release 3 of ECCO (ECCOv4r3; *Fukumori et al.* (2017)) state estimate–like previous versions and
releases–is achieved by fitting a general circulation model to available observations in a weighted least squares sense (*Wunsch*, 2006; *Forget et al.*, 2015a). The model-data misfit (objective or "cost function") is minimized by varying (i.e., inverting for) a set of uncertain control variables, all of which are independent inputs to the model equations being solved. These control variables can be iteratively improved by running the model in forward plus backward–its "adjoint"–mode, which enables the calculation of gradients in the cost function. Each of these runs maintains dynamical and kinematical consistency because,
in contrast to filter-based data assimilation systems (see the Appendix for an example), the only ocean variables that get adjusted are the control variables–not the dynamically active–or prognostic–variables. These control variables are determined using the entire length of the state estimate–as opposed to introducing temporal discontinuities by periodically adjusting them. Importantly for our goal of parameter estimation, the set of control variables may consist not only of initial and boundary conditions, but also of (spatially-varying) model parameters, such as the three used to represent ocean subgrid-scale transport
or mixing (*Liu et al.*, 2012; *Forget et al.*, 2015a). Inaccuracies in variables such as $\kappa_\rho$ in any ocean model can make physical inference less grounded in reality–e.g., the differences in the importance of diapycnal mixing in steric sea level budgets of models used in this study (*Piecuch and Ponte*, 2011; *Palter et al.*, 2014)–and could make the ECCO state estimate itself less accurate–e.g., errors in $\kappa_\rho$ will influence vertical tracer transport and mixed layer depths. Since it remains under-explored how well $\kappa_\rho$, in particular, is estimated with ECCOv4r3, this is one subject of the current study.

The other goal of the present study is to examine how we can provide additional information about $\kappa_\rho$ using either observational estimates of $\kappa_\rho$ itself or a tracer–e.g., oxygen–from observations in ECCO's misfits. $\kappa_\rho$ products have been shown to agree well with each other (*Whalen et al.*, 2015; *de Lavergne et al.*, 2020). However, because $\kappa_\rho$ is derived and not measured, a parameter and state estimation system would need to account for both their structural and measurement errors, and their structural uncertainties are not yet well-understood. This is a potential problem because it is not clear how to weight these data
when constraining the model and conservatively large uncertainties would place little to no constraints on the model. An alternative approach to constraining $\kappa_\rho$ is to find a quantity measured with in situ observations–e.g., a tracer, as proposed here–that provides information about $\kappa_\rho$. Passive transient tracers are known to provide information about ocean mixing (*Mecking et al.*, 2004; *Trossman et al.*, 2014; *Shao et al.*, 2016). However, their concentrations tend to be difficult to detect below a couple thousand meters depth and are not monitored as well as biogeochemical tracers such as dissolved oxygen. Dissolved oxygen has

vertical gradients that can be resolved by most ocean models in the open ocean. Through altering oxygen saturation and ventilation rates, mixing likely plays an important role in controlling the dissolved oxygen concentrations and volumes of tropical oxygen minimum zones (OMZs) (*Brandt et al.*, 2015; *Lévy et al.*, 2021; *Ito et al.*, 2022), the rate of future global deoxygenation (*Duteil and Oschlies*, 2011; *Palter and Trossman*, 2018; *Couespel et al.*, 2019), the abyssal-shadow zone overturning connectivity (*Holzer et al.*, 2021), and the upwelling of low-latitude waters as part of the meridional overturning circulation (*Talley*, 2013). Oxygen utilization rates within subtropical mode water in the North Atlantic Ocean strongly depend upon vertical mixing (*Billheimer et al.*, 2021). Along with temperature and salinity observations, oxygen concentrations help identify particular water masses because oxygen utilization often reflects how recently water has been ventilated by the thermocline (*Jenkins*, 1987). Oxygen concentrations are less numerous than temperature and salinity observations, but tracers have different sources and sinks, are in varying degrees of disequilibrium, and require different amounts of time to equilibrate–similar in argument to why multiple tracers are needed best constrain transit-time distributions (*Waugh et al.*, 2003). Thus, we assess the information that dissolved oxygen concentrations provide about $\kappa_\rho$ with ECCOv4r3–which has already incorporated information about temperature and salinity–in the present study.

Our two primary objectives are: 1) to test whether $\kappa_\rho$ calculated using ECCO agrees with $\kappa_\rho$ from observations, given incomplete temperature, salinity, and pressure observations; and 2) to assess whether dissolved oxygen concentrations and $\kappa_\rho$ from observations provide similar information about how to improve the agreement between $\kappa_\rho$ from ECCO and observations. In the Appendix, we present $\kappa_\rho$ from one example sequential data assimilation framework in order to contrast its potential issues with those of ECCO. We use $\kappa_\rho$ inferred from microstructure (*Waterhouse et al.*, 2014), derived from Argo floats (*Whalen et al.*, 2015) and CTD profiles (*Kunze*, 2017), and calculated from climatological and seafloor data (*de Lavergne et al.*, 2020) to determine whether the ECCO framework needs to improve its $\kappa_\rho$ using observational constraints (Sections 3.1 and 3.2). We then perform model experiments in forward plus adjoint mode to determine whether dissolved oxygen concentration data and observationally-derived $\kappa_\rho$ provide similar information about how to adjust $\kappa_\rho$ (Section 3.3). This will help determine whether $\kappa_\rho$ could be improved by including tracer data in the misfits of a future iterative ocean parameter and state estimation procedure.

## 2   Methods

### 2.1   Observationally-derived data products and measured data

#### 2.1.1   Diapycnal Diffusivities

$\kappa_\rho$ is routinely inferred from the velocity shear measured using microstructure profilers (*Waterhouse et al.*, 2014). We use microstructure-inferred $\kappa_\rho$–referred to as $\kappa_{\rho,micro}$ hereafter– (*Osborn*, 1980; *Lueck et al.*, 1997; *Gregg*, 1989; *Moum et al.*, 2002; *Waterhouse et al.*, 2014) to evaluate a model's $\kappa_\rho$. (We distinguish between "observations" that are measured quantities using in situ instruments and observationally-derived values, which use measured quantities and a theory to derive values. The former data have only measurement uncertainties, while the latter data have both measurement and structural uncertainties. We

further distinguish "observationally-inferred" values, which are from the currently accepted method of observing a quantity such as $\kappa_\rho$ but are not measured, and "observationally-derived" values because the latter data depend on a method that requires additional assumptions. These terms only apply to values calculated making use of observations.) $\kappa_{\rho,micro}$ are based on an expression for the isotropic turbulence field, which is proportional to the viscosity of water and the velocity shear resolved to dissipative scales (*Thorpe* (2007); and references therein). The depth ranges of the data collected by *Waterhouse et al.* (2014) go from the upper several hundred meters to the full water column. The profiles are seasonally biased at higher latitudes and span decades. There are thousands of vertical profiles from 24 different campaigns that comprise this data set, with samples being taken in North Pacific Ocean, North Atlantic Ocean, tropical Pacific, near Drake Passage, near the Kerguelen Plateau, and in the South Atlantic Ocean. Many of the profiles were taken in regions with both smooth and rough bottom topography. To compare the microstructure profiles with model output, the nearest neighbors to each model's grid are selected. A geometric average is taken for each profile because this is more representative than an arithmetic average for a small sample size and when the data are not normally distributed (*Manikandan*, 2011), like the log-normal distribution of $\kappa_\rho$ (*Whalen*, 2021).

We make use of multiple data sets for $\kappa_\rho$ derived from observations. Two of these data sets–listed in Table 1–are derived using a finescale parameterization; they contain values equatorwards of $75^o$S and $75^o$N and deeper than about about 250 meters because the theory does not yield accurate results in the presence of strong upper-ocean density variability (e.g., *D'Asaro* (2014)). $\kappa_\rho$ values are derived from finestructure observations of temperature, salinity, and pressure using a strain-based finescale parameterization, which has been developed and implemented in different ways (*Henyey et al.*, 1986; *Gregg*, 1989; *Polzin et al.*, 1995, 2014), but typically assumes a mixing efficiency of 0.2 (*St. Laurent and Schmitt*, 1999; *Gregg et al.*, 2018). The finescale parameterization assumes that 1) the production of turbulent energy at small scales is due to an energy transfer driven by wave-wave interactions down to a wave breaking scale; 2) nonlinearities in the equation of state, double diffusion, downscale energy transports, and mixing associated with boundary layer physics and hydraulic jumps are neglected; and 3) stationary turbulent energy balance exists where production is matched by dissipation and a buoyancy flux in fixed proportions (*Polzin et al.*, 2014). The implementation by *Whalen et al.* (2015) uses Argo data assumes a shear-to-strain variance ratio of 3 and a flux Richardson number of $R_f = 0.17$, and determines the fraction of turbulent production that goes into the buoyancy flux (and the rest for dissipation). The finestructure method is not expected to be valid in equatorial regions of the ocean, but nevertheless, the $\kappa_\rho$ product compares well with microstructure near the equator (*Whalen et al.*, 2015). We use the 2006-2014 climatology of *Whalen et al.* (2015)–referred to as $\kappa_{\rho,Argo}$ hereafter–which is a gridded product on an approximately $1^o \times 1^o$ horizontal grid and has three vertical levels: 250-500 meters, 500-1000 meters, and 1000-2000 meters depth. *Whalen et al.* (2015) found that 81% (96%) of their $\kappa_{\rho,Argo}$ product is within a factor of 2 (3) of the microstructure measurements. We use this as the basis for the factor of 2-3 uncertainty we cite hereafter.

In addition to the Argo-derived $\kappa_{\rho,Argo}$ product, there is ship-based Conductivity, Temperature, and Depth (CTD) hydrography-derived $\kappa_\rho$ (*Kunze*, 2017)–referred to as $\kappa_{\rho,CTD}$ hereafter–that uses the same finestructure parameterization as in the calculation of the $\kappa_{\rho,Argo}$ product (see Section 2.2). The vertical resolution of the $\kappa_{\rho,CTD}$ product is 256 meters and the horizontal resolution is the spacing between each CTD profile. Data are only included in the $\kappa_{\rho,CTD}$ product when the square of the buoyancy

frequency is greater than $10^{-7}$ rad$^2$ s$^{-2}$ and greater than the square of the Coriolis frequency, $\kappa_{\rho,CTD} < 3 \times 10^{-3}$ m$^2$ s$^{-1}$ is positive, and the depth is deeper than 400 meters.

One last product we use for observationally-derived $\kappa_\rho$–referred to as $\kappa_{\rho,tides}$ hereafter–is based on theory, a spectral parameterization for abyssal hills, and climatological products (*de Lavergne et al.*, 2020). This scheme accounts for the local breaking of high-mode internal tides and remote dissipation of low-mode internal tides. The four processes contributing to the mixing from this scheme include wave-wave interactions that attenuate low-mode internal tides, shoaling that breaks low-mode internal tides, dissipation of low-mode internal tides at critical slopes, and scattering of low-mode internal tides combined with generation of high-mode internal tides via abyssal hills. Note that these tidally-induced mixing process are not equivalent to the suite of internal wave-induced mixing processes that $\kappa_{\rho,Argo}$ and $\kappa_{\rho,CTD}$ account for. The gridded $\kappa_{\rho,tides}$ product is global, nominally $1/2^o$ horizontal resolution, and ranges from 10 to 250 meters in vertical resolution. A stratification field is provided in this product, which is the one we use for the remainder of this study (Section 2.1.3).

### 2.1.2 Dissolved oxygen

Because we have annual mean $\kappa_\rho$ products, we use the annual mean dissolved oxygen concentration climatology from the World Ocean Atlas (2013) for the remainder of our analysis. Any potential information that oxygen concentrations provide about $\kappa_\rho$ is likely through oxygen's vertical gradients because water masses–which tend to be relatively homogenous in oxygen concentrations–are eroded via diapycnal mixing along their peripheries. Thus, we show oxygen's vertical gradients, $\partial O_2/\partial z$ here. We compare $\partial O_2/\partial z$ (Figs. 1a,c,e) with the dissipation rates, $\epsilon = N^2 \kappa_\rho/0.2$ for stratification $N^2$ through the *Osborn* (1980) relationship, from the *Whalen et al.* (2015) product (Fig. 1b,d,f) at the same depth-averaged bins. $\partial O_2/\partial z$ is generally smaller in magnitude in many high-latitude and tropical regions (Figs. 1a,c,e), whereas the Argo-derived dissipation rates can be relatively large in these regions, with the exception of locations in the Southern Ocean (Figs. 1b,d,f). $\partial O_2/\partial z$ is relatively large and positive landward of the Gulf Stream, in the Chukchi and Beaufort Seas, near the Norwegian coast, off the southern coast of India, near the equator in the Atlantic and western Pacific Oceans, and in the Southern Hemisphere's subtropical gyres of the Pacific and Indian Oceans between 250-500 meters depth (Fig. 1a). The largest positive $\partial O_2/\partial z$ are between the subpolar regions and the equator at deeper depths (Figs. 1c,e). The dissipation rates are relatively small in many of these regions (Figs. 1b,d,f). Exceptions to the inverse relationship between $\partial O_2/\partial z$ and the dissipation rates tend to be in the vicinity of intensified jets, likely because lateral exchanges of oxygen concentrations become more important in these regions. Where data exist for both data products, the spatial correlation between $\partial O_2/\partial z$ and the dissipation rates is about $-0.2$ and increases in magnitude on coarser grids. This indicates a possibly non-local relationship between $\partial O_2/\partial z$ and dissipation rates. The spatial correlation between $\partial O_2/\partial z$ and $\kappa_{\rho,Argo}$ is smaller in magnitude–about $-0.1$–which motivates further consideration of the information provided by $N^2$.

### 2.1.3 Stratification

We use an observational climatology for $N^2$, as provided by the *de Lavergne et al.* (2020) data set. $N^2$ is generally about $10^{-7} - 10^{-5}$ s$^{-2}$, with lower values in high-latitude and deeper regions and higher values in the thermocline and in shallow

water areas–which skew its global average (standard deviation) below the mixed layer to about $1.2 \times 10^{-4}$ ($3 \times 10^{-3}$) s$^{-2}$.
The vertical gradients in $N^2$ are typically between $-10^{-5}$ and $10^{-5}$ m$^{-1}$ s$^{-2}$ and have an average value (standard deviation) of about $-10^{-7}$ ($4 \times 10^{-6}$) m$^{-1}$ s$^{-2}$, with its largest magnitudes on continental shelves–high-latitude ones in particular–and in the eastern equatorial Pacific Ocean. The spatial correlation between the annual mean vertical gradients in oxygen (Figs. 1a,c,e) and the annual mean vertical gradients in $N^2$ is about 0.25, which suggests that stratification is one candidate factor in explaining why oxygen concentrations are correlated with $\kappa_\rho$ and $\epsilon$. However, we do not test this with model experiments that incorporate information about $N^2$–which directly compare $N^2$ from our model and observations–because the vertical resolution of our ocean model is so much coarser than that of observations. Instead, we run a set of model experiments that compare oxygen concentrations, $\kappa_\rho$, or $\epsilon$. We perform these model experiments to further explore the potential information that oxygen concentrations provide about $\kappa_\rho$ and indirectly infer–via the *Osborn* (1980) relation–the possible role of stratification.

## 2.2 Modeling system

We use the Estimating the Circulation & Climate of the Ocean (ECCO) framework in our analysis. ECCO uses a time-invariant but spatially varying background $\kappa_\rho$ field–$\kappa_{\rho,bg}$ hereafter–calculated with a parameter and state estimation procedure, and $\kappa_\rho$ associated with temperature and salinity are assumed to be identical. Details about the model simulations we perform are summarized in Table 2.

### 2.2.1 ECCO

The modeling system used here is ECCOv4r3 (*Fukumori et al.*, 2017). The underlying ocean-sea ice model is based on the Massachusetts Institute of Technology general circulation model (MITgcm), which is a global finite volume model. The EC-COv4r3 global configuration uses curvilinear Cartesian coordinates (*Forget et al.* (2015a) - see their Figs. 1-3) at a nominal $1^o$ ($0.4^o$ at equator) resolution and rescaled height coordinates (*Adcroft and Campin*, 2004) with 50 vertical levels and a partial cell representation of bottom topography (*Adcroft et al.*, 1997). The MITgcm uses a dynamic/thermodynamic sea ice component (*Menemenlis et al.*, 2005; *Losch et al.*, 2010; *Heimbach et al.*, 2010) and a nonlinear free surface with freshwater flux boundary conditions (*Campin et al.*, 2004). The wind speed and wind stress are specified as 6-hourly varying input fields over 24 years (1992-2015). Average adjustments to the wind stress, wind speed, specific humidity, shortwave downwelling radiation, and surface air temperature are re-estimated and then applied over 14-day periods. These adjustments are based on estimated prior uncertainties for the chosen atmospheric reanalysis (*Chaudhuri et al.*, 2013), which is ERA-Interim (*Dee et al.*, 2011). The net heat flux is then computed via a bulk formula (*Large and Yeager*, 2009). The ocean variables, on the other hand, do not get periodically adjusted. A parameterization of the effects of geostrophic eddies (*Gent and McWilliams*, 1990) is used. Mixing along isopycnals is accounted for according to the framework provided by *Redi* (1982). Vertical mixing is the sum of diapycnal mixing and the vertical component of the along-isopycnal tensor, where diapycnal mixing is determined according to the *Gaspar et al.* (1990) mixed layer turbulence closure and estimated $\kappa_{\rho,bg}$. Convective adjustment does not act through $\kappa_\rho$ in the MITgcm. Here, $\kappa_\rho$ represents a combination of processes, including–but potentially not limited to–internal wave-induced mixing. $\kappa_{\rho,bg}$, the Redi coefficient, and the Gent-McWilliams coefficient are time-independent because of the under-determined problem of

inverting for initial conditions and model parameters would be even more under-determined if they were allowed to vary in time–explained below. In order to simulate oxygen concentrations, tracers are carried using Biogeochemistry with Light, Iron, Nutrients and Gases (BLING) model (*Galbraith et al.*, 2015). BLING is an intermediate complexity biogeochemistry model that uses eight prognostic tracers and parameterized, implicit representations of iron, macronutrients, and light limitation and photoadaptation. BLING has been shown to compare well with the Geophysical Fluid Dynamics Laboratory's full-complexity biogeochemical model, TOPAZ (*Galbraith et al.*, 2015), and has been adapted for use in the MITgcm with its adjoint (*Verdy and Mazloff*, 2017).

Initial conditions and model parameters for the runs performed here are from ECCOv4r3. The least-squares problem solved by the ECCO model uses the method of Lagrange multipliers through iterative improvement, which relies upon a quasi-Newton gradient search (*Nocedal*, 1980; *Gilbert and Lemarechal*, 1989). Algorithmic (or automatic) differentiation tools (*Griewank*, 1992; *Giering and Kaminski*, 1998) have allowed for the practical use of Lagrange multipliers in a time-varying non-linear inverse problem such as ocean modeling, eliminating the need for discretized adjoint equations to be explicitly hand-coded. Contributions of observations to the model-data misfit function are weighted by best-available estimated data and model representation error variance (*Wunsch and Heimbach*, 2007). The observational data included in the ECCO state estimation procedure are discussed in *Forget et al.* (2015a) and *Fukumori et al.* (2017). These data include satellite-derived ocean bottom pressure anomalies, sea ice concentrations, sea surface temperatures, sea surface salinities, sea surface height anomalies, and mean dynamic topography, as well as profiler- and mooring-derived temperatures and salinities (*Fukumori et al.*, 2017). No ocean subgrid-scale transport parameter, mixing parameter, or biogeochemical tracer data are included in the model's misfits during the parameter and state estimation procedure. The control variables that are inverted for iteratively by ECCO are listed in Table 3, which include the ocean subgrid-scale transport and mixing parameters–e.g., $\kappa_{\rho,bg}$. The error covariances for each of the ocean subgrid-scale transport and mixing parameters are specified by imposing a smoothness operator (*Weaver and Courtier*, 2001) at the scale of three grid points–decorrelation length scale diameter of $\sim 100$ km–which allows for the dynamical model to regionally adjust from the information provided by observations (*Forget et al.*, 2015b). Fifty-nine iterations of the parameter and state estimation procedure–referred to as the "optimization" run hereafter–were performed to arrive at the ECCOv4r3 solution we start from for our experiments. The resulting $\kappa_{\rho,bg}$ field in the ECCOv4r3 solution–plus the *Gaspar et al.* (1990) contribution–will be referred to as $\kappa_{\rho,ECCO}$ hereafter and is shown in Fig. 2–depth-averaged below the model's average mixed layer depth. Note that the initial guess for $\kappa_{\rho,bg}$ in ECCO is $10^{-5}$ m$^2$ s$^{-1}$ and in the absence of observation-driven adjustments, $\kappa_{\rho,bg}$ in ECCO remains at or is close to its initial value in the ECCOv4r3 solution, at least in its depth-average. Also note that in regions away from ocean boundary layers, $\kappa_{\rho,ECCO}$ is approximately $\kappa_{\rho,bg}$ in ECCO. $\kappa_{\rho,ECCO}$ is elevated in regions that undergo deep convection, near the margins of continental shelves and intensified jets, and in the Indonesian Throughflow. We will later compare $\kappa_{\rho,ECCO}$ with $\kappa_{\rho}$ from the first iteration of the same optimization run with ECCO, which will be referred to as $\kappa_{\rho,ECCO,0}$ hereafter. If $\kappa_{\rho,ECCO,0}$ is in closer agreement with $\kappa_{\rho}$ from observational products than $\kappa_{\rho,ECCO}$, then errors in $\kappa_{\rho,ECCO}$ are likely being compensated by errors in other control variables beyond the first iteration of the model's optimization run.

We run ECCO in two configurations: 1) a "re-run," where all control variables are set to be their estimated values from ECCOv4r3 in forward mode–sometimes referred to as an ocean-only free run–and 2) an "adjoint sensitivity" run of the parameter and state estimate in forward plus adjoint modes, where data are included in the model's misfits but not technically "assimilated" because the model input parameters do not change as the model runs. An adjoint sensitivity is essentially the sensitivity of one variable to another, computed by making use of the model's adjoint. Formally, an adjoint sensitivity is $\partial J / \partial X$, where the cost function $J$ is a sum of weighted misfits to observations and a control variable $X$ is a variable that the model estimates by making use of its adjoint and observations–see Section 2.2.2. The adjoint sensitivities provide information about which directions the model's input parameters should change $X$ in order to minimize $J$. *Masuda and Osafune* (2021) showed some examples of adjoint sensitivities of several model parameters in their ocean state estimate to a vertical mixing parameter (slightly different from $\kappa_\rho$). We also compute adjoint sensitivities in the present study, but using ECCO with respect to $X = \kappa_\rho$.

The following is a summary of the ECCO experiments we run (Table 2):

- **E-CTRL** - a forward ECCOv4 simulation that uses the parameters from ECCOv4r3; this simulation can be referred to as a "re-run"

- $E_O$ - an adjoint sensitivity (with respect to $X = \kappa_\rho$) experiment in which only oxygen concentrations from the World Ocean Atlas (2013) climatology are included in the misfit function $J$

- $E_\kappa$ - an adjoint sensitivity (with respect to $X = \kappa_\rho$) experiment in which only the base-10 logarithm of the $\kappa_{\rho,micro}$ data set, $\kappa_{\rho,Argo}$ and $\kappa_{\rho,CTD}$ products, or $\kappa_{\rho,tides}$ product are included in the misfit function $J$

- $E_\epsilon$ - an adjoint sensitivity (with respect to $X = \epsilon$) experiment in which only the base-10 logarithm of the $\epsilon_{Argo} = \kappa_{\rho,Argo} N^2 / 0.2$ and $\epsilon_{CTD} = \kappa_{\rho,CTD} N^2 / 0.2$ products or $\epsilon_{tides} = \kappa_{\rho,tides} N^2 / 0.2$ product are included in the misfit function $J$

The difference between experiment $E_\kappa$ and $E_\epsilon$ is that the latter uses observationally-derived dissipation rates, $\epsilon = N^2 \kappa_\rho / 0.2$ instead of $\kappa_\rho$, in the misfit function via Eq (2). We do not perform the experiment $E_\epsilon$ with microstructure data included in the model's misfit function because of the sparsity of those data. We analyze the adjoint sensitivities with dissipation rates in the misfit function ($E_\epsilon$ in Table 2) in order to assess whether the stratification–a multiplying factor between $\kappa_\rho$ and the dissipation rates according to *Osborn* (1980)–provides information about $\kappa_\rho$. Due to the relatively coarse vertical resolution of ECCO compared with observations, we do not directly compare $N^2$ from ECCO with $N^2$ from observations in another adjoint sensitivity experiment.

We take the ECCOv4r3 solution as the reference state for each of our simulations. We perform an adjoint calculation in each experiment, except for E-CTRL. The adjoint sensitivities are accumulated and averaged over the full integration period. Only one year was run for each of the adjoint simulations, but our results are not qualitatively sensitive to the run length–which is at least partially because we are using time-invariant climatologies. The time-dependence of the $\kappa_\rho$ sensitivities from $E_\kappa$ is weak due to the lack of time-dependence of the observations included in the misfits–$\kappa_\rho$ and oxygen concentrations; initial condition

sensitivities are stronger. Thus, our simulations will suffice to demonstrate whether the inclusion of a biogeochemical tracer in the model's misfits can reduce the bias in $\kappa_\rho$.

We begin $E_O$ from a previously-derived product that has been spun-up from an initial climatology (Fig. 3a) derived from World Ocean Atlas (2013). Thus, disagreements with the World Ocean Atlas (2013) are due to model drift. The depth-averaged differences between the uninterpolated World Ocean Atlas (2013) product and the initial conditions for oxygen concentrations in our ECCO run using BLING are shown in Fig. 3b. The differences are largest in the Arctic Ocean, northeastern Pacific Ocean, and near the coasts, particularly on the eastern side of the American continent, the southwestern side of the African

continent, around the Kuroshio/Sea of Japan region, along almost every coastline of Oceania, and in the Mediterranean Sea (Fig. 3b). Point-wise differences between the initial conditions for oxygen concentrations in ECCO and the World Ocean Atlas (2013) product are shown in Fig. 3c, which suggests that there is strong agreement between the two fields. Where there are disagreements, the initial conditions for oxygen concentrations in ECCO are more often too small (particularly in the Atlantic Ocean, as shown in Fig. 3b) than too large. These differences are likely due to the deficiencies in model resolution, the sparse

observations in regions such as the Arctic Ocean, the locations of sea ice (*Bigdeli et al.*, 2017), and the parameterization of the tracer air-sea fluxes (e.g., *Atamanchuk et al.* (2020)). We need to consider the spatial patterns shown in Fig. 3b when interpreting the signs of the adjoint sensitivities.

### 2.2.2   ECCO adjoint sensitivity analyses

Short of including a particular data set (e.g., dissolved oxygen concentrations) in the misfits of a new optimization run of

ECCO, we assess whether the inclusion of a particular data set in the model's misfits could lead to a more accurate estimate of a control variable that can be observed (e.g., $\kappa_\rho$). In order to understand whether $\kappa_\rho$ could be estimated more accurately through the inclusion of oxygen concentrations in the model's misfit, we need to further explain the details of our adjoint sensitivity experiments with ECCO. We define the objective (or cost) function here to more formally explain what the adjoint sensitivity is. ECCO calculates the cost function to be minimized, $J$, (*Stammer et al.*, 2002) as–focusing here only on the observational

misfit terms while omitting regularization terms for the control variables:

$$J = \sum_{t=1}^{t_f} [\mathbf{y}(t) - \mathbf{S}\tilde{\mathbf{x}}(t)]^T \mathbf{W}(t)[\mathbf{y}(t) - \mathbf{S}\tilde{\mathbf{x}}(t)] \tag{1}$$

where $t_f$ is the final time step, $\tilde{\mathbf{x}}$ is the model-based estimate of the state vector $\mathbf{x}$, $\mathbf{S}$ is the observation matrix that relates the model state vector to observed variables $\mathbf{y}$ (such that $\mathbf{S}\tilde{\mathbf{x}}$ is the model-based estimate of the observables $\mathbf{y}$), and $\mathbf{W}$ is the weight (inverse square of approximate uncertainties accounting for measurement and representation errors) of the observations.

In each of our adjoint sensitivity experiments, the data vector $\mathbf{y}$ only contains the data set specific to the experiment (see Table 2) so we emphasize here that $J$ is different for each of our experiments. The uncertainties in $\kappa_{\rho,ECCO}$ in $E_\kappa$ are set to be three times the values of the observationally-derived $\kappa_\rho$ because of the level of agreement between $\kappa_{\rho,Argo}$ and $\kappa_{\rho,micro}$ (*Whalen et al.*, 2015). The uncertainties in oxygen concentrations in $E_O$ are set to be 2% of the values of the measured dissolved oxygen concentrations.

The adjoint sensitivities computed in this study are the derivatives of $J$ in Eq. 1 with respect to $\kappa_\rho$. We consider evaluating directions in the control space in which to improve $\kappa_\rho$ through improvement of $\kappa_{\rho,bg}$, given the control vector from the ECCOv4r3 solution. While the adjoint sensitivities of $J$ to the control space in experiment $E_O$ must be computed online, those in $E_\kappa$ can either be computed online or offline using an analytical equation (see below). The adjoint sensitivity run with $\kappa_\rho$ included in the misfit calculation of experiment $E_\kappa$ can be calculated offline using output from the E-CTRL run instead of

being calculated online as follows:

$$\frac{\partial J}{\partial X} = -2\frac{(X_{obs} - X_{model})}{\sigma_X^2}. \tag{2}$$

Here, $X = \kappa_\rho$ is the control variable, $X_{obs}$ is the observationally-derived value of $X$ described in the previous section, $X_{model}$ is the value that ECCO estimates for $X$, and $\sigma_X$ is taken to be $3X_{obs}/1.96$ (or the base-10 logarithm of this in the case of $\kappa_\rho$) due to the factor of 3 uncertainty corresponding to an approximate 95% confidence interval in *Whalen et al.* (2015). For $X_{model}$,

we use the offline values calculated from the E-CTRL run following Eq. 2. While this assumes a diagonal $\mathbf{W}$ and minimal impact of the smoothing operator applied over a decorrelation length scale diameter of $\sim 100$ km, the offline Eq. 2 and online sensitivities have been verified to be in agreement.

Because the observations of $\kappa_\rho$ here are not direct measurements, we first need to show that observationally-derived $\kappa_\rho$ has a smaller bias with respect to independent observations than the model's estimate of $\kappa_\rho$. We devote the first portion of our

study to determining whether $|\kappa_{\rho,Argo} - \kappa_{\rho,micro}| < |\kappa_{\rho,ECCO} - \kappa_{\rho,micro}|$ (and, by extension, $\kappa_{\rho,CTD}$ in place of $\kappa_{\rho,Argo}$) is true. We do this because $\kappa_{\rho,micro}$ is limited in its spatial coverage compared to $\kappa_{\rho,Argo}$, $\kappa_{\rho,CTD}$, and $\kappa_{\rho,tides}$. Also, $\kappa_{\rho,Argo}$ and $\kappa_{\rho,CTD}$ are still limited spatial coverage relative to dissolved oxygen concentrations. While $\kappa_{\rho,tides}$ has global spatial coverage, its measurement plus structural uncertainties are not well-known compared to dissolved oxygen concentrations. The data product with higher accuracy (dissolved oxygen concentrations) will have larger weights ($\mathbf{W}$ in Eq. 1) and thus will exert

more influence in constraining $\kappa_{\rho,ECCO}$–bringing it closer to microstructure values. So if we can show that the adjustments to $\kappa_\rho$ in ECCO are similar, whether we provide information from observationally-derived $\kappa_\rho$ or a measured tracer with relatively small uncertainties (dissolved oxygen concentrations), then we would include the tracer in the misfits.

One problem with doing a direct comparison of the adjustments is that the uncertainties in observationally-derived $\kappa_\rho$ products are large, so we first quantify the extent to which the adjoint sensitivities from two runs (here, $E_\kappa$ and $E_O$) have

the same sign at each location and depth. Specifically, we inspect whether $\partial J/\partial \kappa_\rho$ has the same sign in $E_\kappa$ and $E_O$ where $|\kappa_{\rho,Argo} - \kappa_{\rho,ECCO}|$ is significantly different from zero (i.e., $\kappa_{\rho,Argo}$ is more than a factor of three greater or less than a factor of three smaller than $\kappa_{\rho,ECCO}$). We are interested in regions where $\kappa_\rho$ is significantly erroneous and where the errors in oxygen are due to errors in the physics (e.g., $\kappa_\rho$), not initial conditions; hence, these choices. We perform these comparisons in regions where the difference between the observationally-derived $\kappa_\rho$ products and $\kappa_{\rho,ECCO}$ exceeds three times the observational

products' magnitudes (i.e., statistically distinguishable from zero). Because model errors unrelated to $\kappa_\rho$ can confound the correlations between the adjoint sensitivities from $E_\kappa$ and $E_O$, we additionally look at regions where the difference between oxygen concentrations from the model and the World Ocean Atlas (2013) is relatively small to determine whether oxygen

concentrations guide the state estimate's control space to improve the magnitude of $\kappa_\rho$. In this subset of regions, we calculate the correlations between the adjoint sensitivities from $E_\kappa$ and $E_O$–despite the difficulty with determining their significance.

To investigate whether the results are sensitive to our assumptions about the signal-to-noise ratio of our data–through $\mathbf{W}$ in Eq. 1–we additionally perform Monte Carlo simulations for the adjoint sensitivities from $E_\kappa$–using three different data sets for $\kappa_\rho$: $\kappa_{\rho,micro}$, $\kappa_{\rho,Argo}$ together with $\kappa_{\rho,CTD}$, and $\kappa_{\rho,tides}$. In the Monte Carlo simulations, at each location and depth, we randomly sample $\kappa_\rho$ values within its uncertainty–$\sigma_\kappa$–simultaneously with randomly sampled values of $\sigma_\kappa$ between a factor of 2-3 of $\kappa_\rho$. This accounts for the uncertainties in the $\kappa_\rho$ products in both the numerator–corresponding to uncertainties in the observationally-derived estimates–and denominator of Eq (2)–corresponding to the weights. With each of the 10,000 samples of $\kappa_\rho$ and $\sigma_\kappa$, we recompute the adjoint sensitivity for $E_\kappa$ and then its correlation with that for $E_O$. With these Monte Carlo simulations, we report the maximum possible correlation between each experiment's adjoint sensitivities.

## 3 Results

We first show that the disagreements between $\kappa_\rho$ from ECCO and $\kappa_\rho$ from various observations are larger than the observations' approximate 95% confidence intervals. Then we analyze results from pairs of adjoint sensitivity runs: one with misfits to observed $\kappa_\rho$ derived from the finescale parameterization and the other with misfits to observed $O_2$. We use these results to investigate the potential to use $O_2$ as a constraint for improving $\kappa_{\rho,ECCO}$ in a future optimization. We then compare the results of the adjoint sensitivity runs using misfits in $\kappa_\rho$ with ones using misfits in $\epsilon$ to infer a potential role of stratification in any information that $O_2$ provides about $\kappa_\rho$. In the Appendix, we show there is general agreement between $\kappa_\rho$ from observations and a free-running earth system model that calculates a physically-motivated parameterization for $\kappa_\rho$, but poor agreement between $\kappa_\rho$ from observations and a sequential ocean data assimilation system based on the same earth system model (Fig. A1a).

### 3.1 Model-inverted vs microstructure-inferred $\kappa_\rho$ comparisons

We compare the average $\kappa_{\rho,micro}$ profile that is comprised of 24 campaigns' worth of data (*Waterhouse et al.*, 2014) (see their Fig. 6; black curves in Figs. 4a and 5) with average $\kappa_{\rho,ECCO}$ profiles from two different iterations and the $\kappa_{\rho,Argo}$ product (*Whalen et al.*, 2015). The locations of the microstructure measurements are shown in Fig. 2 (black X's). We also compare the average $\kappa_{\rho,tides}$ profile (*de Lavergne et al.*, 2020) (see their Fig. 2e; black curve in Fig. 4b) with the average $\kappa_{\rho,ECCO}$ profile from the final iteration.

The average $\kappa_{\rho,ECCO,0}$ profile–i.e., the first adjusted initial guess of $\kappa_{\rho,ECCO}$–is typically smaller than the microstructure profile, particularly at 1000 m where the difference is approximately an order of magnitude (Fig. 4a). At iteration 59 (which is the ECCOv4r3 solution), the difference between $\kappa_{\rho,ECCO}$ and $\kappa_{\rho,micro}$ decreases. However, agreement between the average profiles of $\kappa_{\rho,ECCO}$ and $\kappa_{\rho,micro}$ is still worse than the agreement between $\kappa_{\rho,Argo}$ and $\kappa_{\rho,micro}$. The agreement between $\kappa_{\rho,Argo}$ and $\kappa_{\rho,micro}$ at each of the three depth bins is well within a factor of three (dotted black curves in Fig. 4a) and the spatial standard deviation of $\kappa_{\rho,micro}$ (dashed black curves in Fig. 4a). The agreement between the average profiles of $\kappa_{\rho,ECCO}$ and $\kappa_{\rho,tides}$ is poor, with $\kappa_{\rho,ECCO}$ typically too small, notably so at deeper depths (Fig. 4b). $\kappa_{\rho,ECCO}$ includes

internal-wave-induced mixing as well as potentially numerical diffusion. However, numerical diffusion cannot explain the errors in $\kappa_{\rho,ECCO}$ where $\kappa_\rho$ is too small in the model relative to the observationally-derived products because numerical diffusion would increase $\kappa_{\rho,ECCO}$. In these regions, one likely explanation is that errors in other model parameters (e.g., the Redi coefficients) compensate for the errors in $\kappa_{\rho,bg}$.

We also compare $\kappa_{\rho,ECCO}$ and $\kappa_{\rho,ECCO,0}$ profiles with $\kappa_{\rho,micro}$ from 16 example campaigns in Fig. 5. In some regions,
the $\kappa_{\rho,ECCO}$ and $\kappa_{\rho,ECCO,0}$ profiles are constant ($10^{-5}$ m$^2$ s$^{-1}$, the default background value) because ECCO does not sufficiently resolve the bathymetry so we exclude those from Fig. 5. We also exclude some others, for example, in the subpolar North Atlantic because temporal variations in $\kappa_\rho$ can be large there (Fig. A1b). The $\kappa_{\rho,ECCO}$ profiles (blue curves in Fig. 5) and $\kappa_{\rho,ECCO,0}$ profiles (grey curves in Fig. 5) are often within the approximate (factor of three) uncertainties in the $\kappa_{\rho,micro}$ profiles (dashed black curves in Fig. 5), but not always. Without taking an average over all of the campaigns, there can be
large regional disagreements between the model and observations. Also, the $\kappa_{\rho,ECCO}$ profiles are not always closer to the $\kappa_{\rho,micro}$ profiles than the $\kappa_{\rho,ECCO,0}$ profiles. This suggests that performing more iterations of the optimization of ECCO is not necessarily going to lead to more accurate representation of $\kappa_\rho$ with the current data constraints.

### 3.2    Model-inverted vs finescale parameterization-derived $\kappa_\rho$ comparisons

We next show $\kappa_{\rho,Argo}$ and $\kappa_{\rho,tides}$ as well as how they contrast with $\kappa_{\rho,ECCO}$ because this highlights the spatial patterns of
the adjoint sensitivities in $E_\kappa$ (see later). The ratio between the $\kappa_{\rho,Argo}$ product (Figs. 6a,c,e) and $\kappa_{\rho,ECCO}$ varies throughout the globe (Figs. 6b,d,f). Red (Blue) areas in Figs. 6b,d,f indicate locations where Argo-derived $\kappa_{\rho,Argo}$ is smaller (larger) than $\kappa_{\rho,ECCO}$. The percent of volume where $\kappa_{\rho,ECCO}$ is at least an order of magnitude different from $\kappa_{\rho,Argo}$ is 43.8%. The values of $\kappa_{\rho,ECCO}$ are smaller than those in the Argo- and hydrography-derived observational product in the Kursoshio Extension (500-1000 meters depth), subpolar North Atlantic (500-1000 meters depth), Southern Ocean, equatorial regions in the Atlantic,
and shallow (250-500 meters depth) Indian and eastern Pacific Oceans (Figs. 6b,d,f). In contrast, $\kappa_{\rho,ECCO}$ tends to be too large relative to the $\kappa_{\rho,tides}$ product (Figs. 7a,c,e) in the Atlantic Ocean below 500 meters depth as well as in many near-equatorial and subpolar regions and $\kappa_{\rho,ECCO}$ tends to be too small everywhere else (Figs. 7b,d,f). Regardless of the observational product, the $\kappa_{\rho,ECCO}$ field is comparatively large in many of the model's near-equatorial regions, where the intermittency of strong mixing events is likely not captured–even in a time-mean sense–because the ECCO framework uses a smoother. However,
the fidelity of each observational product is unknown near the equator. The fact that $\kappa_{\rho,ECCO}$ and each observational product disagree within the deep mixed layers at high latitudes is not consequential for tracer transport. The errors in $\kappa_{\rho,bg}$ could be partially compensating for errors in the vertical component of the along-isopycnal diffusivity tensor, erroneous air-sea fluxes due to inconsistencies between the sea surface and atmospheric forcing fields, and/or the presence of numerical diffusion.

Incomplete historical observations–of temperature, salinity, and pressure–are currently insufficient to accurately estimate
$\kappa_{\rho,bg}$–approximately $\kappa_{\rho,ECCO}$ away from the boundary layers, where we compare with observational products. Even the abundance of Argo data in the upper 2000 meters have not been enough to calculate a realistic $\kappa_{\rho,ECCO}$ in the upper 2000 meters. The sparsity of the observations below 2000 meters depth, in high latitude regions, and in some near-coastal areas–where internal wave-induced mixing can be important–is relevant because complete observational coverage of the ocean's tempera-

ture, salinity, and pressure could, in principle, better constrain $\kappa_{\rho,bg}$ using inverse modeling (*Groeskamp et al.*, 2017). However, the lack of time-dependence of $\kappa_{\rho,bg}$, the presence of numerical mixing, and joint estimation of many under-determined parameters in ECCO could also lead to erroneous $\kappa_{\rho,ECCO}$. These are some reasons why values of $\kappa_{\rho,ECCO}$ do not agree well with $\kappa_\rho$ from observations–$\kappa_{\rho,micro}$, $\kappa_{\rho,Argo}$, $\kappa_{\rho,CTD}$, or $\kappa_{\rho,tides}$.

### 3.3 Adjoint sensitivities in ECCO

Because the data that are currently included in ECCO's misfits are insufficient for $\kappa_{\rho,ECCO}$ to match $\kappa_{\rho,micro}$, $\kappa_{\rho,Argo}$, $\kappa_{\rho,CTD}$, or $\kappa_{\rho,tides}$, including additional variables controlled by mixing in the model's misfits may assist in further improving the modeled mixing parameters. Oxygen is a candidate since its distribution is, in part, determined by the local $\kappa_\rho$. To test this, we run multiple adjoint sensitivity experiments in which either observationally-derived $\kappa_\rho$ or oxygen is included in the misfit calculation to guide constraints on $\kappa_\rho$. We expect that the signs of sensitivities agree most in regions away from where air-sea fluxes and transport of oxygen–e.g., by intensified jets–are large. One of these regions is the subtropical North Atlantic Ocean, away from the Gulf Stream Extension. Further, we expect to find more agreement between the signs of sensitivities in tropical OMZs and other (sub)tropical regions because of the known importance of diapycnal mixing in these regions.

We show the adjoint sensitivity calculations using Eq. 2 for $\kappa_\rho$ misfits (experiment $E_\kappa$ in Table 2) in Fig. 8 using $\kappa_{\rho,Argo}$ and $\kappa_{\rho,CTD}$; these are later compared with the sensitivities for oxygen concentration misfits in experiment $E_O$. A positive adjoint sensitivity implies that the misfit can be reduced by decreasing $\kappa_{\rho,ECCO}$. The signs of $\partial J / \partial \kappa_\rho$ using $\kappa_{\rho,Argo}$ and $\kappa_{\rho,CTD}$ (Fig. 8a) are consistent with the signs of local disagreement with microstructure (Figs. 4a and 5) and Argo-derived observations (Fig. 6b,d,f), by construction. Because $\kappa_{\rho,ECCO}$ tends to be very large inside mixed layers, $\partial J / \partial \kappa_\rho$ tends to be positive and larger at many locations in the subpolar latitudes where there are deep mixed layers in the model but possibly not in the real ocean; conversely, $\partial J / \partial \kappa_\rho$ can be negative where the mixed layer depth is too shallow in ECCO, but this isn't the only reason for $\partial J / \partial \kappa_\rho < 0$. The large positive values of $\partial J / \partial \kappa_\rho$ within the mixed layer and some other regions overwhelm the zonal averages in favor of positive values (Fig. 8c). $\kappa_{\rho,ECCO}$ needs to be decreased in many regions at depths shallower than 500 meters to agree better with $\kappa_{\rho,Argo}$ and $\kappa_{\rho,CTD}$ (yellow regions in Figs. 6b,d,f), but microstructure measurements (X's in Fig. 2) were often taken in locations where $\kappa_{\rho,ECCO}$ should be increased (blue regions in Figs. 6b,d,f) or stay the same. Microstructure measurements tend to be regions where there are prominent topographic features and where the centers of subtropical gyres are found, which–judging from the predominant signs of disagreement in Figs. 4a and 5 versus Figs. 6b,d,f–aren't representative of the ocean where Argo measurements were taken.

We next compare $\partial J / \partial \kappa_\rho$ from $E_\kappa$ using $\kappa_{\rho,Argo}$ and $\kappa_{\rho,CTD}$ with $\partial J / \partial \kappa_\rho$ from $E_O$. In $E_O$, $\partial J / \partial \kappa_\rho$ is generally negative in subtropical regions (Figs. 8b,d). Overall, the locations of the positive/negative signs of $\partial J / \partial \kappa_\rho$ are not the same everywhere between the $E_\kappa$ and the $E_O$ experiments, but they agree in many regions (Figs. 8a,c and Figs. 8b,d) using $\kappa_{\rho,Argo}$ and $\kappa_{\rho,CTD}$, which account for nearly two-thirds (three-fourths) of the ocean's volume where they can be compared (in the subtropics, between $20^o$-$50^o$N/S; Fig. 9; Table 4). The ocean basin with the highest percent volume of agreement in adjoint sensitivity signs between $E_\kappa$ and $E_O$ is the subtropical North Atlantic Ocean, with nearly $85\%$ volume agreement. The subtropical South Atlantic Ocean is the only subtropical basin with less than half of its volume in agreement in adjoint sensitivity sign. In general,

the tropical regions (between $20^o$S and $20^o$N) have adjoint sensitivity signs in lesser agreement than the subtropical regions and the subpolar regions (poleward of $50^o$N/S) are the regions with the lowest percent volume agreements in adjoint sensitivity signs.

We also show the adjoint sensitivity calculations using Eq. 2 for $\kappa_\rho$ misfits (experiment $E_\kappa$ in Table 2) in Fig. 10 using $\kappa_{\rho,tides}$ and compare $\partial J / \partial \kappa_\rho$ from $E_\kappa$ using $\kappa_{\rho,tides}$ with $\partial J / \partial \kappa_\rho$ from $E_O$. The signs of $\partial J / \partial \kappa_\rho$ using $\kappa_{\rho,tides}$ (Fig. 10a) are consistent with the signs of local disagreement with the *de Lavergne et al.* (2020) product (Figs. 4b and 7b,d,f), by construction. The positive values of $\partial J / \partial \kappa_\rho$ outside of the Atlantic Ocean and in the vicinity of intensified jets overwhelm the zonal averages in favor of positive values at most depths (Fig. 10c). $\kappa_{\rho,ECCO}$ needs to be decreased in many regions at depths shallower than about 2500 meters to agree better with $\kappa_{\rho,tides}$ (Fig. 4b; yellow regions in Figs. 7b,d,f). The regions where $\kappa_{\rho,ECCO}$ needs to be increased become dominant closer to the seafloor, particularly in the Atlantic Ocean. The signs of $\partial J / \partial \kappa_\rho$ from $E_\kappa$ agree with $\partial J / \partial \kappa_\rho$ from $E_O$ in fewer regions (Figs. 10a,c and Figs. 10b,d) using $\kappa_{\rho,tides}$ instead of $\kappa_{\rho,Argo}$ and $\kappa_{\rho,CTD}$. The regions with agreement in signs of sensitivities using $\kappa_{\rho,tides}$ account for just over half of the ocean's volume where they can be compared globally (Fig. 11; Table 4); this is also true for the subtropics. The equatorial regions have the highest percent volume of agreement in signs of sensitivities using $\kappa_{\rho,tides}$ over all depths, but the North Atlantic also has fairly high agreement (Fig. 11a). There is high agreement in the regions of the Arctic Ocean that are north of the Greenland and Barents Seas too. Compared with shallower depths, regions below 3000 meters depth tend to be derived from Antarctic Bottom Water (*Marshall and Speer*, 2012 - see their Figure 1) and therefore have different oxygen concentration characteristics such as weaker vertical gradients, have differences between $\kappa_{\rho,ECCO}$ and observationally-derived $\kappa_\rho$ that are more commonly statistically indistinguishable, and have less overwhelmingly positive adjoint sensitivities from $E_\kappa$ using $\kappa_{\rho,tides}$ (Fig. 10c). As a result, all of the depths with the highest levels of agreement in signs of sensitivities using $\kappa_{\rho,tides}$ are between the mixed layer depth and 3000 meters depth (Fig. 11b). Most differences in the spatial distribution of agreements between the signs of sensitivities across different observationally-derived products (Figs. 9 and 11) are at least partially due to their spatial coverage–Argo versus global–and the $< 100\%$ overlap in processes accounted for by the various $\kappa_\rho$ products. Thus, the level of agreement in signs of sensitivities from $E_\kappa$ and $E_O$ is high over many regions, and is qualitatively consistent in its spatial distribution across the different observationally-derived products for $\kappa_\rho$.

We need to address whether any of the agreement in signs of sensitivities is random–as their correlation is due to the large uncertainties in observationally-derived $\kappa_\rho$–or underpinned by physical reasons. We first focus on the locations with statistically indistinguishable errors in $\kappa_{\rho,ECCO}$. These regions and those where there can be significant differences between oxygen concentrations in ECCO and the World Ocean Atlas (2013) product correspond to the white regions in Fig. 9 that are red or blue in Fig. 8–likewise for Fig. 11 versus Fig. 10. The vast majority of the locations where disagreements occur in sensitivity signs are in places with statistically indistinguishable differences between $\kappa_{\rho,ECCO}$ and observationally-derived $\kappa_\rho$. The regions with statistically indistinguishable differences in $\kappa_\rho$ account for 56.2% (24.1%) of the volume of the ocean where the adjoint sensitivities from $E_O$ and $E_\kappa$ can be compared using $\kappa_{\rho,Argo}$ and $\kappa_{\rho,CTD}$ ($\kappa_{\rho,tides}$). Thus, we exclude a large portion of the ocean from the remainder of our analysis because we cannot determine whether agreements in signs of sensitivities are by random chance in these regions.

We next inspect the sensitivity sign patterns in regions with statistically significant $\kappa_\rho$ misfits. The regions where the signs of $\partial J / \partial \kappa_\rho$ agree from the two experiments and have large differences between $\kappa_{\rho,ECCO}$ and the combined $\kappa_{\rho,Argo}$ and $\kappa_{\rho,CTD}$ product tend to have relatively small oxygen concentration misfits (Fig. 3b). This is also true when using the $\kappa_{\rho,tides}$ product. For example, when only regions with less than one standard deviations above the average oxygen concentration misfits are selected, the signs of the adjoint sensitivities agree between $E_O$ and $E_\kappa$ over 60.8% (50.5%) of the volume with sufficient data using $\kappa_{\rho,Argo}$ and $\kappa_{\rho,CTD}$ ($\kappa_{\rho,tides}$). However, the larger the oxygen concentration misfits, the more often the signs of the sensitivities agree. When only regions with more than three standard deviations above the average oxygen concentration misfits are selected, the signs of the sensitivities always agree. Thus, the regions with the largest disagreements in oxygen concentrations can always decrease their oxygen misfits by changing $\kappa_{\rho,ECCO}$ with a sign consistent with decreasing its disagreement with observationally-derived $\kappa_\rho$, wherever differences in $\kappa_\rho$ are detectable.

Where there are statistically significant differences in $\kappa_\rho$, we still need to determine whether there is a physical basis for the agreements in signs of sensitivities. We next show results that are consistent with our hypothesis that the tropical OMZs and other (sub)tropical regions are where oxygen concentrations can inform $\kappa_\rho$ in the model such that there is better agreement with observationally-derived $\kappa_\rho$ products. The regions with the highest percent volume agreement in sensitivity signs, regardless of which observationally-derived $\kappa_\rho$ product is used, include tropical OMZs and other (sub)tropical regions below several hundred meters depth (Figs. 9b and 11b). Differences between the signs of the sensitivities tend to be more common in locations where $\kappa_\rho$ is not expected to dominate the variability in oxygen. These regions include, for example, the open subpolar North Atlantic Ocean (e.g., the Labrador Sea in Figs. 9a and 11a), where *Atamanchuk et al.* (2020) present observational evidence that air-sea fluxes mediated by bubble injection–not represented by ECCO–dominate the variability in oxygen down to 1000 meters depth. While there can be a high percent volume agreement in sensitivity signs in the equatorial Pacific and Atlantic Oceans, these are also regions where *Palter and Trossman* (2018) and *Brandt et al.* (2021) point out that ocean circulation changes significantly influence long-term changes in oxygen. This suggests that changes in both ocean circulation and $\kappa_\rho$ could be important in explaining oxygen concentration variations in the tropics. When the tropical and subpolar regions (outside of the $20^o - 50^o$N/S bands) are excluded, the percent volume of the ocean where the signs of the adjoint sensitivities agree between $E_\kappa$ and $E_O$ increases, regardless of which observationally-derived $\kappa_\rho$ product we use. Given that there are known physical processes not dominated by $\kappa_\rho$ causing variations in oxygen concentrations in regions outside of the tropical OMZs and other (sub)tropical regions, our interpretation of the patterns shown in Figs. 8-11 is that $\kappa_\rho$ controls much of the variability in oxygen concentrations in large portions of the tropical OMZs and other (sub)tropical regions. This is one indication that dissolved oxygen concentrations could provide information about $\kappa_\rho$, at least for some regions of the ocean.

We further address whether the potential information dissolved oxygen concentrations provide about $\kappa_\rho$ is due to the information oxygen contains about stratification. To determine whether oxygen provides information about stratification–and through stratification, about $\kappa_\rho$–we use the adjoint sensitivity results obtained from experiment $E_\epsilon$ with observationally-derived dissipation rates, $\epsilon = N^2 \kappa_\rho / 0.2$ (e.g., Figs. 1b,d,f) instead of $\kappa_\rho$ (e.g., Figs. 6b,d,f), in the misfit function via Eq (2) and multiply the adjoint sensitivity of $E_O$ by $0.2/N^2$ so that their sensitivities are each taken with respect to $\epsilon$ (parentheses in Table 4). We find approximately equal agreement between the signs of the adjoint sensitivities from $E_O$ (scaled by $0.2/N^2$) and $E_\epsilon$ as

we do between those from $E_O$ and $E_\kappa$ in every region, regardless of which observationally-derived product we use. Because $\epsilon$ is related to $\kappa_\rho$ through the stratification, we suggest that the information oxygen concentrations provide about $\kappa_\rho$ is likely independent of the stratification field.

Lastly, given that the general agreement in signs of sensitivities between $E_\kappa$ and $E_O$ are likely underpinned by physical reasons unrelated to stratification, we pursue whether there is a statistically significant relationship between the adjoint sensitivities from $E_\kappa$ and $E_O$. We again focus on the regions where the difference between $\kappa_{\rho,ECCO}$ and observational $\kappa_\rho$ products (from tides, Argo/CTD, and microstructure) is statistically significant (greater than a factor of three), but also filter out the adjoint sensitivities where the differences between oxygen concentrations from ECCO and those from the World Ocean Atlas (2013) are statistically significant. The simple correlations between the adjoint sensitivities from $E_\kappa$ and $E_O$ in the remaining regions tend to be small but positive (Fig. 12). In addition to taking simple correlations, we've performed Monte Carlo simulations to get a maximum possible correlation between each experiment's adjoint sensitivities. The maximum correlations from the Monte Carlo simulations are larger than the simple correlations. This is particularly the case where the adjoint sensitivities are both negative (Figs. 12a,c,e), but also true where the adjoint sensitivities are both positive (Figs. 12b,d,f). If we only consider comparing locations where we have both observationally-derived $\kappa_\rho$ data and oxygen data, our results are qualitatively the same and the correlations increase to as much as $0.47$ in the case of the $\kappa_{\rho,tides}$ product using a Monte Carlo approach. If we further only consider regions where the vertical gradients in stratification are less than their global mean and where the vertical gradients in oxygen concentrations are greater than their global mean, the correlations are approximately the same, indicating that the information oxygen provides about $\kappa_\rho$ is not conditional on the stratification. This suggests that $\kappa_{\rho,ECCO}$ may be constrained by the information provided by oxygen concentrations. That is, oxygen concentrations inform adjoint sensitivities that typically direct $\kappa_{\rho,ECCO}$ to improve relative to observationally-derived $\kappa_\rho$. However, inclusion of accurately known oxygen concentrations in the model's misfits is not a perfect substitute for the inclusion of accurately known $\kappa_\rho$ itself in the model's misfits.

# 4 Discussion and Concluding Remarks

## 4.1 Discussion

This study evaluated the potential to improve the diapycnal diffusivities ($\kappa_\rho$) in the ECCOv4 ocean parameter and state estimation framework. We assessed the fidelity of $\kappa_{\rho,ECCO}$ by first comparing the average vertical profiles of $\kappa_{\rho,ECCO}$ with those inferred from microstructure. The comparison was not favorable. In regions where we compare with observational products, $\kappa_{\rho,ECCO}$ is approximately $\kappa_{\rho,bg}$ in ECCO, which is inverted for through constraints of vertical profiles of temperature and salinity–e.g., from Argo profiles. Model choices–e.g., the initial guess of background $\kappa_\rho = 10^{-5}$ m$^2$ s$^{-1}$ everywhere–can lead to errors in $\kappa_{\rho,ECCO}$ even in the presence of globally complete hydrographic observations (see Section 4.2), but we investigated whether $\kappa_{\rho,ECCO}$ can benefit from new information.

We then investigated which additional observations can be used as new constraints to improve the fidelity of the inverted $\kappa_{\rho,ECCO}$. The products we used were observationally-derived $\kappa_\rho$ based on Argo and ship-based CTD hydrographic data,

observationally-derived $\kappa_\rho$ based on climatological and seafloor data, and climatological oxygen concentrations. To justify the use of the observationally-derived $\kappa_\rho$ products, we also evaluated them by comparing them with the microstructure-inferred product. $\kappa_{\rho,Argo}$ and $\kappa_{\rho,CTD}$ have better agreement with the microstructure-inferred data than $\kappa_{\rho,ECCO}$ does.

We inspected the misfit of the model parameter $\kappa_{\rho,ECCO}$ with respect to $\kappa_{\rho,Argo}$ and $\kappa_{\rho,CTD}$ as well as to $\kappa_{\rho,tides}$ and motivated use of dissolved oxygen concentration data as a potential constraint in ECCO. One drawback of the observationally-derived data products for $\kappa_\rho$ is that they have large uncertainties–here, approximated by a factor of three. Observed oxygen concentrations, on the other hand, have a relatively small uncertainties. More importantly, we showed that vertical oxygen gradients have similar geographical patterns to energy dissipation rates. We therefore performed an additional adjoint sensitivity experiment with oxygen concentration data as the only data in the misfit function. Adjoint sensitivities results were compared between the experiment with measured oxygen in the misfit function and observationally-derived $\kappa_\rho$ in the misfit function. Regions where the sensitivities agree in signs between the two experiments are locations where adjustments in $\kappa_\rho$, as informed by these data, can potentially help improve $\kappa_{\rho,ECCO}$. These regions include well over half of the volume of comparable seawater in the (sub)tropical regions–including tropical OMZs. These spatial patterns are consistent with where we expected $\kappa_\rho$ to explain much of the variability in oxygen concentrations. Correlations between adjoint sensitivities from each experiment are positive where differences between the oxygen concentrations in the model and observations are relatively small. These findings suggest that dissolved oxygen concentrations could be used to more accurately estimate $\kappa_\rho$ through $\kappa_{\rho,bg}$ in a newly optimized ECCO solution. However, given the magnitudes of the correlations between the adjoint sensitivities, inclusion of observationally-derived $\kappa_\rho$ in the model's misfits could (additionally) be necessary, especially if their uncertainties are reduced.

### 4.2   Caveats and future directions

Many factors–including a significant absence of independent observations for assessment, a combination of measurement and structural errors, numerical diffusion in our simulations, and unconstrained parameters in the biogeochemical modules– have stymied progress in state estimation of ocean subgrid-scale transport and mixing parameters. First, the observational measurement errors used here are only approximate. We assumed uncertainties equal to a factor of three of the observationally-derived $\kappa_\rho$ and 2% of the oxygen concentrations. These do not account for interpolation/averaging errors that entered the data prior to our calculations, but are conservative estimates nonetheless. The observational uncertainties affect the weights given in the misfits that enter the adjoint sensitivity calculations and our Monte Carlo simulations of the correlations between the adjoint sensitivities account for the possibility that these weights are misspecified. Second, only one ocean subgrid-scale transport or mixing parameter–namely, $\kappa_\rho$–has been compared with independent observational data–microstructure. This is the primary reason why we focused on $\kappa_\rho$ in our study. Third, the ECCO-estimated $\kappa_{\rho,bg}$ accounts for other model errors–e.g., structural ones suggested by *Polzin et al.* (2014)–which explains some of the model biases relative to microstructure observations. For instance, the ECCO-estimated $\kappa_{\rho,bg}$ should be time-dependent as well as spatially-varying, but it is only spatially-varying. In the presence of other estimated model parameters and initial conditions, some parameters could be compensating for errors in $\kappa_{\rho,bg}$. The ECCO-estimated $\kappa_{\rho,bg}$ can also be sensitive to the *a-priori* estimate of $\kappa_{\rho,bg}$ and we showed how one particular initial guess–$10^{-5}$ m$^2$ s$^{-1}$ everywhere–can evolve from the first optimization iteration to the final one. Additionally, there is numerical

diffusion in the model, which could confound some physical inferences about the model–e.g., regarding how sensitive the model's state is to $\kappa_{\rho,bg}$ relative to along-isopycnal diffusion. Numerical errors could remain and result in the primary source of error in the ocean state estimate even if additional constraints are placed on $\kappa_{\rho,bg}$ in ECCO. Retaining a parameter which absorbs structural model errors that are not expressed in the model's functional form may be necessary to improving the ECCO state estimate itself, but minimizing numerical errors would be beneficial to improving the ECCO-estimated $\kappa_{\rho,bg}$. Lastly, there are several unconstrained parameters in biogeochemical modules used to calculate biogeochemical tracers (*Verdy and Mazloff*, 2017), so some of the disagreements in signs of the adjoint sensitivities found here could be associated with other inaccurate parameters.

These challenges can continue to be overcome by allowing models and observations to inform each other. First, the observationally-derived $\kappa_{\rho}$ from the finescale parameterization could be further scrutinized using ship-based CTD data taken concurrently with microstructure velocity shear data. A preliminary analysis suggests that the percent difference between the full depth-averaged CTD-derived $\kappa_{\rho}$ from the finescale parameterization and the microstructure-inferred $\kappa_{\rho}$ at the same locations is 1.68%, which is indistinguishable from zero, but the quality of the the CTD data taken concomitantly with microstructure has not been fully assessed. Second, we can potentially overcome several computational challenges to improve $\kappa_{\rho,bg}$. One example of this is to account for the time-dependence of $\kappa_{\rho,bg}$ in a future ocean state estimate, but the under-determined nature of the parameter estimation procedure makes this difficult. These efforts would also benefit from minimizing spurious mixing due to numerical diffusion (e.g., *Holmes et al.* (2021)) through choosing a different advection scheme, but this would add computational expense. If the goal is to improve the other control variables and the state estimate itself–instead of estimating $\kappa_{\rho,bg}$ with ECCO–then we could potentially reduce the influence of numerical diffusion and other confounding factors in the estimation of $\kappa_{\rho,bg}$ by no longer treating $\kappa_{\rho,bg}$ as a control variable. Using an observationally-derived $\kappa_{\rho}$, such as the *de Lavergne et al.* (2020) product, would make the estimation of other control variables less under-determined, but this would not resolve the problem with a lack of time-dependence of $\kappa_{\rho,bg}$ nor would it assuage potential problems with model drift. One potential solution to this is to use the *de Lavergne et al.* (2020) product for $\kappa_{\rho,bg}$ and use the Prandtl number as a control variable to help determine the vertical viscosities. This would help with neither the estimation of under-determined parameters nor time-dependence of $\kappa_{\rho,bg}$ problems, but could help with numerical issues. Third, we could potentially circumvent the issue of unconstrained parameters in the biogeochemical modules. One potential way to do this is by including preformed oxygen–i.e., oxygen without any biological influence, making it a passive tracer–in the model's misfits instead of oxygen concentrations. Observationally-derived transit-time distributions with a maximum entropy-based method from previous studies (e.g., *Khatiwala et al.* (2009); *Zanna et al.* (2019)) or from a tracer-informed ocean state estimate (*DeVries and Holzer*, 2019) can help derive preformed oxygen from oxygen concentration observations. Fourth, we could optimize the information from existing oxygen observations with the purpose of constraining $\kappa_{\rho}$. One way to do this is to run observing system experiments. A complementary approach that uses the effective proxy potential framework of *Loose and Heimbach* (2021) could also help indicate whether measurements of oxygen concentrations in particular locations are redundant or more informative of $\kappa_{\rho}$ than in other locations. We did not pursue this in the present study because our adjoint runs use a global misfit. If we perform an ensemble of adjoint sensitivity runs with a single observation in each run, then we could calculate the effective proxy potential at each of these observation

locations. Lastly, the (imperfectly-known) initial conditions of biogeochemical tracers will also need to be included in the input control vector during optimization of the ocean state estimate. If biogeochemical tracers are included in the misfit calculation in an optimization run, their impact on variables such as $\kappa_\rho$ would depend upon how they are weighted relative to the physical variables–e.g., temperature, salinity, and pressure. A more complete representation and understanding of $\kappa_\rho$ is possible through these analyses and methods.

# Appendix A: Model with a sequential data assimilation framework

## A1  GEOS-5 and the GMAO S2S Ocean Analysis

To demonstrate a problem with $\kappa_\rho$ in a sequential data assimilation framework, we present example output from a reanalysis product and output from an identical ocean model hindcast without any data assimilation. GEOS-5 includes a global, finite volume atmospheric general circulation model that is used for numerical weather prediction, seasonal-to-decadal forecasts, and as the background field for atmospheric reanalyses (*Molod et al.*, 2015). The ocean is represented by the GFDL Modular Ocean Model (*Griffies et al.*, 2015), version 5 (MOM5) and the Los Alamos Community Ice CodE sea ice model (*Hunke et al.*, 2013), version 4.1 (CICE4.1). We use a configuration of the GEOS-5 modeling system with a $1^o$ ($0.5^o$ at equator) resolution on a tripolar (*Murray*, 1996) staggered Arakawa B-grid (*Mesinger and Arakawa*, 1976) and $50$ geopotential levels for MOM5, $2^o$ resolution and $24$ pressure levels for the atmospheric model, and $1^o$ resolution and 3 layers for CICE4.1. Historical aerosols (sulfate, dust, and sea salt) and biomass burning emissions (black and organic carbon) updated from the Goddard Chemistry Aerosol Radiation and Transport (GOCART) model (*Chin et al.*, 2002) are used over the time period 1992 through 2016. Initial conditions are based on a long spin-up that used MOM4 coupled to one version of the GEOS-5 atmosphere model (*Molod et al.*, 2012) and hundreds of additional years of spin-up that used MOM4 coupled to a slightly different version of the GEOS-5 atmosphere model. The differences between the two versions of the GEOS-5 atmospheric model used in the two phases of spin-up include developments in cloud microphysics and atmospheric chemistry.

$\kappa_\rho$, Redi coefficients, and Gent-McWilliams coefficients are determined in MOM5 as follows. $\kappa_\rho$ in MOM5–$\kappa_{\rho,GEOS5}$ hereafter–is represented by the K-Profile Parameterization (KPP; *Large et al.* (1994)) and a parameterization for mixing due to internal tides (*Simmons et al.*, 2004). Shear-driven mixing, gravitational instabilities that can cause vertical convection, and double-diffusive processes, which can cause the temperature diffusivity to be different from the salinity diffusivity, are accounted for in the interior (*Large et al.*, 1994). The resulting $\kappa_{\rho,GEOS5}$ field spatio-temporally varies. However, this combination of parameterizations does not make use of an explicit energy budget that accounts for conversion between kinetic and potential energy when determining $\kappa_{\rho,GEOS5}$. The Redi coefficients (*Redi*, 1982) and Gent-McWilliams coefficients of the (*Gent and McWilliams*, 1990) parameterization for mesoscale eddies are, by default, prescribed to be $600$ m$^2$ s$^{-1}$ everywhere, except for some variation in western boundary current regions for the Gent-McWilliams coefficients. The Redi coefficients and Gent-McWilliams coefficients are, thus, constant in time and in most locations. A mixed layer instability scheme for the submesoscale transport by *Fox-Kemper et al.* (2011) is used.

We use a reanalysis product, which uses the same underlying modeling system as the GEOS-5 coupled earth system model, called the Global Modeling and Assimilation Office sub-seasonal to seasonal (GMAO S2S) Ocean Analysis. The output of the GMAO S2S Ocean Analysis highlights how $\kappa_\rho$ can behave due to the disruption of dynamical balance that can be the result of the use of a sequential data assimilation system (*Stammer et al.*, 2016; *Pilo et al.*, 2018). The GMAO S2S Ocean Analysis only assimilates hydrographic information to constrain $\kappa_\rho$ and relies on the same parameterizations as GEOS-5's ocean component to calculate $\kappa_\rho$.

The NASA GMAO has recently updated their GEOS-5 sub-seasonal to seasonal forecast system (S2S-v2.1; https://gmao.gsfc.nasa.gov/cgi-bin/products/climateforecasts/geos5/S2S_2/index.cgi). This new system is the current contribution of the GMAO to the North American Multi-Model project (http://www.cpc.ncep.noaa.gov/products/NMME/about.html) and NOAA's experimental sub-seasonal ensemble project (http://cola.gmu.edu/kpegion/subx/index.html). A configuration of the modeling system is used that is nominally $0.5^o$ resolution on a tripolar (*Murray*, 1996) staggered Arakawa B-grid (*Mesinger and Arakawa*, 1976) and $40$ geopotential levels for MOM5, and $0.5^o$ resolution and $5$ layers for CICE4.1 with atmospheric forcing from MERRA-2 (Modern-Era Retrospective analysis for Research and Applications, Version 2) reanalysis (*Gelaro et al.*, 2017). The GMAO S2S Ocean Analysis (*Molod et al.*, 2020) is a reanalysis product that uses a system similar to the Local Ensemble Transform Kalman Filter (LETKF) data assimilation procedure described by (*Penny et al.*, 2013), but where the background error is calculated offline using ensemble members of freely coupled simulations. The background error does not explicitly account for uncertainties in any of the ocean subgrid-scale transport or mixing parameters, as it is only a function of the observed and background temperatures and salinities. The temperature and salinity would change and so would the calculated covariances if the mixing parameterizations were changed, but each of the 21 background free-running simulations have the same mixing parameterization, as they only differ in their initialization.

The following datasets were used by the GMAO S2S data assimilation modeling system. A relaxation procedure, or update, is applied towards the MERRA-2 sea surface temperatures and sea ice fraction from the NASA TEAM-2 product (*Markus et al.*, 2009) at a 5-day assimilation cycle. No ocean subgrid-scale transport or mixing parameter data are assimilated. Assimilated in situ observational data that provide temperatures and salinities come from TAO, PIRATA, RAMA, XBT, CTD, and Argo instruments. Satellite altimetry data that provide sea level anomalies come from TOPEX, ERS-1+2, Geosat FO, Jason-1, Jason-2, Jason-3, Envisat, Cryosat-2, Saral, HY-2A, and Sentinel 3A. The absolute dynamic topography is calculated as the sum of the sea level anomaly and the mean dynamic topography, which is estimated using GOCE and GRACE data, all available altimetry, and in situ data. Absolute dynamic topography data are assimilated into the model system using the same method as for the in situ data, except these data are thinned along-track and a Gaussian weighted mean using a decorrelation scale of 1000 km is calculated prior to assimilation. In addition, the global trend was removed from the absolute dynamic topography before assimilation and zero net input of water was applied. Precipitation is corrected using the Global Precipitation Climatology Project version 2.1 (GPCPv2.1, provided by the NASA/Goddard Space Flight Center's Laboratory for Atmospheres, which calculates the dataset as a contribution to the GEWEX Global Precipitation Climatology Project) and Climate Prediction Center (CPC) Merged Analysis of Precipitation (CMAP, provided by the NOAA/OAR/ESRL PSD, Boulder, Colorado, USA, from their website at http://www.esrl.noaa.gov/psd/), as described by *Reichle et al.* (2011) except for MERRA-2 instead of MERRA data. All other atmospheric forcing fields used in the construction of the reanalysis came from MERRA-2. The GMAO S2S modeling system is an update to the one described in *Borovikov et al.* (2017). As such, the model only ran for the period: May of 2012 to March of 2019.

## A2 Steric sea level budget framework

In order to examine whether the analysis increments can dynamically impact $\kappa_\rho$, we analyze a model's buoyancy budget, which is broken down into heat and salt budgets and used to calculate the steric sea level budget. The tracer tendency equation terms required for the heat and salt budgets were computed online and saved as the reanalysis was running. The tracer equations can be broken down into individual contributions (*Palter et al.*, 2014),

$$\rho\frac{d\Theta}{dt} + \rho A^\Theta = -\nabla \cdot \mathbf{J}^\Theta + \rho Q^\Theta \tag{A1}$$
$$\rho\frac{dS}{dt} + \rho A^S = -\nabla \cdot \mathbf{J}^S + \rho Q^S,$$

where $d/dt = \partial/\partial t + (\mathbf{v} + \mathbf{v}^*) \cdot \nabla$ is the material derivative, $\mathbf{v}$ is the resolved velocity field, $\mathbf{v}^*$ is the eddy-induced or quasi-Stokes velocity field that represents parameterized motions, $\Theta$ is the potential temperature, $S$ is the salinity, $\rho$ is the locally referenced potential density, $\mathbf{J}^\Theta$ and $\mathbf{J}^S$ are the parameterized along-isopycnal and diapycnal mixing fluxes associated with potential temperature and salinity, $Q^\Theta$ and $Q^S$ are the sums of sources and sinks of potential temperature and salinity, and $A^\Theta$ and $A^S$ are the analysis increments for potential temperature and salinity due to the assimilation of data by a sequential filter-based data assimilation ocean modeling system. The analysis increments in a sequential filter-based data assimilation system can obscure the physics, as $A^\Theta$ and $A^S$ effectively correspond to unphysical, time-varying, three-dimensional fluxes of heat and salt.

The heat and salt budget terms summarized by Equation (A1) are computed as follows. The resolved, mesoscale, and submesoscale transports are accounted for in the material derivatives $\Theta$ and $S$, the neutral and diapycnal diffusion of $\Theta$ and $S$ are accounted for by $\mathbf{J}^\Theta$ and $\mathbf{J}^S$, and the analysis increments of $\Theta$ and $S$ are accounted for by $A^\Theta$ and $A^S$. The neutral diffusion term includes cabbeling, thermobaricity, and a dianeutral contribution that mixes properties by providing for the exponential transition to horizontal diffusion in regions of steep isoneutral slopes according to *Treguier* (1992) and *Ferrari et al.* (2008, 2010) where the surface boundary layer is encountered and following *Gerdes et al.* (1990) next to solid walls. The diapycnal diffusion term is not added to the vertical component of the along-isopycnal diffusion term, but because of convention (e.g., *Palter et al.*, 2014) is nevertheless referred to as the vertical diffusion term hereafter. The vertical diffusion term also includes penetrating shortwave radiation flux. The sources and sinks of $\Theta$ and $S$ accounted for by $Q^\Theta$ and $Q^S$ include nonlocal convection (the transport where turbulent fluxes don't depend upon local gradients in $\Theta$ or $S$ because buoyant water gets entrained into the mixed layer when the surface buoyancy forcing drives convection above a stratified water column); surface buoyancy fluxes (latent, sensible, shortwave, longwave, and frazil heat fluxes); precipitation minus evaporation; runoff mixing (mixes properties associated with river outflows); downslope mixing (mixes properties downslope to represent the overflow dense waters from marginal seas); sigma-diffusion (mixing properties along terrain-following coordinates in regions with partial bottom cells); numerical smoothing of the free surface (intended to reduce B-grid checkerboard noise); numerical sponge (intended to absorb the Kelvin waves set off by the assimilation of some data); calving of land ice; and frazil ice formation. The runoff mixing, downslope mixing, and sigma-diffusion terms are considered sources or sinks here because they are associated with numerical schemes that aim to resolve problems created by coarse model resolution, the vertical coordinate

system used near boundary layers, and imperfect bathymetry. There is no geothermal heating included in the GMAO S2S

Ocean Analysis. The vertical diffusion term includes a subsurface shortwave heating contribution to a function of the $\kappa_\rho$ field, the mesoscale transport term assumes constant Gent-McWilliams coefficients, and the neutral diffusion term assumes constant Redi coefficients.

At each time step, the model evaluates a tendency term for every process that contributes to (A1) from their parameterized or dynamically calculated values, their units are converted to W m$^{-2}$ and kg m$^{-2}$ s$^{-1}$ for $\Theta$ and $S$, and their monthly averages

are saved to the output files used in this analysis. Implicit in these output tendency terms is that each term is weighted by the thicknesses of each layer as the model runs and writes the output to file. The heat and salt budget terms saved to file are used to calculate the steric sea level budget as follows. The steric sea level budget terms are computed by scaling the heat tendency terms by $\alpha/C_p$ and the salt tendency terms by $-1000\beta$, where $C_p$ (units in J kg$^{-1}$ K$^{-1}$) is the specific heat of seawater, $\alpha = -[1/\rho](\partial\rho/\partial T)$ (units in K$^{-1}$) is the thermal expansion coefficient, and $\beta = [1/\rho](\partial\rho/\partial S)$ (units in kg g$^{-1}$) is the haline

contraction coefficient. In order to get a longitude-latitude map of the terms that depend upon depth shown here, we integrate over depth by summing over the depth dimension. We analyze part of the steric sea level budget of the GMAO S2S Ocean Analysis to examine the relationships between different terms.

## Appendix B: Results for the sequential data assimilation framework

### B1 Assessments of $\kappa_\rho$ from models

First, we compare the average $\kappa_{\rho,micro}$ profile that is comprised of 24 campaigns worth of data (*Waterhouse et al.*, 2014) (see their Fig. 6; black curve in Fig. A1a) with the average model-calculated $\kappa_\rho$ profiles and $\kappa_{\rho,Argo}$. A geometric average is taken for each profile because a geometric average is more representative than an arithmetic average for a small sample size and when the data are not normally distributed (*Manikandan*, 2011), like the log-normal distribution of $\kappa_\rho$ (*Whalen*, 2021).

We compare microstructure (black curve in Fig. A1a) with GEOS-5 (red curve in Fig. A1a). $\kappa_{\rho,GEOS5}$, on average, is in close

agreement with microstructure over the upper 250-2000 meters. On average, the disagreement with microstructure and Argo is approximately the same as the disagreement between microstructure and GEOS-5. All three $\kappa_\rho$ are well within the uncertainty of the Argo product. The profiles are also within the temporal variability in $\kappa_{\rho,GEOS5}$ below the mixed layer depths (Fig. A1b; also see Fig. 9 in *Whalen et al.* (2015)). The temporal variability in $\kappa_\rho$ is only large near regions with active deep convection (e.g., between 40-50$^o$N in the North Atlantic, as shown in Fig. A1b). The blue and green diamonds in Fig. 1c of *Waterhouse*

*et al.* (2014) show that there are only a few microstructure profiles are within the 40-50$^o$N band in the North Atlantic. These are all near the east coast of North America, not in regions that experience deep convection so the temporal variability in microstructure is not expected to be large enough that the disagreements in $\kappa_\rho$ can be explain by temporal sampling/aliasing.

While the average $\kappa_{\rho,GEOS5}$ profile is fairly accurate, particularly below 500 meters depth (red curve in Fig. A1), $\kappa_{\rho,GMAO}$ is in much worse agreement with microstructure (green curves in Fig. A1). The large values of $\kappa_{\rho,GMAO}$ are not due to a few

isolated locations. $\kappa_{\rho,GMAO}$ is too large below about 250 meters depth (solid green curve in Fig. A1). The average profile of $\kappa_{\rho,GMAO}$ is generally constant or decreases with depth, as opposed to the average profiles of $\kappa_{\rho,GEOS5}$ and microstructure,

which generally increase with depth. Potential reasons for the large disagreements between $\kappa_{\rho,GMAO}$ and microstructure include dynamical adjustments due to the GMAO S2S Ocean Analysis' analysis increments, inconsistencies between the model's atmosphere and ocean due to the strong relaxation to sea surface temperatures, fixed zero net water input for global sea level, and numerics such as the techniques applied to damp the waves created from assimilating some observations.

## B2   Model- vs finescale parameterization-derived $\kappa_\rho$ comparisons

While comparisons with microstructure reveal general agreement with the average profile of $\kappa_{\rho,GEOS5}$–except near the surface and at deep depths–we also want to assess whether there are deficiencies in the average geographic distribution of $\kappa_{\rho,GEOS5}$ by comparing the output of GEOS-5 with the $\kappa_{\rho,Argo}$ product. Comparing the $\kappa_{\rho,GEOS5}$ field with the $\kappa_{\rho,Argo}$ product results in better agreement than the similar comparisons between $\kappa_{\rho,GMAO}$ and $\kappa_{\rho,Argo}$. For example, $\kappa_{\rho,GEOS5}$ only disagrees with $\kappa_{\rho,Argo}$ by more than a factor of 3 over 36.6% of grid points with available data (Fig. A2b), while the disagreement doubles in percentage (79.1%) for $\kappa_{\rho,GMAO}$ (Fig. A2a). The errors in $\kappa_{\rho,GEOS5}$ are smaller than $\kappa_{\rho,GMAO}$. Thus, when the objective of the GMAO S2S Ocean Analysis is to minimize the misfit between the model and observations of temperature, salinity, and some surface characteristics, $\kappa_\rho$ can be better represented without any observational constraints; i.e., the GMAO S2S Ocean Analysis improves temperature and salinity misfits for the wrong reasons.

The regions with the largest disagreement between $\kappa_{\rho,GEOS5}$ and $\kappa_{\rho,Argo}$ are along the equator, in the Southern Ocean, in the Labrador and Irminger Seas, and in the Gulf Stream and Kuroshio Extensions (Fig. A2b). Along the equator the values of $\kappa_{\rho,GEOS5}$ tend to be larger than the observational product, but the discrepancy changes sign slightly poleward in the near-equator tropics. Inadequate resolution and parameterization of diapycnal mixing can cause too little mixing to occur in these regions as well as in the Southern Ocean and along mid-ocean ridges (*MacKinnon et al.*, 2017). The values of $\kappa_{\rho,GEOS5}$ are smaller than the observations both in regions where deep convection is prevalent and in the vicinity of the Antarctic Circumpolar Current (ACC). In the Gulf Stream Extension region, the Malvinas Current region, part of the Kuroshio Extension region, and the Indian Ocean sector of the ACC above 500 meters depth, the values of $\kappa_{\rho,GEOS5}$ are too large. This is because $\kappa_{\rho,GEOS5}$ can be much increased inside the mixed layer depth, which can be deeper than 250 meters due to vertical convection. One possible source of these errors in the abyssal $\kappa_\rho$ is the improper treatment of remote internal tide-induced mixing, discussed in *Melet et al.* (2016), but several other processes can impact $\kappa_\rho$ in the upper water column. For example, the wind-driven near-inertial waves (*Alford et al.*, 2016) can be important near the surface in many locations, and internal tide breaking is important near the seafloor at low latitudes in the Northern Hemisphere (*Arbic et al.*, 2004; *Nycander*, 2005; *Melet et al.*, 2013; *MacKinnon et al.*, 2017) and beneath the ACC, where lee wave breaking is important (*Nikurashin and Ferrari*, 2011; *Scott et al.*, 2011; *Naveira Garabato et al.*, 2013; *Melet et al.*, 2014; *Wright et al.*, 2014; *Trossman et al.*, 2013, 2016; *Yang et al.*, 2018). *MacKinnon et al.* (2017) discusses other candidates for more accurate representation of $\kappa_\rho$. Identifying the sources of errors in $\kappa_{\rho,GEOS5}$, particularly in the abyss, is beyond the scope of the present study. We emphasize the much greater errors in $\kappa_{\rho,GMAO}$ and next examine whether the analysis increments could be one source of these larger errors (either directly or by way of altering the velocity field).

## B3 Relationships between steric sea level budget terms

There are distortions in temperature and salinity fields from applying analysis increments, violating conservation principles and potentially causing the model to undergo baroclinic adjustment (*Stammer et al.*, 2016). Thus, we examine whether the velocity field itself changes because of the analysis increments. To do this, we show the relationship between the analysis increments and resolved advection terms in the steric sea level budget for the GMAO S2S Ocean Analysis in Fig. A3a. The Pearson correlation coefficient between the analysis increments and resolved advection terms in the steric sea level budget is about $-0.3$. The magnitudes of the analysis increments are determined by the temperature, salinity, and sea surface height fields, and the analysis increments and the resolved advection term in the GMAO S2S Ocean Analysis are comparable in size for both heat and salt tendencies–the largest terms in each budget in their zonal averages at most latitudes. However, previous studies have shown that analysis increments induce changes in the velocity field via dynamic adjustment (*Stammer et al.*, 2016; *Pilo et al.*, 2018). The correlation between the analysis increments and resolved advection terms shown in Fig. A3a are consistent with the findings of these previous studies. The analysis increments, by a similar argument, could induce physically inconsistent air-sea exchanges through changing the temperature and salinity fields in the top model layer. We next show that these factors at least partially cause errors in $\kappa_{\rho,GMAO}$. The Pearson correlation coefficients between the diapycnal diffusion terms and the analysis increment terms in the heat and salt budgets over all locations are about $0.7$ (Fig. A3b), suggesting that the analysis increments are associated with errors in $\kappa_{\rho,GMAO}$. Problems with the physical consistency of air-sea exchanges–due to relaxation of sea surface temperatures and requiring net zero water input–could also contribute to the errors in $\kappa_{\rho,GMAO}$. However, it is possible that instead of the air-sea exchanges impacting the diapycnal diffusivities directly, the analysis increments affect both the air-sea exchanges and diapycnal diffusivities, as the changes in temperature and salinity at depth also change the mixed layer depths, which perturb the diapycnal diffusivity profiles and therefore their contribution to steric sea level through the altered thermal expansion/haline contraction coefficients. The correlation between the surface flux and diapycnal diffusion terms in the heat and salt tendency budgets are fairly well correlated–Pearson correlation coefficient of about $-0.4$ (Fig. A3c), suggesting that there is an association between the surface flux errors and errors in $\kappa_{\rho,GMAO}$. Given these correlations and the way analysis increments and physical inconsistencies of air-sea exchanges are implemented in the GMAO S2S Ocean Analysis, errors in $\kappa_{\rho,GMAO}$ must be caused by analysis increments (and possibly adjustments of air-sea exchanges) rather than the other way around.

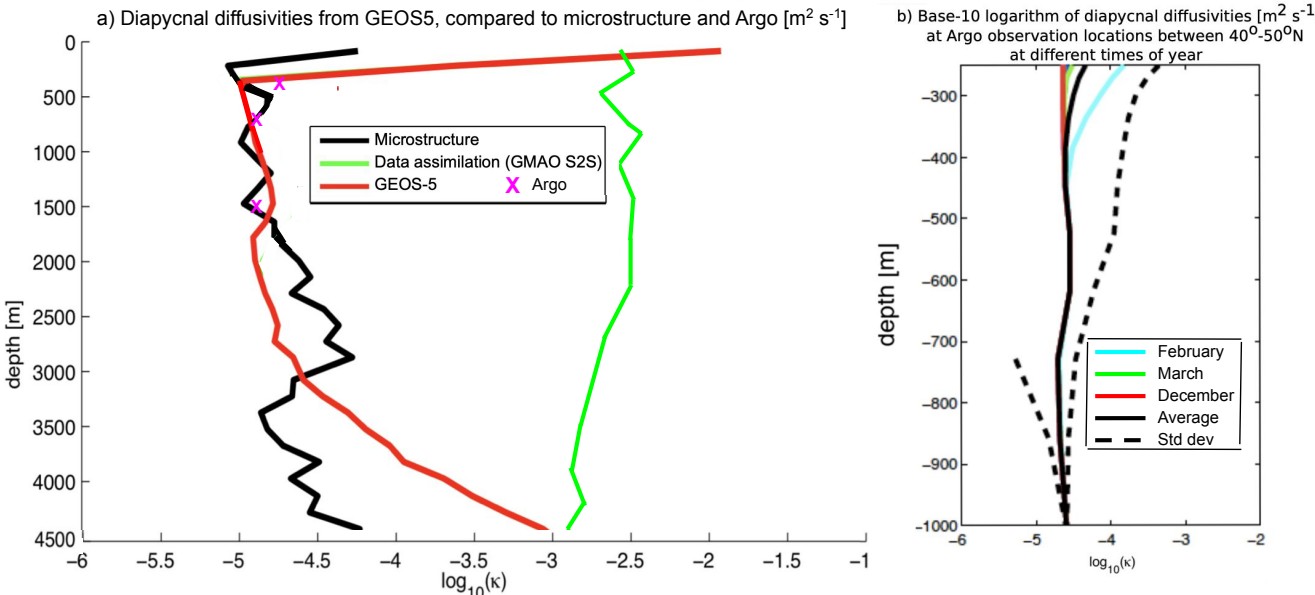

**Figure A1.** $\kappa_\rho$ profiles (panel a; units in m$^2$ s$^{-1}$) averaged over the full-depth microstructure locations from the observations (black curve) presented in *Waterhouse et al.* (2014) (see their Fig. 6), the *Whalen et al.* (2015) Argo-derived product for three depth bins (magenta X's), the temporally-averaged output of a free-running coupled earth system model simulation (GEOS-5 - red curve), and the temporally-averaged output of an equivalent run with data assimilation (GMAO S2S - green curve). Also shown are (panel b) $\kappa_\rho$ profiles from the free-running GEOS-5 simulation averaged over $40 - 50^o$N in the North Atlantic Ocean and averaged over all January months (lighter colors), ..., and all December months (darker colors). The base-10 logarithms of the geometric averages are shown in each panel.

a) Base-10 logarithms of diapycnal diffusivity ratios (GMAO S2S to Argo) b) Base-10 logarithms of diapycnal diffusivity ratios (GEOS5 to Argo)

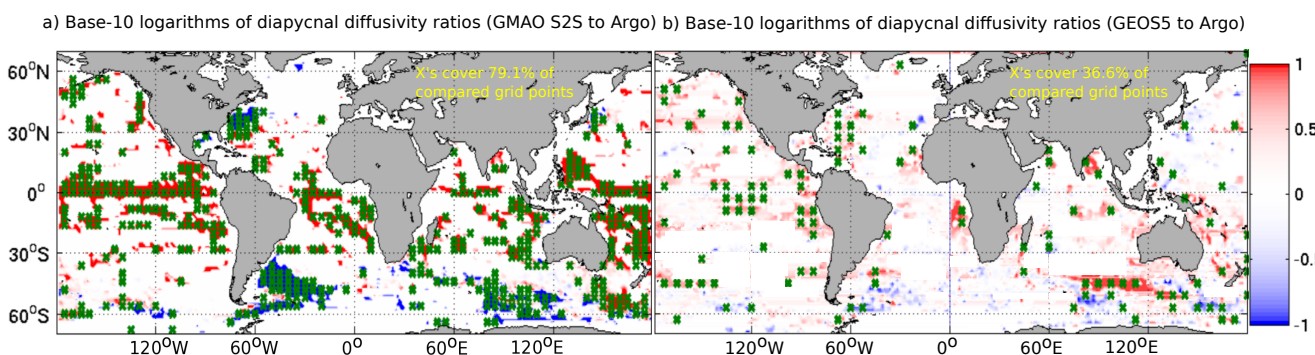

**Figure A2.** Shown are (a) the base-10 logarithms of the ratios of the time-averaged $\kappa_{\rho,GMAO}$ to $\kappa_{\rho,Argo}$, and (b) the base-10 logarithms of the ratios of the time-averaged $\kappa_{\rho,GEOS5}$ to $\kappa_{\rho,Argo}$. Each panels shows an average over 250-2000 meters depth. White areas in the ocean indicate insufficient Argo data to derive $\kappa_{\rho,Argo}$. The green X's indicate regions where the disagreement between $\kappa_{\rho,GMAO}$ or $\kappa_{\rho,GEOS5}$ and $\kappa_{\rho,Argo}$ is greater than a factor of 3.

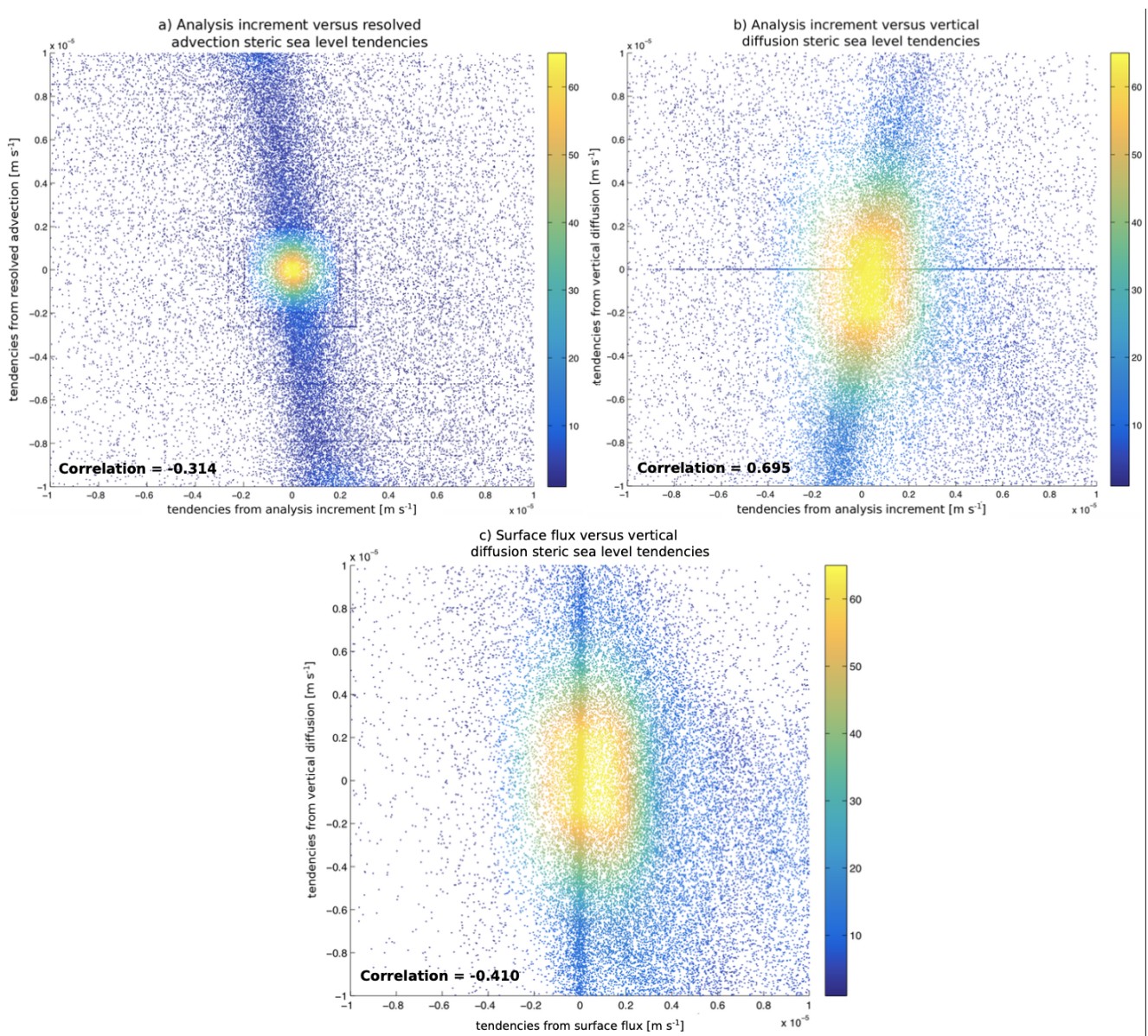

**Figure A3.** Scatterplots between several of the most locally dominant tendency terms in the steric sea level budget of the GMAO S2S Ocean Analysis, averaged over the entire run's time period (2012-2017): shown are (panel a) the analysis increment (abscissa) versus the resolved advection (ordinate) terms, (panel b) the analysis increment (abscissa) versus the vertical diffusion (ordinate) terms, and (panel c) the surface flux (abscissa) versus the vertical diffusion (ordinate) terms. The more yellow colors indicate a greater density of dots in the scatterplots. The more blue colors indicate a lower density of dots in the scatterplots. Also listed in each panel are the correlations between each of the comparisons.

*Data availability.* The data used in this are available at: https://zenodo.org/record/6494658#.YmhaZy-B28U . Also, the GMAO S2S Ocean Analysis output is available at: ftp://gmaoftp.gsfc.nasa.gov/pub/data/kovach/S2S_OceanAnalysis/ . The hydrography-derived diapycnal diffusivities from the finescale parameterization used in this study, courtesy of Eric Kunze, are available by logging in as a guest at: ftp://ftp.nwra.com/outgoing . The microstructure data used in this study are available at: https://microstructure.ucsd.edu/ .

*Author contributions.* David S. Trossman conceived of the idea, did the ECCO and GEOS-5 simulations, performed the analyses, and wrote up the manuscript. Caitlin Whalen provided the Argo-derived data and helped with their use and the writing. Thomas Haine helped with the interpretation of the results and the writing. Amy Waterhouse provided the microstructure data and helped with their use and the writing. An Nguyen helped with the interpretation of the ECCO results and the writing. Arash Bigdeli provided help with getting the adjoint to work with ECCO. Patrick Heimbach provided help with getting the forward MITgcm runs to get set up correctly on the TACC machines. Matthew 830   Mazloff helped with the interpretation of the results and the writing.

*Competing interests.* The authors declare no conflict of interests.

*Acknowledgements.* The authors thank the reviewers of this manuscript for their suggestions. David Trossman was supported by the Goddard Earth Sciences Technology And Research (GESTAR) cooperative agreement between the GMAO of the NASA Goddard Space Flight Center base and Johns Hopkins University as well as NASA SLCT grant 80NSSC17K0675 at the University of Texas-Austin. Thomas W. N. Haine 835   was supported by NOAA award NA15OAR4310172 and NSF award OCE-1338814. Amy Waterhouse was supported by NSF award OCE-0968721. Caitlin Whalen was supported by National Science Foundation Award OCE-1923558. Patrick Heimbach was supported by the ECCO project through a JPL/Caltech subcontract. An Nguyen and Arash Bigdeli were supported by NSF awards NSF-OPP-1603903 and NSF-OPP-1708289.

  The authors acknowledge the Texas Advanced Computing Center (TACC) at The University of Texas at Austin for providing HPC 840   resources for the ECCO simulations (URL: http://www.tacc.utexas.edu) and the NASA Center for Climate Simulation (NCCS) for the computer time spent on the GEOS-5 simulations that have contributed to the research results reported within this paper.

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

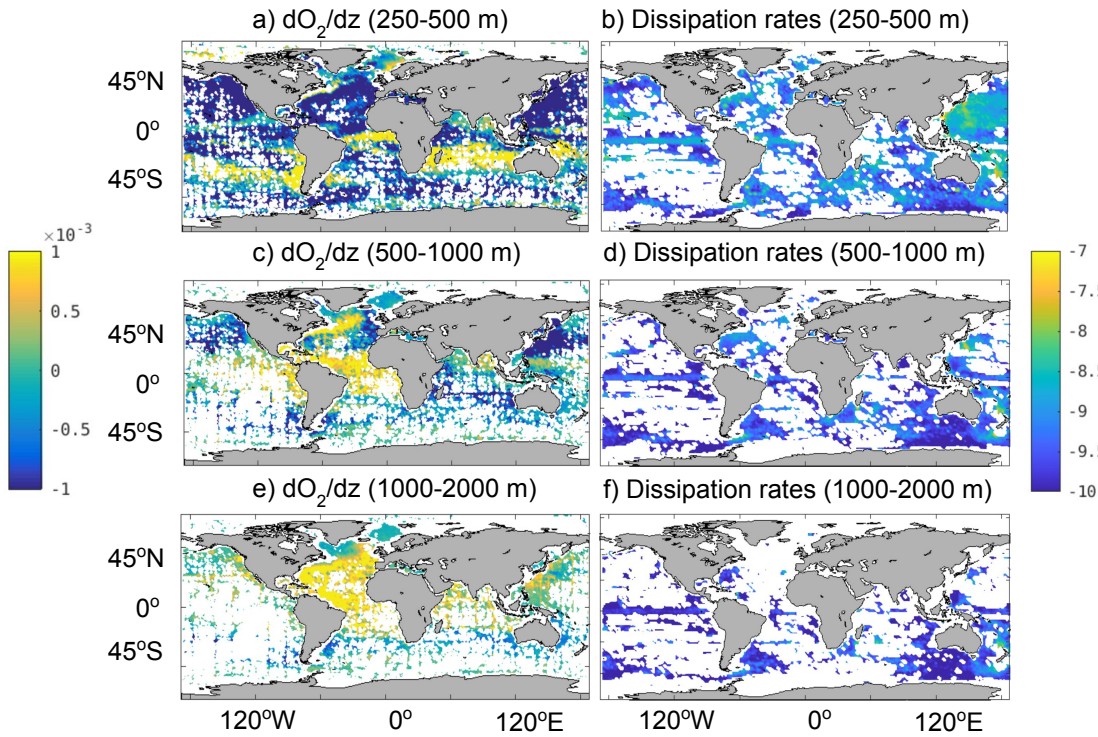

**Figure 1.** Shown are the vertical gradients of oxygen concentrations (units in ml/l/m) from the World Ocean Atlas (2013) (panels a,c,e) and the base-10 logarithms of the dissipation rates (units in $W\,kg^{-1}$) from *Whalen et al.* (2015) (panels b,d,f). Panels a-b show an average over 250-500 meters depth. Panels c-d show an average over 500-1000 meters depth. Panels e-f show an average over 1000-2000 meters depth. White areas in the ocean indicate insufficient data.

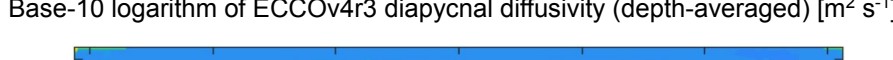

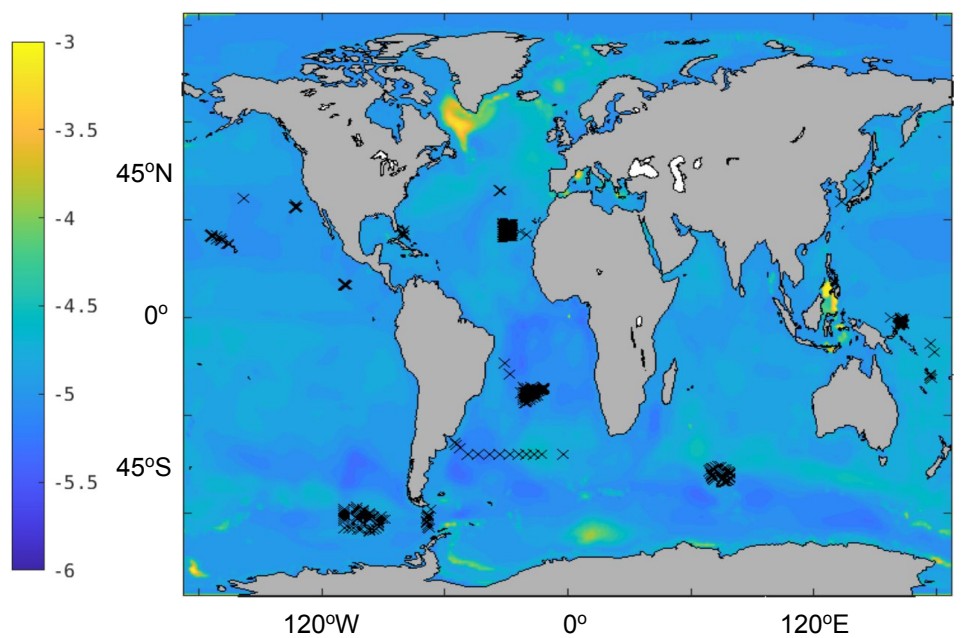

**Figure 2.** Shown is the base-10 logarithm of $\kappa_{\rho,ECCO}$ (units in m$^2$ s$^{-1}$), depth-averaged over all depths below the average mixed layer depth to exclude very large values within the mixed layer. Black X's indicate locations where there are microstructure measurements used in this study.

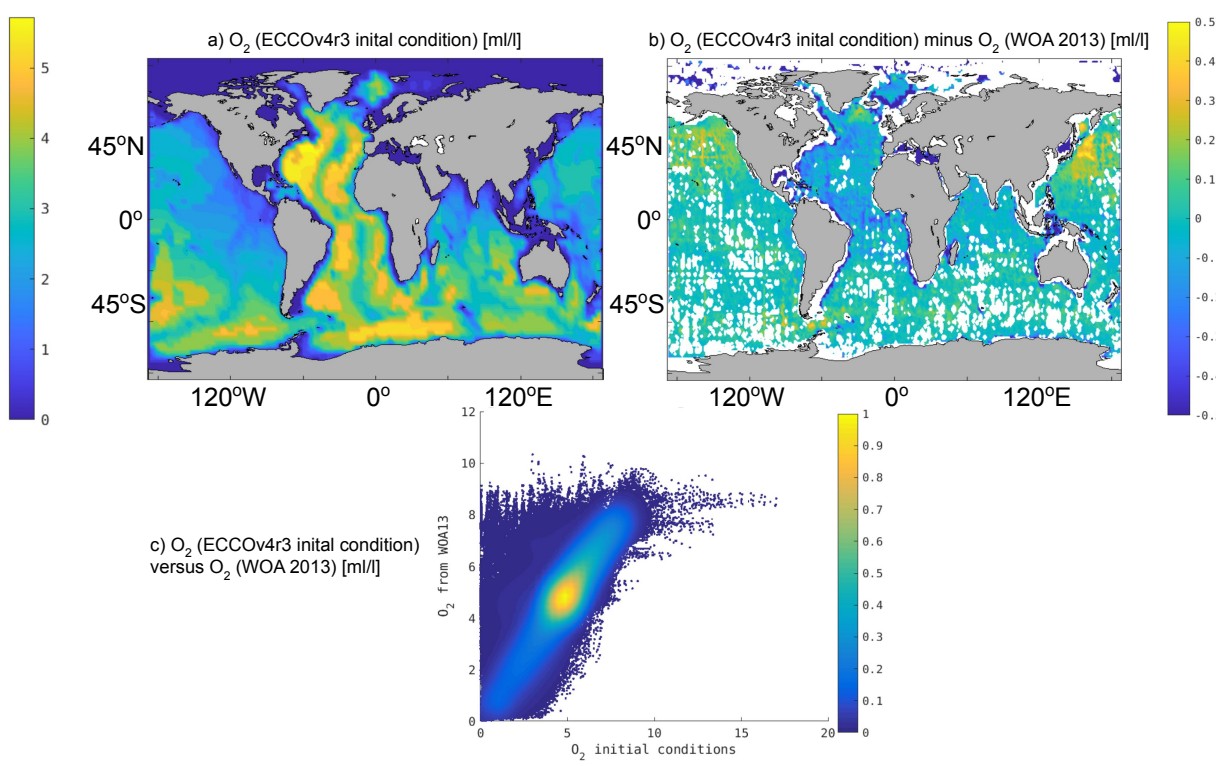

**Figure 3.** Shown are (a) the depth-averaged oxygen concentrations' initial conditions in ECCO (units in ml/l), (b) the depth-averaged oxygen concentrations' initial conditions in ECCO minus the depth-averaged observational climatologies from the World Ocean Atlas (2013) (units in ml/l) at locations where observations were sampled, and (c) the point-wise comparisons between the oxygen concentrations' initial conditions and the observational climatologies from the World Ocean Atlas (2013) (units in ml/l), which are used for the ECCO adjoint sensitivity experiments in the model's cost function. White areas in panel b in the ocean indicate insufficient data to calculate a depth-average.

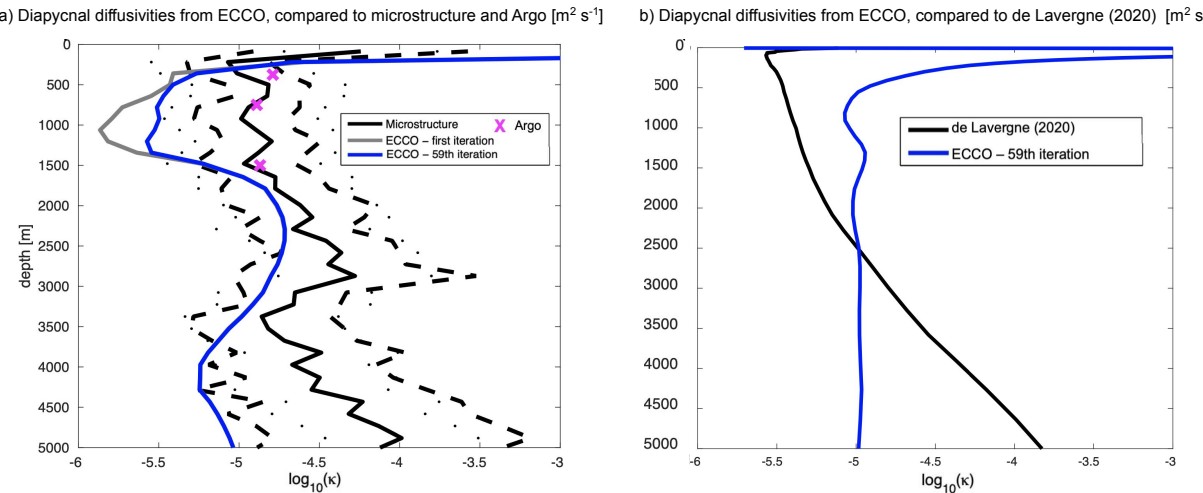

**Figure 4.** Panel a: shown are $\kappa_\rho$ profiles averaged over all microstructure observation locations (shown in Fig. 2) from the first iteration of the optimization (E-CTRL$_0$ - grey curve), and from the (final) fifty-ninth iteration of the optimization (E-CTRL - blue curve). Also shown is the average of $\kappa_\rho$ profiles from the full-depth microstructure observations (black curve) presented in *Waterhouse et al.* (2014) (see their Fig. 6; also see Fig. 5 of the present study) with one spatial standard deviation flanking the average (dashed black curves) and an approximate factor of three uncertainty flanking the average (dotted black curves), and the average of $\kappa_\rho$ (magenta X's) at each of the depth bins in the *Whalen et al.* (2015) product. At each location, the simulated profiles are extracted and the base-10 logarithms of the geometric averages of the observed and ECCO-estimated $\kappa_\rho$ (units in m$^2$ s$^{-1}$) are shown. Panel b: shown are $\kappa_\rho$ profiles averaged over the entire ocean from the fifty-ninth iteration of the optimization (E-CTRL - blue curve) and from the *de Lavergne et al.* (2020) tidal mixing product (black curve).

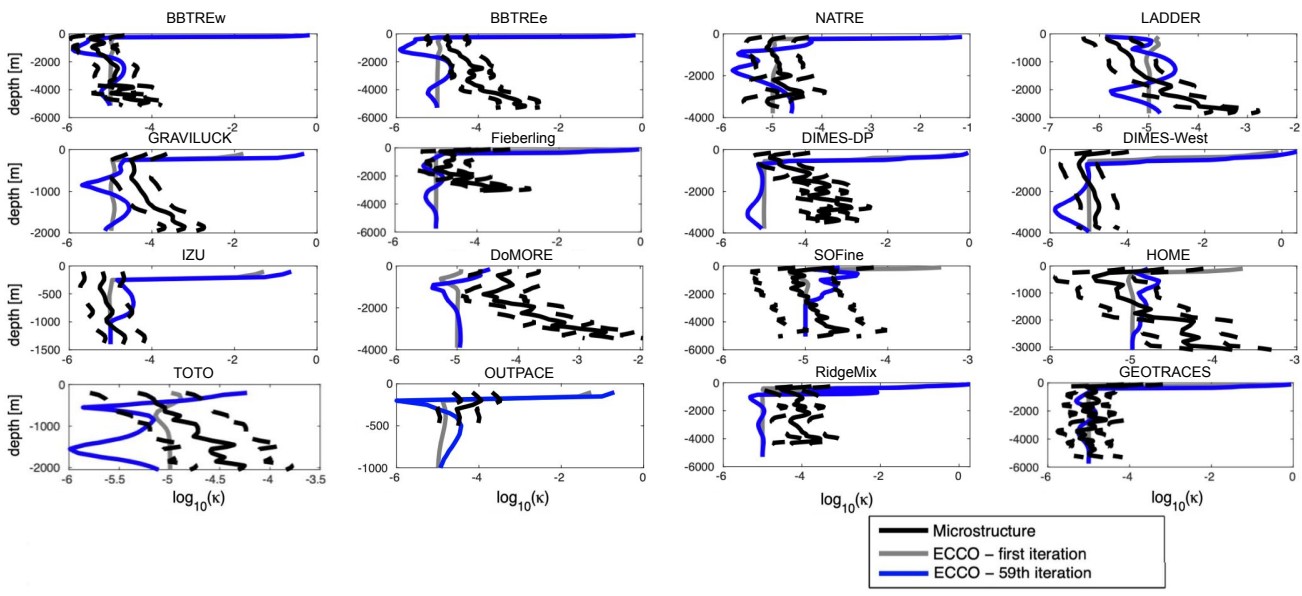

**Figure 5.** In each panel, shown are $\kappa_\rho$ profiles averaged over 16 example microstructure observation campaigns (see Fig. 2): from the first iteration of the optimization (E-CTRL$_0$ - grey curve), from the (final) fifty-ninth iteration of the optimization (E-CTRL - blue curve), and from the full-depth microstructure observations (black curve) presented in *Waterhouse et al.* (2014) (see their Fig. 6). Approximate factor of three uncertainties flanking the $\kappa_\rho$ profiles from microstructure are shown with dashed black curves.

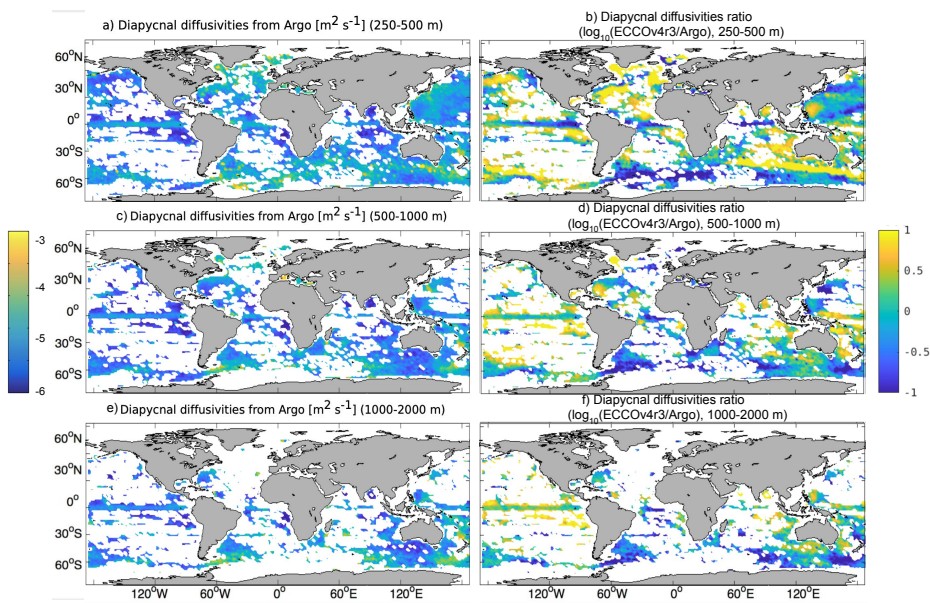

**Figure 6.** Shown are (a,c,e) the base-10 logarithms of $\kappa_{\rho,Argo}$ (units in m$^2$ s$^{-1}$) and (b,d,f) the base-10 logarithms of the ratios of the time-averaged $\kappa_{\rho,ECCO}$ to $\kappa_{\rho,Argo}$. Panels a-b show an average over 250-500 meters depth. Panels c-d show an average over 500-1000 meters depth. Panels e-f show an average over 1000-2000 meters depth. White areas in the ocean indicate insufficient Argo data to derive $\kappa_{\rho,Argo}$.

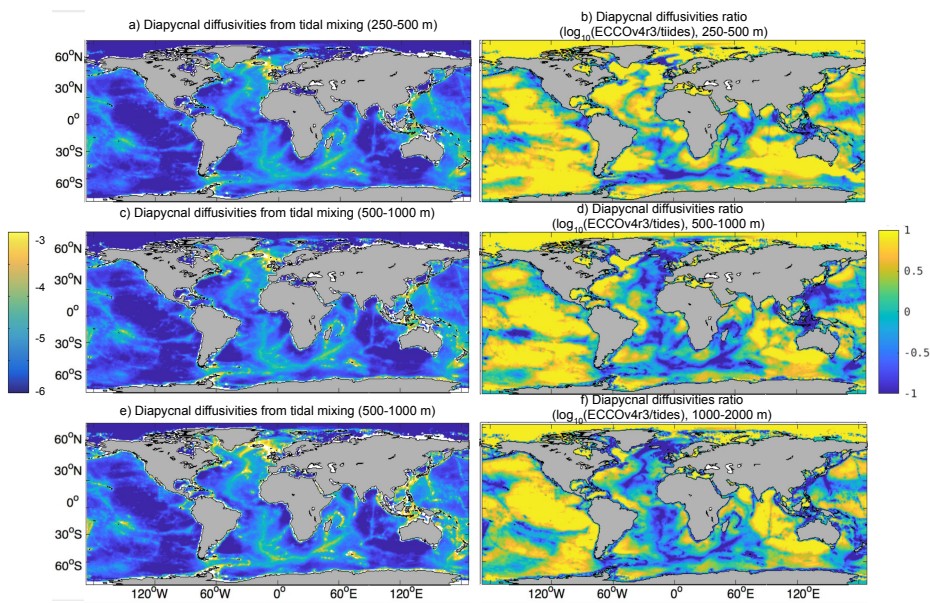

**Figure 7.** Shown are (a,c,e) the base-10 logarithms of $\kappa_{\rho,tides}$ (units in m$^2$ s$^{-1}$) and (b,d,f) the base-10 logarithms of the ratios of the time-averaged $\kappa_{\rho,ECCO}$ to $\kappa_{\rho,tides}$. Panels a-b show an average over 250-500 meters depth. Panels c-d show an average over 500-1000 meters depth. Panels e-f show an average over 1000-2000 meters depth.

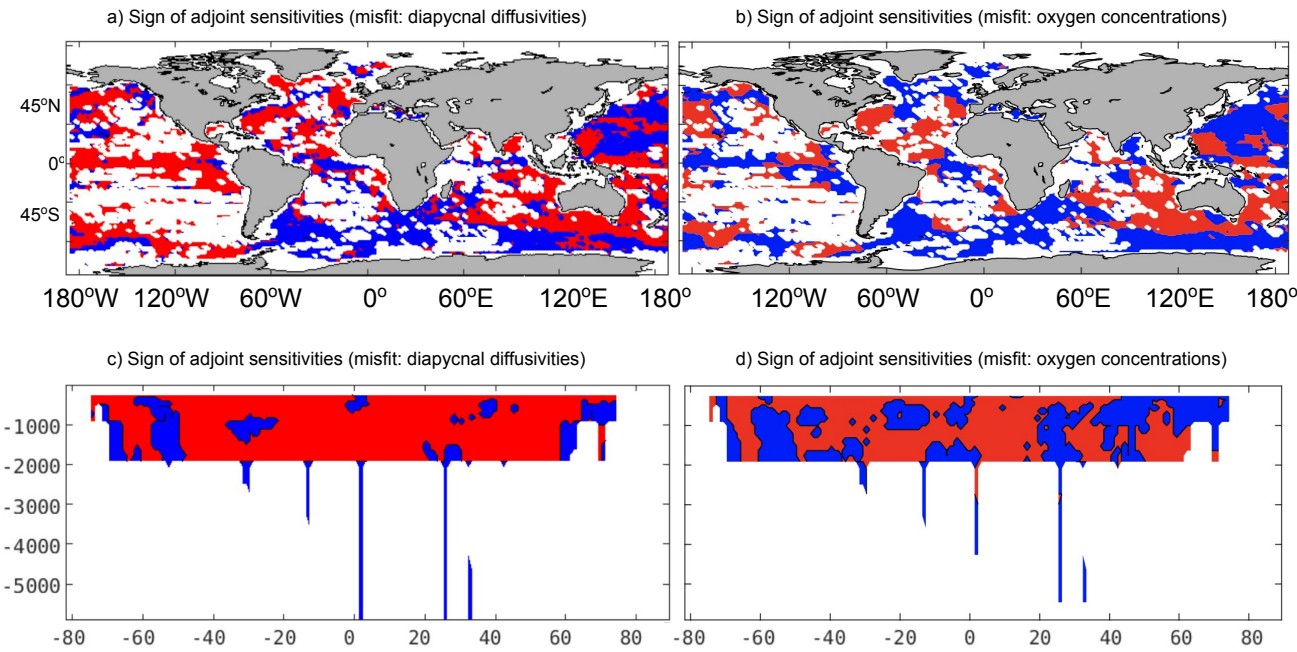

**Figure 8.** Adjoint sensitivity sign comparisons: Results from $E_\kappa$–using the *Whalen et al.* (2015) and *Kunze* (2017) products–(panels a and c) and $E_O$ (panels b and d) are shown for the adjoint sensitivities (units in s m$^{-2}$) with respect to $\kappa_\rho$: averaged over 250-2000 meters depth (panels a-b) and zonally averaged (panels c-d). The red regions indicate that the adjoint sensitivities are positive ($\partial J/\partial \kappa_\rho > 0$) and blue regions indicate negative adjoint sensitivities. $\kappa_{\rho,Argo}$ and $\kappa_{\rho,CTD}$ are the only quantities used in the misfit calculation of an adjoint run shown in panels a and c. The climatological oxygen concentrations from the World Ocean Atlas (2013) are the only observations used in the misfit calculation of a separate adjoint run shown in panels b and d. The adjoint sensitivities in panels a and c are computed offline (i.e., not using ECCO, but by plugging in the value the model reads in for the base-10 logarithm of $\kappa_\rho$ and comparing that with the above observationally-derived base-10 logarithm of the $\kappa_\rho$ products using the finescale parameterization via Eq. 2). The adjoint sensitivities in panels b and d are computed online (i.e., using ECCO, which uses the base-10 logarithm of $\kappa_\rho$ as a control variable). The white regions are locations with bathymetry or insufficient observations. The adjoint sensitivities are calculated over just one year (1992).

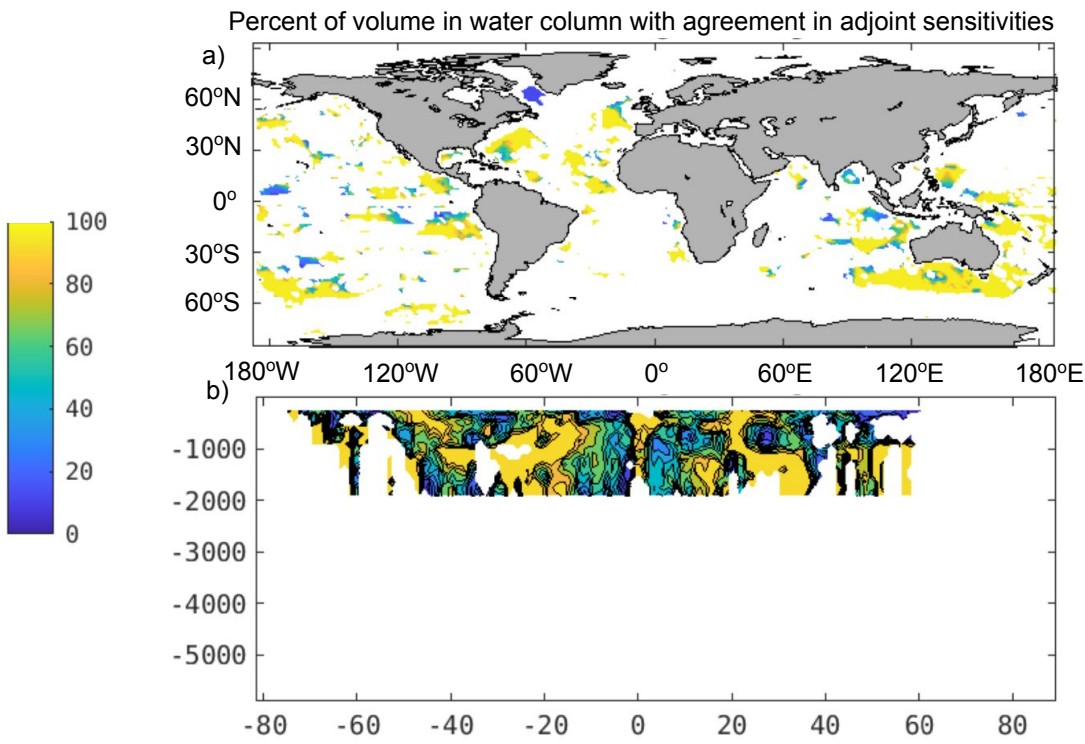

**Figure 9.** Adjoint sensitivity sign comparisons: Shown are the percents of volume over the water column for each horizontal location (panel a) and percent of volume over all longitudes for each depth and latitude (panel b) where the sign of $\partial J/\partial \kappa_\rho$ agrees between $E_\kappa$–using the *Whalen et al.* (2015) and *Kunze* (2017) products–and $E_O$. The white areas are locations where the disagreements between $\kappa_{\rho,ECCO}$ and $\kappa_{\rho,Argo}$ supplemented with $\kappa_{\rho,CTD}$ are within three times the value fo the observationally-derived $\kappa_\rho$ so these were excluded.

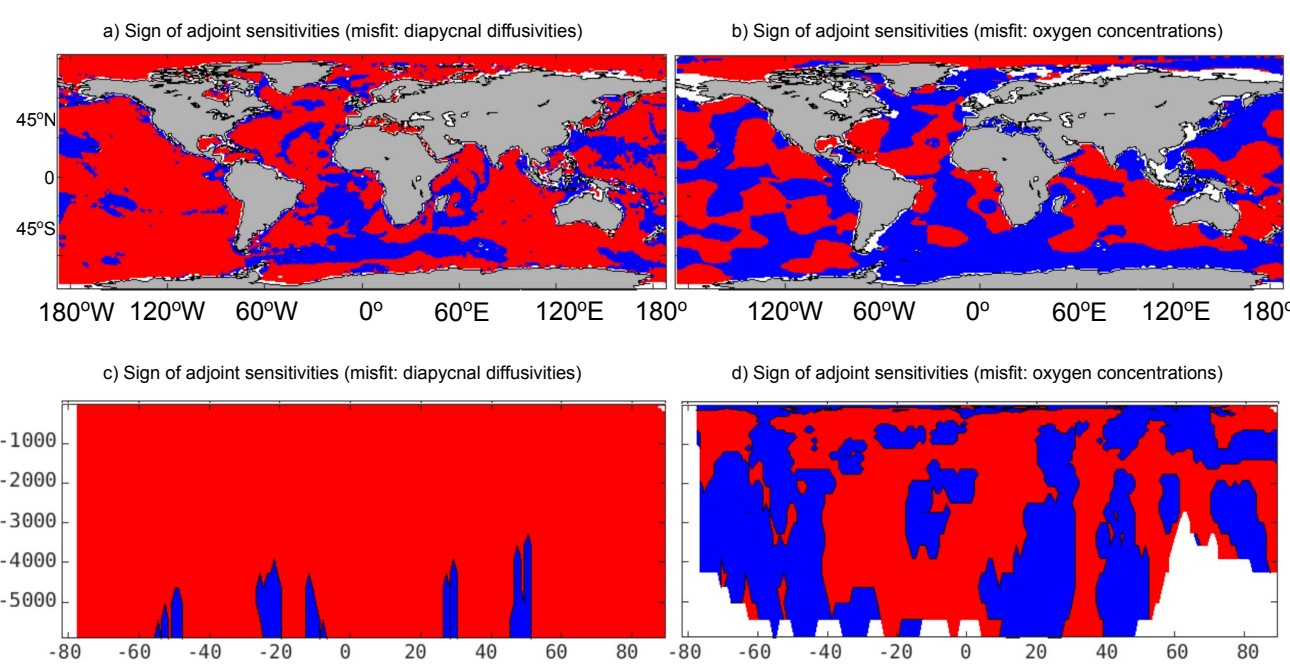

**Figure 10.** Adjoint sensitivity sign comparisons: Same as Figure 8, but with the *de Lavergne et al.* (2020) product, $\kappa_{\rho,tides}$.

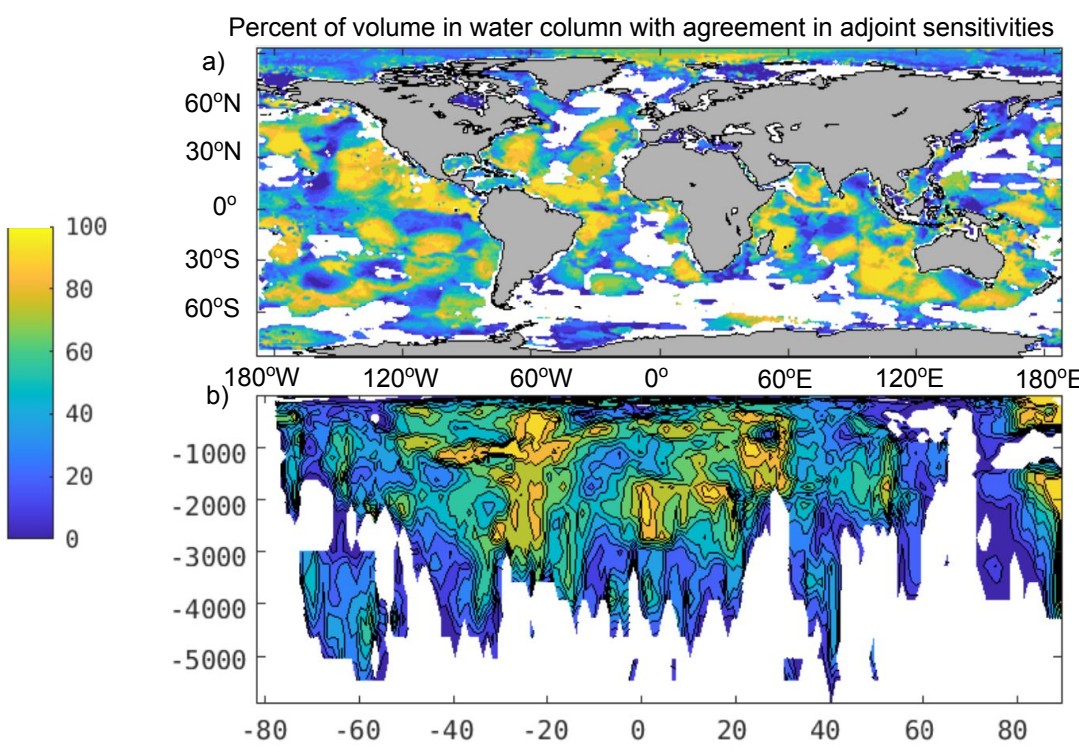

**Figure 11.** Adjoint sensitivity sign comparisons: Same as Figure 10, except using the *de Lavergne et al.* (2020) product, $\kappa_{\rho,tides}$.

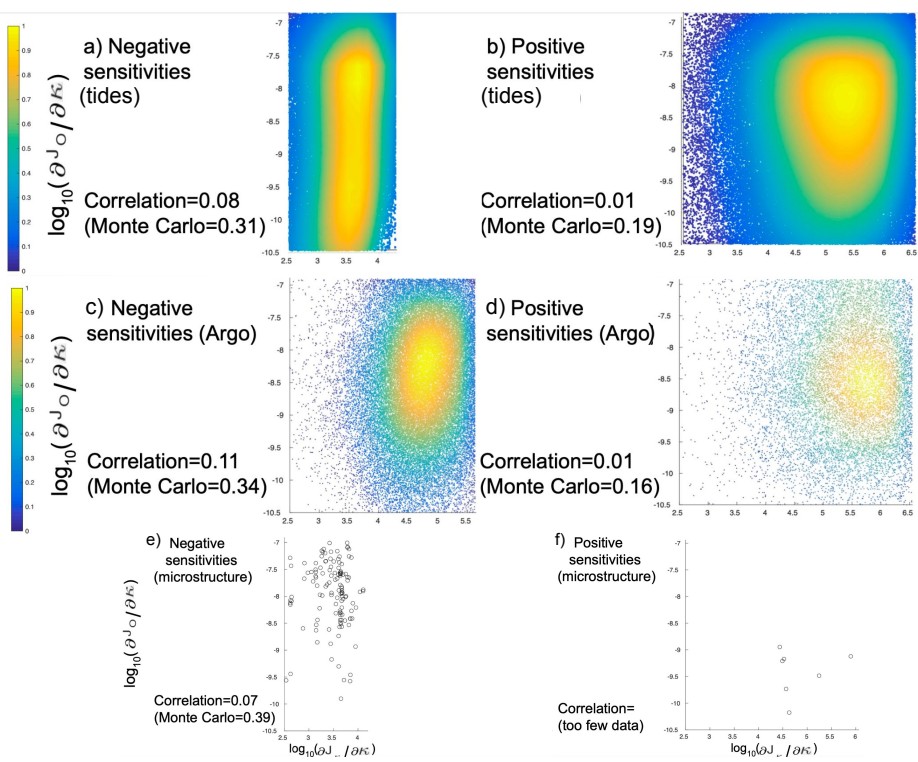

**Figure 12.** Shown are scatterplots between the adjoint sensitivities from $E_O$ and $E_\kappa$ where they are both negative (panels a, c, and e) and where they are both positive (panels b, d, and f), where $E_\kappa$ has its adjoint sensitivities calculated with: the tidal mixing product $\kappa_{\rho,tides}$ (panels a-b), Argo-derived $\kappa_\rho$ ($\kappa_{\rho,Argo}$ and $\kappa_{\rho,CTD}$; panels c-d) or microstructure-inferred $\kappa_\rho$ (panels e-f). Only the adjoint sensitivities where the differences between $\kappa_{\rho,ECCO}$ and observational $\kappa_\rho$ products are statistically significant (greater than a factor of three) and where the differences between oxygen concentrations from ECCO and those from the World Ocean Atlas (2013) are statistically insignificant (within 2% of the latter) are included. The correlations for all of the data points shown in each panel are listed. Also listed below each panel are the maximum possible correlations from a Monte Carlo-based approach in which 10,000 random samples of $\kappa_\rho$ within the uncertainties of the observational $\kappa_\rho$ products are used to recompute the adjoint sensitivities for $E_\kappa$.

**Table 1.** The latitude and depth ranges of each observationally-derived product from a parameterization used in this study. The longitude range for each dataset spans $(180^oE, 180^oW)$. Also listed is the time period of the observations each product is based on and the range of values in each product (to the nearest order of magnitude in units of $m^2s^{-1}$).

| data source | range [$m^2s^{-1}$] | latitude range | depth range | time period |
|---|---|---|---|---|
| Argo ($\kappa_{\rho,Argo}$) | $(10^{-7}, 10^{-2})$ | $(75^oS, 75^oN)$ | (250,2000) | 2006-2014 |
| Ship-based CTD hydrography ($\kappa_{\rho,CTD}$) | $(10^{-8}, 10^{-3})$ | $(77.35^oS, 78.70^oN)$ | (173,6044.5) | 1981-2010 |
| Climatology and seafloor ($\kappa_{\rho,tides}$) | $(10^{-8}, 10^{-2})$ | $(90^oS, 90^oN)$ | (surface,seafloor) | N/A |

**Table 2.** Listed are the ECCO simulations performed and analyzed in the present study as well as the observationally-derived data or measured data included in each simulation. Either observationally-derived data or measured data are included in the experiments through its misfit calculation (Eq. 1). Here, $\kappa_{\rho,obs}$ denotes an observationally-derived $\kappa_\rho$ product derived from a parameterization ($\kappa_{\rho,Argo}$ and $\kappa_{\rho,CTD}$ or $\kappa_{\rho,tides}$) or inferred from microstructure ($\kappa_{\rho,micro}$), $\epsilon = \kappa_\rho N^2/0.2$ indicates an observationally-derived dissipation rate ($N^2$ is the stratification from the World Ocean Atlas or *WOA* (2013)), and $O_2$ is the climatology of measured oxygen concentrations from *WOA* (2013). The misfits for the experiments with $\kappa_\rho$ and $\epsilon$ are calculated using Eq. 2.

| experiment | observationally-derived data | measured data |
|---|---|---|
| E-CTRL | N/A | see Section 2.2.1 |
| $E_\kappa$ | $\kappa_{\rho,obs}$ | N/A |
| $E_O$ | N/A | $O_2$ [*WOA*, 2013] |
| $E_\epsilon$ | $\kappa_{\rho,obs}$ | T/S [*WOA*, 2013] |

**Table 3.** The control variables that ECCO inverts for and optimizes. Some of these control variables are initial conditions only (indicated with the "initial condition" column). Other control variables are time-varying (indicated with the "time-varying" column). The rest are not initial conditions, but also are time-independent. Also noted is whether the control variable's field is two-dimensional or three-dimensional–there are no control variables that vary in both time and over all locations and depths of the ocean.

| control variable | initial condition? | time-varying? | dimensions |
|---|---|---|---|
| sea surface heights | yes | no | 2 |
| ocean velocities | yes | no | 3 |
| temperatures | yes | no | 3 |
| salinities | yes | no | 3 |
| Redi coefficients (*Redi*, 1982) | no | no | 3 |
| Gent-McWilliams coefficients (*Gent and McWilliams*, 1990) | no | no | 3 |
| $\kappa_{\rho,bg}$ | no | no | 3 |
| surface forcing fields | no | yes | 3 |

**Table 4.** Listed are the percent volumes where the signs of the adjoint sensitivities agree between $E_\kappa$–for both the *Whalen et al.* (2015) and *de Lavergne et al.* (2020) products–and $E_O$ for different regions of the ocean. The boundaries of subtropical/equatorial regions are set to be at $20^oN/S$. The boundaries of subtropical/subpolar or subtropical/Southern Ocean regions are set to be $50^oN/S$. The tropical Oxygen Minimum Zones (OMZs) are where oxygen concentrations are less than 2 ml/l between $20^oS$ and $20^oN$. The percentages are only calculated where sufficient observations are available to derive $\kappa_\rho$ and where the difference between the model-calculated and observationally-derived $\kappa_\rho$ is greater than the uncertainty (i.e., three times the observationally-derived $\kappa_\rho$). In parentheses are the same, except for the dissipation rates, $\epsilon_\rho = N^2 \kappa_\rho / 0.2$, where $N^2$ is the stratification and $0.2$ is an empirical coefficient (see, e.g., *Gregg et al.* (2018)).

| region | percent agreement (*Whalen et al.*, 2015) | percent agreement (*de Lavergne et al.*, 2020) |
|---|---|---|
| Global | 60.8% (59.9%) | 51.4% (58.8%) |
| Subtropics | 72.3% (72.9%) | 53.2% (58.7%) |
| Tropical OMZs | 61.2% (60.2%) | 58.3% (56.5%) |
| Subtropical South Pacific | 76.9% (79.2%) | 53.0% (56.8%) |
| Tropical Pacific | 57.9% (54.9%) | 57.2% (56.3%) |
| Subtropical North Pacific | 60.4% (59.6%) | 48.3% (44.3%) |
| Southern | 49.5% (47.9%) | 32.0% (28.4%) |
| Indian | 67.8% (68.7%) | 58.8% (56.8%) |
| Subtropical South Atlantic | 44.6% (35.8%) | 33.1% (30.7%) |
| Tropical Atlantic | 62.1% (62.1%) | 69.5% (54.5%) |
| Subtropical North Atlantic | 84.7% (85.4%) | 56.7% (55.2%) |
| Subpolar North Atlantic | 12.7% (12.8%) | 37.4% (36.3%) |