# Peer review of "Tracer and observationally-derived constraints on diapycnal diffusivities in an ocean state estimate"

_Ocean Science, 2021_

## Author Response (AR1)

Dear editor,

We are resubmitting the manuscript, "Tracer and observationally-derived constraints on diapycnal diffusivities in an ocean state estimate," to *Ocean Science*. We have performed some additionally analyses, as suggested by the reviewers. Specifically, we now include the *de Lavergne et al.* (2020) tidal mixing product in our analyses, with results in a couple of figures. We also include two new figures of comparisons of diapycnal diffusivities between various observational products and our model as well as added some textual clarification and edited some typos, many of which the reviewers rightfully pointed out. We address each of the reviewer's points below.

**1 Reviewer #1 (Comments to Author (shown to authors):**

- State estimates like ECCO take an important place in our field. It bridges a gap between observations and numerical models. Like observations and numerical models, it has its own pros and cons. This paper particularly looks at vertical mixing resulting from the state estimate and tries to understand and improve this number. It does so by first establishing how good it is (comparing to observational products), and then trying to add the tracer oxygen, to see if this could provide extra information to constrain and improve vertical mixing estimates in ECCO. Although it seems to do so, the results are somewhat disappointing, as the authors also notice. Regardless, this is a result in its own and therefore worthy contribution to publish. Before publication, I do think there are aspects of the analyses that need

to be addressed and places where clarification to be made. Hence, with some major revisions, I think this could become a good publication. Comments are in order of text and thoughts. Major comments indicated.

- **Thank you for the thoughtful review. We have performed additional analyses as well as added some textual clarification, as described below.**

- L20-25 – GM is not mixing, I think. It is advection.

- **The GM coefficient has been considered a "mixing parameter" in the literature, but you are right that it is effectively an advective term. We have changed the text to say "Ocean mixing is typically conceptualized in terms of diffusion along and across isopycnal surfaces. Subgrid-scale transport of isopycnal thickness (or bolus), which is effectively an advective contribution to tracer budgets, also must be parameterized. Ocean models often represent these unresolved processes with three parameters:..."**

- L35 – I don't understand what is meant with the rotation derived component... Is this about along isopycnal rotation? Perhaps better to explain this more carefully.

- **The formulation of *Katsumata* (2016) separates out the parallel and perpendicular directions of the horizontal components of the eddy advection tensor to density contours on a 1000 dbar surface. The perpendicular direction is the component that corresponds to the Gent-McWilliams scheme, while the parallel direction includes the rotational eddy flux, which is part of the eddy flux not in the Gent-McWilliams scheme. The**

formulation of the rotational component of eddy transport in ECCO is different from that of *Katsumata* (2016), which is why a comparison was not performed in that paper. We have modified the text to say: "... the formulations of the perpendicular and parallel components of the eddy advection tensor relative to isopycnal surfaces are not the same in many models as in the observationally-derived Gent-McWilliams coefficient product."

- Major L115 – The results are compared to observations of mixing. This is good. But I think it would be worth to compare the results against the parametrization of Lavergne et al 2020. This is semi-observational estimate compares well to Whalen's work, but covers all grids and full depth. Without data-gaps this may provide better comparison product for the model. The work is also "averaged", like that from ECCO, and not instantaneous like that from observations. Another reason why this might be a better comparison for this effort. Of course, neither observation, nor Lavargne's method are perfect for comparison, but having them both really adds value.

- **We now include the diapycnal diffusivities from *de Lavergne et al.* (2020) in our manuscript. Below about 2000 or so meters, the diapycnal diffusivities from ECCO are too small in comparison to every observationally-derived product. However, in contrast to the comparison with microstructure-inferred and the Argo-derived diapycnal diffusivities, the diapycnal diffusivities from *de Lavergne et al.* (2020) are smaller than the ones from ECCO in the upper 2000 or so meters.**

- Major - L120-125 - I think you should update WOA13 to WOA18, at least 5 years more data, including BGC data. This is certainly worth the upgrade, and it should be straightforward. Also, which is used, monthly means, annual means?

- **We used annual means because ECCO uses a constant diapycnal diffusivity field. At the time we began this project, we also chose to use WOA13 (WOA18 wasn't available yet) because the initial conditions for oxygen concentrations in the model are derived from WOA13. Thus, the regions where there are disagreements between WOA13 and the model are regions where model drift must be occurring due to model specification/parameter errors. The adjoint sensitivities tell us how to change the model parameters so that the model won't drift as far from the initial conditions. Further, much of the extra data in the WOA18 database is at high latitudes where we don't have diapycnal diffusivity data from microstructure or Argo. If you still find this to be an important update for our manuscript, we could perform the adjoint sensitivity runs again, but we would need to get computer resources to run the additional simulations.**

- L170 – for by?

- **Corrected, thanks.**

- Major - L200 – N2 is used from WOA13. However, WOA13 has problems of its own: to start with is is averaged on isobars and not on isopycnals. This leads to false mixing and we also see that N2 is often unstable. However, on average time scales

of months, the ocean is "stable" as overturns generally have much shorter duration. Have you made N2 stable? There are simple tools for this in the TEOS-10 manual. I think this is important for your comparison.

- **Yes, we did not explain this, but we use TEOS-10 to calculate $N^2$ from WOA13 data so that it's stable. We also repeated our calculations using the $N^2$ provided in the *de Lavergne et al.* (2020) data and found no qualitative difference.**

- L220 – Fig 2a = fig2

- **Corrected, thanks.**

- Major - L225-230 – Equation 1 and surrounding text. The results are sensitive to prior choices made by the modeler, in the form of weights. Here a certain choice is made, but sensitivity to this choice is not studied. The sensitivity is both to the equation weighting (as done here), and to the variable weighting. The latter means that the a-priory estimate of the variable, influences the result. This should as well be considered somehow because this a-priori estimate can be wrong or vary within a range. I see no sensitivity to either of these choices. Or at the very least, a discussion on this possibility. Places to read about this are McIntosh and Rintoul 2001, but also Groeskamp et al 2014. I'm aware that this can be a pain and sometimes limited by computer power. I don't know the situation for this study. I therefore strongly recommend to do it if you can, it will strengthen the confidence in your model and the results. IF you really can't do this, then at least discuss it.

- **As for choices made by the modeler (equation weighting), we performed**

Monte Carlo simulations to consider the uncertainties in the diapycnal diffusivity estimates as well as the uncertainties in the weights themselves for the offline estimates of the adjoint sensitivities. Our results are very close to the same when considering only the uncertainty in the diapycnal diffusivity estimates. As for sensitivity to variable weighting, because we only consider one variable at a time in the context of adjoint sensitivity experiments (not optimizations), we do not consider variable weighting as a factor, but in practice, if oxygen is used to help constrain the ECCO-estimated diapycnal diffusivities, then this is an issue to consider. We show how the first iteration's estimate of the diapycnal diffusivities is different from the final iteration's estimate because we agree with this point. We now include the following statements in the discussion section: "The ECCO-estimated $\kappa_\rho$ can also be sensitive to the *a-priori* estimate of $\kappa_\rho$ and we showed how one particular initial guess ($10^{-5}$ m$^2$ s$^{-1}$ everywhere) can evolve from the first optimization iteration to the final one. ... If biogeochemical tracers are included in the misfit calculation in an optimization run, their impact on variables such as $\kappa_\rho$ would depend upon how they are weighted relative to the physical variables (e.g., temperature, salinity, and pressure)."

- 240-245 – Below eq 2, do you here mean to fill in equation 1 into 2, so you then get a "y". You refer to "y", where there is none in eq 2.

- **The "y" refers to the y in Eq 1, but we have removed all references to variables in Eq 1 after Eq 2 for clarity.**

- Section 2: You focus on analyzing $k_{rho}$. But what about all other variables. It could well be that if you better constrain $k_{rho}$, error is increased on other variables. Hence, how do you keep watch that the rest of the variables are not underperforming?

- **It is likely that errors in one of the three mixing parameters bleed into each other, which makes each of their values biased. This is an issue that can potentially be resolved in a similar way that issues with atmospheric forcing fields have been handled, which is to only allow them to vary within a certain range of reanalysis products/observational estimates. This issue is important, but is a next step beyond the scope of our current study because we are motivating new optimizations of ECCO with our manuscript, not attempting them until we have reason to do so. Sidenote: in a previous version of this manuscript, we included results for the adjoint sensitivities of oxygen with respect to the diapycnal diffusivities, the Redi coefficients, and the Gent-McWilliams coefficients. They're all different, of course, and some are larger than others. The adjoint sensitivities for the Redi coefficients, for example, are the smallest, and there are many studies that have examined how strongly oxygen can respond to changing the Redi coefficients (e.g., *Gnanadesikan et al.*, 2013). The same type of analysis we have performed with the diapycnal diffusivities was performed with the Redi coefficients, and the signs of the adjoint sensitivities agreed about 50% of the time (what you'd expect from random chance). This could be because there are imperfections with the scaling factors used in the calculation of observationally-derived Redi**

coefficients; in fact, if we scale down the observationally-derived Redi coefficients by a constant factor between 2 and 3, the percent volume agreement between the adjoint sensitivities goes up to 60-70%. More work should be done on observationally-derived Redi coefficients before we used them for our analysis. And the same analysis cannot be done with the Gent-McWilliams for reasons we've explained and are discussed in *Katsumata* (2016).

- Line 283-284 – it says, "not shown". I would have liked to see this in the same figure, I think. This would be informative, and it sounds like you have the data anyway.

- Because the uncertainty is a factor of three and we're plotting the profiles on a depth versus log-diffusivity scale, the uncertainty looks like it tracks the profiles linearly, just offset by about one-third on the abscissa. So we didn't think it would be important to explicitly show the uncertainties. However, given your next concern, we decided to show the uncertainties in the microstructure profiles of the diapycnal diffusivities for both these average profiles and for individual campaigns to demonstrate how ECCO's diapycnal diffusivities can be within the uncertainties for some campaigns with large uncertainties and well outside of the uncertainties for other campaigns.

- Major - Section 3.1 – Observation are instantaneous profiles in a time varying field. We here are dealing with averages in a model. So I was wondering, is taking an average to compare the right thing to do? Is it not better to check individual profiles

and add their error together somehow? I mean, averaging a diffusivity is always a strange thig to do. Oké, we must do something, I agree with that. But it seems like many profiles that are physically unrelated are now averaged in one big profile that therefore loses all sense of information. Is there not a possibility to compare and show some selected individual profiles and come up with a metric that addresses the error between the obs and the model, for the sum of each individual profile? I think that would be more meaningful.

- **_de Lavergne et al._ (2020) did this kind of comparison for the reasons you argue. We argue in the Appendix that the diapycnal diffusivities don't significantly vary over time except in a few select regions of the ocean, like in the top 1000 meters between 40-50$^o$N in the North Atlantic Ocean. So taking an average (a geometric one, in particular) makes sense to show. However, we now also show comparisons between the diapycnal diffusivities from individual microstructure campaigns and ECCO. We also show the approximate factor of three uncertainties on the microstructure profiles.**

- Grid lines in Fig 3. And Fig. 7 would be very helpful.

- **We're not sure what you mean because those figures are not maps, but we have edited both of these figures in any case. We hope they're presentable.**

- L295 – Do air-sea fluxes influence the results here or is this stuff too deep. What about advective processes?

- The diapycnal diffusivities are all below the mixed layer and our comparisons are therefore for values below the mixed layer. However, over long (multi-year) time scales, the air-sea fluxes could influence the results because, as we show in the Appendix, the steric sea level budget term related to the diapycnal diffusivities are associated with the steric sea level budget term related to the air-sea fluxes. Thus, we have changed the text to say "The errors in $\kappa_{\rho,ECCO}$ could be partially compensating for errors in the vertical component of the along-isopycnal diffusivity tensor, erroneous air-sea fluxes due to inconsistencies between the sea surface and atmospheric forcing fields, and/or the presence of numerical diffusion." As for advective processes, if they influence the gradients in oxygen concentrations, then that could explain at least part of the disagreements in the signs of the adjoint sensitivities because the diapycnal diffusivities are not the only factor influencing oxygen concentrations. However, our results suggest that oxygen could play an overall positive role in estimating a more realistic diapycnal diffusivity field. We will need to perform an optimization to show this definitely.

- L310 - There is no table 2.2.

- **Corrected, thanks.**

- Section 3.3 discussion figure 5. How would this figure look if the differences where small? I mean, it would still be red or blue right? It only gives a sign, but does it say something about the magnitude of the mismatch? It only seems to give a direction of the mismatch. Is it not better to also say something about the magnitude of the

mismatch?

- **It depends on the sign of the small value (could be red or blue). If the value is small then the sign of the adjoint sensitivity shouldn't matter much. What we want the reader to see in our previous version's Figure 5 is the correspondence of signs between the different adjoint sensitivities. We white out regions where the differences are small by an uncertainty-based metric in the following figure (Figure 6 of the previous version) to quantify the percent volume over which the signs of the adjoint sensitivities agree. We show the magnitudes of the adjoint sensitivities (in Figure 7 of the previous version) when we show the scatterplots and quantify their correlations. In short, we focus on the agreement in sign in the figure in question, we indicate where the magnitudes are small in the following figure, and we focus on correlations between the magnitudes with our final figure.**

- L342 – We cannot compare white and white. Please make these locations referred to, gray in figure 6.

- **Here we're referring to the white regions in Figure 6 that are not white in Figure 5.**

- Major - L360 – Oke, O2 gives extra information. But which part? Is it the vertical gradients of O2? If so, when do they give extra information compared to N2? It should be possible to argue this upfront and discuss (not do, yet) which other tracers could be of interest in future work. That is, when N2 does not have vertical

gradients, but O2 does, then for that region it is worth adding O2. If both don't have gradients there, then probably some other tracer may work. A discussion on this would be helpful. Possibly a figure.

- **The similar magnitude of the correlations between the vertical gradients of oxygen and the diapycnal diffusivity estimates and the correlations between the adjoint sensitivities suggests that the information that oxygen provides about the diapycnal diffusivities is related to the vertical gradients in oxygen. Our comparisons with the dissipation rates instead of the diapycnal diffusivities suggests that $N^2$ doesn't appear to provide any information in addition to that which oxygen provides, but we did an additional calculation where we check whether this conclusion is contingent upon the vertical gradients in $N^2$ and in oxygen. We found that the correlation between the vertical gradients in $N^2$ and in oxygen is about $0.25$, which is higher than the correlation between the diapycnal diffusivities and the vertical gradients in oxygen. This is not perfect, so we performed our comparisons between the adjoint sensitivities using only regions where the vertical gradients in $N^2$ are relatively small and the vertical gradients in oxygen are relatively large. The correlations are about the same, again suggesting that our results are independent of the stratification. We have edited the text to say: "The spatial correlation between the annual mean vertical gradients in oxygen and the annual mean vertical gradients in stratification ($N^2$) from the World Ocean Atlas (2013) is about $0.25$, but this is only suggestive. To determine whether**

oxygen provides information about stratification (and through stratification, about $\kappa_\rho$),..."

- Section 4 – Could oxygen also increase the errors? It is not said that things get better. I think it is argued that the observation of O2 is accurate and thus it provides information. But for some places where there is lack fo data or the values are not averaged the right way in constructing the climatology, errors could be substantial. You have regions where estimates improve, and where they get worse. This could be why. This is related to the previous comment. Again, a discussion would be helpful.

- **The diapycnal diffusivities would get worse in regions where the signs of the adjoint sensitivities disagree if oxygen is the only information provided as a constraint. In practice, oxygen would be weighted as a constraint along with temperature and salinity, which may not lead to worse diapycnal diffusivities. This would take some experimentation to get the weights such that the diapycnal diffusivities are improved. All we are saying here is that the diapycnal diffusivities could improve if we include oxygen in the misfit. It is a fair point to suggest that the agreements between the adjoint sensitivities are highly correlated in regions where we have diapycnal diffusivity and oxygen data and/or the diapycnal diffusivities might actually have their errors increased where we do not have data. In fact, when we mask out all regions where we don't have both diapycnal diffusivity data from *de Lavergne et al.* (2020) and oxygen data from WOA13, the correlations between the adjoint sen-**

sitivities go up to $0.47$ in the Monte Carlo simulations. We added the following sentence: **"If we only consider comparing locations where we have both observationally-derived $\kappa_\rho$ data and oxygen data, our results are qualitatively the same and the correlations increase to as much as $0.47$ in the case of the *de Lavergne et al.* (2020) data using a Monte Carlo approach."**

- L380-381 – I think with everything going on in ECCO also leading to errors compared to the real world (e.g., parameter choices), I find it is too strong a statement to simply say that data alone is insufficient. You do tone down this point a little in the sentence below, but I still think this statement as written down here is too strong.

- **Okay, we have rephrased this statement: "Some model specifications would lead to errors in $\kappa_{\rho,ECCO}$ even in the presence of globally complete hydrographic observations (see Section 4.2), but we investigated whether $\kappa_{\rho,ECCO}$ can benefit from new information."**

- As mentioned before, I think this might also be a place to discuss that the data product itself (WOA) has errors in there due to interpolations, and averaging techniques (e.g., horizontal instead of isopycnal averaging).

- **This is true, but we only included observations where they were taken, not interpolated/averaged (except in the vertical), in the misfit function in the simulations we performed. We have added the following sentences: "We assumed a factor of three for the observationally-derived $\kappa_\rho$ and**

2% of the oxygen concentrations. These do not account for interpolation/averaging errors that entered the data prior to our calculations, but are conservative estimates nonetheless. The observational uncertainties affect the weights given in the misfits that enter the adjoint sensitivity calculations and our Monte Carlo simulations of the correlations between the adjoint sensitivities account for the possibility that these weights are misspecified."

**2 Reviewer #2 (Comments to Author (shown to authors):**

- In this study, the authors used the ECCO framework to explore a few ways to better constrain the estimates of diapycnal diffusivity. Based on a set of sensitivity analyses, they investigated the impacts of including diapycnal diffusivity estimates that are obtained from microstructure measurements or inferred from CTD measurements, as well as dissolved oxygen measurements. They concluded that both ways could improve the presentation of diapycnal mixing in ECCO. I think this is potentially a very important paper, and I really appreciate the authors put tremendous efforts to address this interesting yet difficult question. The paper should eventually be published. However, before I recommend acceptance, I would like the authors to clarify a few questions and concerns I list below.

- **Thank you for taking your time to provide thoughtful comments.**

- My major concern is the definition of ECCO diapycnal diffusivity used in this study.

From what I know about ECCO v4, there are many components involved in the calculation of vertical fluxes as well as diapycnal diffusivity. The diapycnal diffusivity at least consists of three parts: the background diffusivity, which was adjusted through the adjoint process; the parameterized part based on Gaspar et al. (1990); and convective adjustment. In this paper, the authors briefly described some of those terms. But it is still not clear to me what the exact definition of the diapycnal diffusivity the authors analyzed is. The combination of all or some of the components mentioned above? or just the adjusted background diffusivity? This information is critical for the interpretation of almost all the results presented in this study. And the authors should make that information more explicitly presented.

- **The MITgcm uses a quantity they call "diffkr" which is the diapycnal diffusion coefficient in their r-coordinate system and is referred to as a vertical diffusivity (not quite a diapycnal diffusivity). To transform this value into a diapycnal diffusivity that's equivalent to the observational product values, we need to subtract out the (3,3) entry of the along-isopycnal diffusivity tensor, which is not perpendicular to the isopycnal contours but parallel to them. After subtracting this from the vertical diffusivity, we are left with the diapycnal diffusivity. As for what this coefficient physically represents, it is the adjusted background diffusivity via the adjoint estimation process. In other words, it corresponds to the mixing that is not associated with the instabilities determined from Gaspar et al. (1990) or convective adjustment. We have edited this sentence to read: "Vertical mixing–diapycnal plus the vertical component of**

**the along-isopycnal tensor–is determined according to the Gaspar et al.**
**(1990) mixed layer turbulence closure, simple convective adjustment, and**
**estimated background $\kappa_\rho$ for internal wave-induced mixing." We should**
**clarify, however, that it isn't vital to determine the specific processes**
**the estimated mixing is representing. We have also added the follow-**
**ing sentences: "Here, $\kappa_\rho$ represents a combination of processes." To be**
**clear, the central point of our manuscript is that oxygen observations can**
**potentially help qualitatively infer where there is enhanced turbulence.**

- It is good that the authors reminded the readers a couple of times through the text
  that the "observed" diapycnal diffusivities based on either microstructure measure-
  ments or fine-scale parameterization with CTD measurements include uncertainties
  as well.

- **The uncertainties in these observational products are essential to con-**
  **sider to understand the purpose of our manuscript and we're glad our**
  **reiterations helped communicate this point.**

  Detailed comments:

- Lines: 3-5: The authors concluded that "the assimilation of existing in situ temper-
  ature, salinity, and pressure observations is not sufficient to constrain $\kappa_\rho$ estimated
  with ECCO". However, since the number of iterations or the adjoint runs in ECCO
  is limited, is it possible that by running more iterations, the ECCO diffusivities will
  be better adjusted to the truth, even without including other new datasets? From
  what has been presented in this paper, I don't believe that possibility has been ruled

out, and therefore the statement above is not solid, IMO.

- It's true that running more iterations of ECCO could adjust the diapycnal diffusivity field such that it's closer to that of the real ocean, but we now show in a new figure that this wasn't always true between the first and fifty-ninth iterations for each microstructure campaign. It's possible that the other estimated fields absorb some of the errors in the diapycnal diffusivity field due to the under-determined nature of the ECCO estimation problem. The purpose of our manuscript is not to sort out which would happen with additional iterations of ECCO, but we do show some evidence that your hypothesis may be closer to the truth, in which case it's entirely possible that infinite computational resources are required to converge to the truth. Given that future ECCO optimizations will likely be insufficient in number for the diapycnal diffusivity (or other) field(s) to converge 100% (especially considering their errors), we seek new data sets that could help guide the diapycnal diffusivity field within the number of iterations performed.

- Line 14: I did not find how this conclusion was reached.

- The evidence we have for this is in the correlations between the adjoint sensitivities for the experiments with oxygen in the misfit and those for the experiments with diapycnal diffusivities in the misfit. The adjoint sensitivities tend to agree in sign but are very well correlated, suggesting that the values of the diapycnal diffusivities would be different if one data set were assimilated instead of the other. This part of our abstract now

reads: "Information provided by more accurately measured dissolved oxygen concentrations is not equivalent to that from less accurately measured $\kappa_\rho$. However, we show that adjoint sensitivities of dissolved oxygen concentration misfits to the state estimate's control space typically direct $\kappa_\rho$ to improve relative to the Argo-derived and microstructure-inferred values."

- Line 141: The background $\kappa_\rho$ is time-invariant. But if the parameterized part and/or convective adjustment were included, it should be time-variant.

- **That is true, which is why ECCO only estimates the background diapycnal diffusivity field. We tried to make this more clear with the edits to a sentence we quoted above.**

- Line 152: What do "14-day adjustments" mean? Could the authors explain that?

- **The adjoint averages adjustments to the atmospheric forcing fields, which are re-estimated and then applied over 14-day periods. We have rephrased how we state this sentence: "Average adjustments to the wind stress, wind speed, specific humidity, shortwave downwelling radiation, and surface air temperature are re-estimated and then applied over 14-day periods."**

- Lines 156-158: This information is important but confusing. Could the authors elaborate on how the ECCO $\kappa_\rho$ used in this study was obtained? Is it based on parameterization (Gaspar et al., 1990)? or is it adjusted through the adjoint? or other ways?

- It is the background vertical diffusivity field that ECCO estimates through its adjoint minus the (3,3) component of the along-isopycnal diffusivity tensor to get the time-invariant background diapycnal diffusivity field. We tried to clarify this in the text.

- Lines 174-176: It seems that the authors were likely asked by other reviewers to add the appendix. I personally think it is not necessary. But I am OK if the authors choose to keep it.

- Because there are two different types of data assimilation systems that have different problems with their diapycnal diffusivity field, we had two manuscripts initially: one that's the main text and one that was an elaboration of the Appendix. We were unable to perform additional experiments using the data assimilation system of interest (NASA GMAO S2S) in the Appendix because we were not given access. We were able to perform additional simulations using another modeling system (without data assimilation), but exclude these simulations from this manuscript. We simply suggest that it's worth considering the equivalent problem we're pointing out in ECCO but with sequential data assimilation systems because there are likely consequences for forecasting systems.

- Line 200: It would be helpful if the authors could explicitly state the differences between the two runs $E_\kappa$ and $E_\epsilon$.

- The difference is that $E_\kappa$ includes the diapycnal diffusivities in the misfit function and $E_\epsilon$ include the dissipation rates in the misfit function. The

values of $\epsilon$ and $\kappa$ are related to each other through the Osborn (1980) relation so any qualitative differences between comparisons with $\mathbf{E}_\kappa$ and comparisons with $\mathbf{E}_\epsilon$ are related to the stratification. We edited the bullet points describing exactly how the $\mathbf{E}_\kappa$ and $\mathbf{E}_\epsilon$ were performed now.

- Lines 214-223: I am confused here. Why does using a previously derived product as an initial condition minimize model drifts? Also, how did the authors conclude that using other products as initial conditions would be worse than using that one?

- **The way we described where the initial conditions for oxygen concentrations come from should be improved. The initial conditions come from MITgcm/verification/global_oce_biogeo_bling/input in the model's package, which was derived from World Ocean Atlas (2013). Because the initial conditions are observationally-derived, we concluded that using other products as initial conditions would be worse choices. A figure in our manuscript shows the differences between the model and World Ocean Atlas (2013) product by taking the nearest neighbors to the model grid instead of introducing interpolation errors. Still, this figure shows there are larger errors in some regions than others, which suggests that model is drifting from the initial conditions in many regions.**

- Lines 234-237: Since uncertainties associated with undersampling (spatially and temporally) were not considered, the prescribed uncertainties are likely lower bounds. If so, the readers should be reminded about this.

- **This was discussed in response to a comment the other reviewer made.**

Regions where we do not have observations can disagree in the signs of their adjoint sensitivities (or lower the correlation with another experiment's adjoint sensitivities) because we don't have observations there. The sampling issue when calculating climatological fields could also be an issue because our weights for computing the misfits could be inappropriate if there are, for example, seasonally aliased values. This could also explain some of the disagreements in signs of the adjoint sensitivities (or lower the correlation with another experiment's adjoint sensitivities).

- Lines 296-299: As commented above, is it possible that with more iterations $\kappa_\rho$, ECCO will be closer to the observationally-derived $\kappa_\rho$?

- **Yes, but a new data set could conceivably accelerate the convergence to a more realistic diapycnal diffusivity in ECCO, which could be important because ECCO optimizations will not likely be iterated many more times than the version we're analyzing. However, the new figure we now include showing comparisons with individual microstructure campaigns suggests that more iterations can actually stray further the microstructure-inferred values than the first iteration.**

- Lines 356-363: Now I understand the purpose of the run $E_\epsilon$. It would help the readers if some of the information here were added around line 200.

- **Okay, we added an explanation earlier in the text. Thanks.**

- Line 389: What is the amount of the available DO measurements? Is it comparable to or much larger than T/S data? If not, not sure it will be that useful.

- In the World Ocean Database and Argo, there are less dissolved oxygen data relative to T/S, but the reason why oxygen can serve as a useful constraint on the diapycnal diffusivities is that oxygen provides unique information about gradients, as oxygen has a different source function and history compared to T/S. Further, oxygen concentrations can be weighted more than other physical variables (e.g., temperature, salinity, and pressure) to compensate for the relative dearth of oxygen concentration observations. We haven't performed this type of sensitivity analysis because we didn't perform optimization runs, but this has been considered in SOSE.

- Line 420: "microstructure CTD-derived $\kappa_\rho$"? Do you mean CTD-derived $\kappa_\rho$?

- These were CTD data taken (approximately) concomitant with the microstructure, which is all we were trying to say. We now say: "A preliminary analysis suggests that the percent difference between the full depth-averaged CTD-derived $\kappa_\rho$ from the finescale parameterization and the microstructure-inferred $\kappa_\rho$ at the same locations is indistinguishable from zero (1.68%), but the quality of the the CTD data taken concomitantly with microstructure has not been fully assessed."

- Lines 345-437: The last sentence is disconnected from the sentences before it.

- We have edited the text so that this text reads more smoothly: "A more complete representation and understanding of $\kappa_\rho$ is possible through these analyses and methods."

- Figure 4: since there are many data gaps (white area) in the right panels, it might make sense to use a different colorbar.

- **Because the purpose of this figure is to show that there are many regions where the difference between the ECCO-estimated diapycnal diffusivities are an order of magnitude different from observationally-derived ones, the only regions that need the reader's focus are the blue and red ones in panels b,d,f. The difference between the regions with white because there are no data and the regions with white because the ECCO-estimated diapycnal diffusivities are close to the observationally-derived ones is not very important if there are large regions of the ocean with red or blue colors. However, we have replotted this figure using a different colorbar to make it more apparent where there are no data and where there are small differences.**

**3 Reviewer #3 (Comments to Author (shown to authors):**

- The paper seeks to evaluate if we can improve oceanic mixing estimates on a global scale by considering assimilation constraints to some novel ocean parameters - diapycnal diffusivities (kp) and dissolved oxygen. To do that, the authors used an assimilation framework (mitGCM/ECCO), a biogeochemical model (BLING) and a reduced order cost function to estimate sensitivities of numerical parameters in an idealised assimilation scenario. The author concluded that a) a global analysis

using only classical ocean observations stays at min. 1 order away from in-situ, b) that the possible "innovations" by assimilating kp can reduce/increase kp in important regions and that most regions are in agreement with "innovations" driven by oxygen alone and c) that oxygen seems to provide extra information to constraint kp. The paper is well structured, concise, the English are clear, and the description of results are cohesive with figures/the general presentation/discussion. However, in my opinion, the paper lacks in detail and suffer from some oversimplifications, omissions, and insufficient discussion regarding some choices/results. The paper is 20% results/discussion but still lack important details to interpret the results. I felt the text missed on important aspects, which are skipped, not even mentioned at all, or mentioned too late - this mostly occurs at Introduction/Methods, but also in results. My major concern is that a lot of details are missing discussion: Aspects of Kp parametrisation in the MIT ocean are lacking, a priori decisions in terms of the inversion. The authors appear to know well the details of the observations estimates but not much detail is provided in terms of the numerical model. My point here is that some information or estimates are sprinkled throughout the text or the next session, generating a back-forth reading and raising too many questions along the way - causing the user to try to build bread from crumbles. I think the authors need to reorganize the paper again - the paper reads like an incomplete explorative analysis, without good justifications or reasoning regarding decisions and the way the work was set up - maybe this was chopped off? The abstract need a lift-up - it is not complete and could be more laser-focused. Results: There are some good insights here and there, but I found the paper lacks important milestones along the sections

to put the reader in the right mindset. Datasets are barely presented, figures are not self-described. What each simulation is using/how it is performed is very rushed and not clear from the start. The aspects of assimilation are not evenly formally described. Although I'm not a specialist in the ECCO model, the description here lacks enough detail. Maybe the author wished to simplify, but I think it end up cutting too much!?. Some de-facto nomenclature in data assimilation is brushed and only add to the confusion to the ones not familiar with ECCO. In short, I think the paper deserves a major review, mostly for clarity of what has been done and to reduce the guesswork, raised questions, and mental gymnastics from the reader. In general, I believe the paper results may be actual good results, there are some good insights, but the lack of attention to detail and description of what was performed put me in the uncomfortable position of having one step back in trusting it. The shallow and rushy discussion at the end with all the caveats clobbered together without any linkage is disheartened. Finally, there is an appendix that contains more description/methods/results which although related to the paper, do not link well to the text/results - there is only a single reference out of context. The author does not explore the aspects of the appendix results and the main research topic - it appears as just a dump of information. I would remove or overhaul the text to point to the results presented in the appendix.

- **Thank for the detailed review. We have worked to clarify the abstract, describe the data sets and ECCO, better streamline the caveats, and link the Appendix with the main text. Specific edits based on your comments are listed below. We hope you find the manuscript to be improved.**

**Specific comments**

- Below you will find some informal notes/suggestions with my personal opinion that I hope will help the author to be more clearer and better understand more points of view of the work. These notes were made mostly in chronological order, so some aspects are discussed in a rolled/back forth fashion. Although some are a matter of opinion, I'm certain that at least some of them can help improve the paper. 3-4: "is not sufficient to constraint Kp". I would disagree - if the Kp is being analysed/is part of the state vector/changes at analysis time, it is being constrained. This is using a lax formalism of "constraint" - which doesn't bode well with an assimilation subject paper.

- **We should point out that there is a difference between the sequential data assimilation systems you are thinking of and the parameter and state estimation framework we're using in the main portion of the text here. We separate out the two different types of observationally constrained modeling systems into the main text and Appendix to be clear about this. The parameter and state estimation framework we use (ECCO) uses observational constraints to estimate parameters and initial conditions. The "analysis time" is the entire length of the model simulation, which should be contrasted with the ten-daily analysis time of the GMAO S2S data assimilation system we use in the Appendix. We are only suggesting that oxygen provides information about $\kappa_\rho$ here because we're comparing adjoint sensitivities and not actually "assimilating" the oxygen. While oxygen is included in the cost function in ECCO, this is not the same as**

estimating the parameters and initial conditions with an optimization run with ECCO, nor is this the same as assimilating oxygen with a sequential data assimilation system to calculate analysis increments. The adjoint sensitivity approach we use in our manuscript can rule out whether a data set, like oxygen concentrations from WOA13, provides information about a particular parameter, like $\kappa_\rho$, but our main result is that we cannot rule out this possibility. We cannot make a stronger conclusion, though, because we have not performed the parameter and state estimate optimization with all observational data sets included in the cost function. This is why we have the language we used in the abstract. We have attempted to clarify that these three types of simulations are very distinct and each can provide information, but for different applications.

- 5-10: I would rewrite this.

- Saying "assimilated" in any ECCO-related context should be changed to "included in the cost function" in our manuscript. This is because the parameter and state estimation framework is not a data assimilation framework in the sense most people think about it. The adjoint inverts for parameters and initial conditions using observational constraints instead of adjusting its state using increments (as a sequential data assimilation system would). You could think about ECCO at a long-term 4D-VAR framework but the way we are using ECCO in this study is as a framework allows us to assess the similarities between sensitivities of the cost function to a model parameter. We have rewritten most of the abstract and hope that this resolves at least some of the confusion. We say, "... we show that the inclusion of misfits to observed physical variables–such as in situ temperature, salinity, and pressure–currently accounted for in ECCO is not sufficient to constrain $\kappa_\rho$, as $\kappa_\rho$ from ECCO does not agree closely with any observationally-derived product."

- 12: What about Kp misfits?

- **We compare simulations with Kp misfits to simulations with oxygen misfits in our manuscript, which we now clarify this with our rewrite of the abstract now. We say, "[w]ith the goal of improving the representation of $\kappa_\rho$ in ECCO, we investigate whether adjustments in $\kappa_\rho$ due to inclusion of misfits to a tracer–dissolved oxygen concentrations from an annual climatology–would be similar to those due to inclusion of misfits to observationally-derived $\kappa_\rho$ products."**

- 14-15: Why? Describe why - including in the results (see notes below).

- **In the abstract we now explain why with our last sentence, "... we show that adjoint sensitivities of dissolved oxygen concentration misfits to the state estimate's control space typically direct $\kappa_\rho$ to improve relative to the observationally-derived and microstructure-inferred values."**

- 54-60: Here is a good opportunity to introduce how ecco state estimate works differently from others and if the configuration you use is dynamical consistent instead of what is being said.

- We have added the following sentences, "These control variables can be iteratively improved by running the model in forward and backward– its "adjoint"–modes, which enables the calculation of gradients in the cost function. Each of these runs maintains dynamical and kinematical consistency because, in contrast to sequential data assimilation systems (see the Appendix for an example), the only variables that get adjusted are the control variables, not the dynamically and kinematically active variables."

- 69-72: I don't understand this focus here. Data assimilation could account for structural and measurement errors - you just need to adapt the observational error in the equation, inflate/deflate errors according to the product.

- It's true that this is one way to resolve the problem with the combination of both measurement and structural errors. However, the equation you would adapt the observational error into in this case requires vertical resolution finer than the model's grid. We mention in the conclusions a potential way to assess the structural errors: compare CTD-derived $\kappa_\rho$ using the finescale turbulence parameterization with microstructure-inferred $\kappa_\rho$ where those measurements are colocated. These $\kappa_\rho$ agree, on average, to within 2%, but there are spatial disagreements, depending upon the vertical bin sizes chosen, and the fidelity of the CTD data is currently unknown. The structural errors associated with the finescale turbulence parameterization's $\kappa_\rho$ would only be a guess right now.

- 73-74: Humm... these lax definitions are unnecessary if you define it more formally

above.

- **The adjoint is just the model being run backwards. We define what the adjoint is a bit earlier in the text now, but its formal definition is just the model code being run in reverse.**

- 80:85: I wonder if this distinction is actually necessary here and the paper in general (in sensitivity terms). For example, you say you use different kp obs sources in the Kp experiment, but you assign a constant error for both (apparently).

- **The distinction is necessary for the reader to be reminded of because the errors assigned to the inferred/derived products of $\kappa_\rho$ are approximate, whereas the measurement errors for quantities like dissolved oxygen concentrations are more well-known. Further, the derived $\kappa_\rho$ are based on Argo and CTD measurements that are used in a theory that requires spectral calculations, whereas the inferred $\kappa_\rho$ are based on microstructure measurements that are used in a simple relationship. We wanted to distinguish between "inferred" and "derived" because the microstructure-inferred $\kappa_\rho$ are considered the gold standard that (virtually?) every oceanographer trusts. We compare the other observationally-derived $\kappa_\rho$ with the microstructure-inferred $\kappa_\rho$ for this reason.**

- 83: which method?

- **We were referring to the finescale turbulence parameterization here. We have clarified this in the text.**

- 85-115: I understand you want to distinguish the observations and how they are derived, but for the assimilation what is improtant is the error you assign to the respective obs set. The nomenclature of Kp is a bit jarring too. Kp from a free-run (missing), the Kp from the state estimate at the start (missing), Kp after iteration 59 ( I assumed here and thereafter $Kp_{ECCO}$), $Kp_{micro}$, $Kp_{w15}$, $Kp_{K17}$.

- **The $\kappa_\rho$ from a free-run (E-CTRL) is the $\kappa_\rho$ after iteration 59. ECCO estimates $\kappa_\rho$ over the 59 iterations and that $\kappa_\rho$ is used for the re-runs. $\kappa_{\rho,ECCO}$ always refers to $\kappa_\rho$ after iteration 59 in the text. $\kappa_\rho$ from the first iteration is also discussed in the text, though, so we have let $\kappa_{\rho,ECCO,0}$ be this $\kappa_\rho$.**

- 95/110-115: I miss a figure here showing these observations estimates in a simple way - you got 3 different estimates, 1 gridded, 2 scattered. This would be a figure.

- **We now include a figure that shows the gridded initial conditions of $O_2$ for the ECCO simulations, the difference between the initial conditions and the WOA13 product where observations were taken (depth-averaged–as in the previous draft of the manuscript), and a pointwise scatterplot between the initial conditions and the WOA13 product where observations were taken.**

- Better yet if shows the actual Kp from ECCO which is absent and we know nothing about the spatial variability. There is not a good picture of obs coverage in the paper.

- **We show the Argo-derived $\kappa_\rho$ and how this differs from the ECCO-**

estimated $\kappa_\rho$. The maps that show how these two differ also have the locations where microstructure observations were taken. It is important to show the Argo-derived $\kappa_\rho$ because this shows the coverage. We show the profiles of $\kappa_\rho$ from ECCO over 16 different example microstructure campaigns and averaged over all microstructure campaigns. We now also show the *de Lavergne et al.* (2020) $\kappa_\rho$ and how this differs from the ECCO-estimated $\kappa_\rho$. So we show the observational coverage, but we haven't explicitly shown the $\kappa_\rho$ from ECCO in a map (only sampled profiles and maps of differences). We now include a figure of $\kappa_\rho$ from ECCO averaged over all depths below the mixed layer (because if the mixed layer is included, then $\kappa_\rho$ wouldn't be representative of values anywhere but inside the mixed layer).

- 123: Describe what the Oxygen from WOA is, units, coverage, mean/standard deviation figure? Limitations? What about the N2? Seems to me that observations here are treated like the holy grail but this is barely the truth when fitting and assigning errors.

- The oxygen concentrations from WOA13 are in ml/l, its coverage is essentially shown in the figure (areas that are non-white) we have now altered to include the initial conditions of oxygen in ECCO and a scatterplot, and the ranges of point-wise values are shown in the scatterplot we now include. We use the annual climatology because $\kappa_\rho$ in ECCO is not time-varying. If $\kappa_\rho$ in ECCO were time-varying, then we would need to account for temporal variations in oxygen concentrations to evaluate the information they provide about $\kappa_\rho$. N2 is a potential source of information as well, but this is tricky to compare with the model's N2 because of its vertical resolution compared to a typical observation's vertical resolution. Also, because N2 is determined from temperature, salinity, and pressure, that information may already be constraining $\kappa_\rho$ in ECCO. These are the primary reasons why we didn't perform experiments with N2. We now include this sentence in the main text: "Due to the relatively coarse vertical resolution of ECCO compared with observations and the likelihood that information from $N^2$ is already provided by temperature, salinity, and pressure, we do not directly compare $N^2$ from ECCO with $N^2$ from observations in another adjoint sensitivity experiment."

- 125-130: Why the picture if you are outsourcing the most important thing to the Whalen 2015 reference? Just include the picture here for the sake of your readers. Also, This "justification" is not accompained by a discussion of the methods/results. You need to explain in more detail what the insight here is and not outsource.

- I'm not sure if I understand this comment. The purpose of this figure is to show that there are similarities in the spatial distribution of vertical gradients in oxygen with the spatial distribution of $\kappa_{\rho,W15}$. The correlation between the two products isn't strong, but a visual comparison of the two products suggests this is because the similarities in their spatial distributions is approximate. If there is non-local information in the oxygen data about $\kappa_\rho$, then we could potentially see it through the adjoint sensitivity experiments we later perform in our study. This is supposed

to help motivate our approach. We added the following sentence: "Any potential information that oxygen concentrations can provide about $\kappa_\rho$ is likely through oxygen's vertical gradients because diapycnal mixing acts to erode water masses–which tend to be relatively homogenous in oxygen concentrations–along their peripheries." We also added the following sentence at the end of the ensuing paragraph: "Because of the possibly non-local relationship between $\partial O_2/\partial z$ and $\kappa_\rho$, we perform model experiments to further explore the potential information that oxygen concentrations provide about $\kappa_\rho$."

- Figure 1. The justification at 125-130 is shallow and a bit out of place here that I wonder why the do/dz is the first figure in the paper. The first figure is an important milestone and the reason why this has to be do/dz is not fully commented - I don't think there is a backreference to this figure at all.

- **We decided to add the *Whalen et al.* (2015) dissipation rates in this figure next to $\partial O_2/\partial z$ to show the spatial co-location of magnitudes more clearly. We also rewrote this entire subsection and now back-reference this figure later in the manuscript. This figure is one motivation (in addition to the arguments we make) for doing our simulations because the simulations can reveal whether oxygen concentrations provides information about $\kappa_\rho$ more clearly and convincingly than spatial correlations can.**

- 135-140: Humm objectives in methods? I think you could say that in the introduction and just explain what the model/assimiation is here.

- We have move the two objectives to the last paragraph in the introduction.

- 141: time-invariant but spatially varying Kp field? Show it - after all that is what you are trying to improve. Figure, figure, figure.

- $\kappa_\rho$ is constant in time because ECCO is already solving an under-determined problem with all of the parameters and initial conditions it's estimating. We show in the Appendix that parameterizations in free-running models like KPP suggest it should be significantly time-varying in the subpolar North Atlantic in particular, but hardly anywhere else. We now include a figure of the depth-averaged (below the average mixed layer) of $\kappa_{\rho,ECCO}$.

- 144: IMO table 2 provide enough details for the reader to clearly identify what the simulation is all about.

- We supplement this with a description in the main text to be certain that it's clear because other reviewers did not find Table 2 to suffice.

- 150: What are the 14-day adjustments? this is the state estimate, right!? if so, say it here. All in all, Less model description, more framework description.

- The 14-day adjustments are parameter estimates that adjust the reanalysis forcing fields we began with. We clarified this issue with another reviewer as follows: the adjoint averages adjustments to the atmospheric forcing fields, which are re-estimated and then applied over 14-day periods. We have rephrased how we state this sentence: "Average adjustments to the wind stress, wind speed, specific humidity, shortwave

downwelling radiation, and surface air temperature are re-estimated and then applied over 14-day periods."

- 155-160: I miss a quick discussion about the Kp and your parametrisations. Why it is time-independent, caveats with the other parametrisations influences, and possible impacts and results (e.g. mixed-layer/convective adjustment and the oxygen results).

- **We have edited the end of this paragraph to say the following: "Vertical mixing–diapycnal plus the vertical component of the along-isopycnal tensor–is determined according to the *Gaspar et al.* (1990) mixed layer turbulence closure, simple convective adjustment, and estimated background $\kappa_\rho$. Here, $\kappa_\rho$ represents a combination of processes, including–but potentially not limited to–internal wave-induced mixing. $\kappa_\rho$, the Redi coefficient, and the Gent-McWilliams coefficient are time-independent because of the under-determined problem of inverting for initial conditions and model parameters would be even more under-determined if they were allowed to vary in time–explained below."**

- 170: So ECCO is fitting Kp and others already against other state variables. This should be at a table or in a more accessible location.

- **The list of variables has been transformed into a table now. We include columns that indicate whether they're initial conditions or parameters, time-varying or time-independent, and two-dimensional or three-dimensional. Note that there are some variables that are time-independent**

and three-dimensional (only spatially-varying over each wet point) and other variables that are time-varying and three-dimensional (spatiotemporally-varying only over the surface).

- 173: So you already start from an Optimized Kp - what about you starting from the 0 iteration with the data presented here? Also, what about the averaging done? The inner-loop is a whole year? Too so many gaps in information.

- **I had to request the $\kappa_\rho$ field from the first iteration. One of my co-authors had this field available because he works with the people who did the optimization of ECCO. The $\kappa_\rho$ field from the final iteration is more publicly available. The $\kappa_\rho$ field is time-invariant so there's no temporal averaging.**

- 175-180: weak appendix link - you need to provide more material along the paper to entice the reader to read the appendix.

- **We have included a reference to the first figure in the Appendix to justify why comparisons of $\kappa_\rho$ from ECCO with $\kappa_\rho$ from observations at particular times (e.g., from microstructure) and locations (in the subpolar North Atlantic) may not be appropriate. $\kappa_\rho$ is likely to vary in time by about an order of magnitude over a year's time in the subpolar North Atlantic. Nowhere else does $\kappa_\rho$ vary much, though, according to the model we use in the Appendix. The more relevant link with the Appendix in this manuscript is that sequential data assimilation systems have different issues with $\kappa_\rho$. These data assimilation systems can distort dynamical**

tracer fields because of their application of analysis increments, result-ing in a violation of conservation principles. This potentially causes the model to undergo baroclinic adjustment, which can induce spurious ver-tical velocities and mixing. This isn't a problem in ECCO because only non-dynamically active fields are adjusted as the model is run forward and backward, and then the model is run over its entire time period length. In our adjoint sensitivity experiments, there is not adjustment in the control variables–only gradients are computed to inform how an optimization could be improved when we include new information in the cost function.

- I don't understand the -rerun here - you don't create a symbol for that and I can't see a reference anywhere. you are being repetitive here since you explain better in 190-195.

- **The re-run is E-CTRL. The adjoint sensitivity experiments are $\mathbf{E}_\kappa$, $\mathbf{E}_{O_2}$, and $\mathbf{E}_\epsilon$. We are more explicit about this in the bulletpoints, as you point out, but need to be clear that there are three types of simulations that can be done with ECCO in the application we're focusing on in our study: re-runs, adjoint sensitivity experiments, and optimization runs. We do not perform optimization runs here; the purpose of our study is to examine a potential motivation to do those more expensive simulations with observationally-derived $\kappa_\rho$ or oxygen concentrations.**

- 185-190: out of place - better around 150.

- **Yes, we agree. We moved this passage.**

- 195-200: can you please define more formally what a forward ECOOv4 simulation is? a free run?

- **Yes, the re-run is a free run, except it uses parameters that were estimated from the optimization. So the initial conditions and parameters (e.g., $\kappa_\rho$ and surface forcing fields) are inputs for the free run. We have added the following phrase to the text: "sometimes referred to as an ocean-only free run".**

- No explanation previously of using N2 from WoA - I assume this is why you try to justify the do/dz presented earlier right? This is badly connected.

- **When we first examined the vertical gradients in relation to dissipation rates and $\kappa_\rho$ from Argo floats, we now say: "The spatial correlation between $\partial O_2/\partial z$ and $\kappa_{\rho,W15}$ is smaller in magnitude–about $-0.1$–which motivates further consideration of the information provided by $N^2$–derived from World Ocean Atlas (2013) temperature and salinity data with the TEOS-10 package (*MacDougall and Barker*, 2011)–later in this study."**

- $Kp_{ECCO}$ is 10-5 after optimisations or before? You said it was spatially vrying! $Kp_{ECCO}$ is before or after state estimate!? is $Kp_{ECCO}$ from E-CTRL? You are not making the reader life's easier without naming simulations and parameters properly.

- **$\kappa_\rho$ is initially set to $10^{-5}$ m$^2$ s$^{-1}$ before optimization. We show what the first iteration's estimate of $\kappa_\rho$ ($\kappa_{\rho,ECCO,0}$) and final iteration's estimate of $\kappa_\rho$ ($\kappa_{\rho,ECCO}$) are in comparison to microstructure. We also include a**

**figure with a map of depth-averaged $\kappa_\rho$ below the mixed layer from the final iteration. Each of these show that $\kappa_{\rho,ECCO}$ is not $10^{-5}$ m$^2$ s$^{-1}$ after optimization.**

- 210-215: Explain better why the results are independent of the run length. Is that just because you are using data in a climatological mode (averaging everything into a clm year!?). If you adjust fluxes and use observations, how the length doesn't count if by varying length you vary the amount of parameters to fit and such the Cost function?

- **The results are not necessarily independent of the run length, but when we ran a longer simulation, we found similar results. This is likely due to our use of climatological fields for observations. We have clarified this by editing a sentence to say: "The adjoint sensitivities from $E_\kappa$ are not as sensitive to the run length as they are to the initial conditions of the run due to the lack of time-dependence of the observations included in the misfits–$\kappa_\rho$ and oxygen concentrations."**

- Your Eo starts from a different initial condition? Results in the methods? Sorry but this part is a bit of a mess - I think you should explain the sensitivity analysis and the assimilation before this part because this raises all types of questions (how the cost function is, background covar, B/R/Gain matrices, etc).

- **Our simulations each begin from the same initial conditions. The only results we're presenting that had different initial conditions are the $\kappa_{\rho,ECCO,0}$ profiles because the initial conditions estimated by ECCO after the first**

**iteration are different from the initial conditions estimated by ECCO after the final iteration. We now say: "We take the ECCOv4r3 solution as initial conditions for each of our simulations. We perform an adjoint calculation in each experiment, except for E-CTRL."**

- Figure2. This is a bit of out place and the reason why is not clear. The picture could show much more, such as the standard deviation of both model and obs. The colorbar got strange fonts, was this just pasted on the side. I would flick the centre of the picture (in fact all of them) to 180E picture to the pacific - since this is where errors are larger and where more data points are present (landmasses are distracting here). What is the depth of averaging? I would at least expect that you follow the 250-500/500-1000/1000-2000 you used in other figures so to give us something to reference regarding your results later.

- **We now present a point-wise scatterplot to show the general agreement between the initial conditions for oxygen concentrations in ECCO and the oxygen concentrations from the World Ocean Atlas (2013). This includes every depth available in each data product. This should give the reader an idea of the standard deviations within each data product as well as the standard deviations of their disagreement. We also show the initial conditions for oxygen concentrations in ECCO for reference.**

- 225-230: I found the description here too simplistic and miss something more formal. This session falls apart without the apriori information. How the background error covariance is computed? Is it independent (apparently yes)!? What is the decorrelation length scale used? Several apriori facts are important here and the

lack of these details are very concerning. How the setup and the equations are optimized/solved is nowhere to be seen. This is the time to describe more fully how

things work - so far all we got were sprinkles of incomplete infromation. Use a clear

equation with the full state vector in each experiment. Is the 4dvar inner-loop a

whole year?

- **The ECCO framework do not use 4DVar inner-loops because this miti-
gates the problem of non-linearity over long time scales; this is unique to
the ECCO framework. But it sounds like you're referring to a sequential
data assimilation system. We use the default decorrelation length scales
for the smooth package in ECCO with the *Weaver and Courtier* (2001)
method. This is explained in *Forget et al.* (2015). The smoother is ap-
plied to 1 grid cell in each direction, which means that the decorrelation
length scales are $3 * 100/e \sim 100$ km. Computing a background error co-
variance offline isn't necessary for our runs. The a-priori information in
the model are climatologies that are documented in the ECCO literature.
We are not excluding any information that is necessary to performing the
adjoint sensitivity runs here.**

- 235: That 2% is optimistic. We need more details on how you declare the obs error.
I think a lack of detail here is jarring and gives a bad indication/lack of attention
on how the assimilation was setup. What is being corrected? Why you are not
correcting Kp jointly with other observations? Just so your equations/W/J are
simpler? If you don't use other state paremters in the equation, you are not using
the full potential of the system (fitting the errors with T/S/SSH+Kp). I'm puzzled

and can't see how these experimenets are being conducted.

- The 2% comes from measurement errors for instruments commonly used to detect oxygen concentrations, which may be optimistic if we use an interpolated observational product, but we are only including oxygen concentration observations where they were collected (at the nearest model grid point). Again, there is no "assimilation". It's just including these observations in a misfit to compute its gradients with the model's forward and adjoint matrices. We are considering only directions in the control space for how to improve $\kappa_\rho$, given the optimization of the other variables. Of course the other control variables will be affected by the inclusion of a new variable like oxygen concentrations, but we did adjoint sensitivity experiments for these (e.g., Redi coefficients) and could not find more than about 50% agreement (random chance) between the adjoint sensitivities in an analogous pair of experiments to the ones we perform for $\kappa_\rho$. This could be due to a lack of fidelity of the *Cole et al.* (2015) Redi coefficient product or the fact that it's more strongly time-varying than $\kappa_\rho$ (below the mixed layer). In any case, we are examining whether there's motivation for an optimization run, not performing an optimization run ourselves, so the only variable that's influencing the control space solution is the new observational product. We have added the following sentence to the text: "We consider evaluating directions in the control space in which to improve $\kappa_\rho$, given the values of the other control variables from the model's optimization."

- 238: concerning Kp (and others).

- **Thanks.**

- 240-245: What!? So you use model error/obs error as "sensitivity" for the Kp experiment? Explain why you decided to do this. I don't think you can use this equation - W is the solution - and you are imposing it? so you are just looking at the sign of "forecast error" here and scaling it by the obs error!? Maybe using better wording or explaining better would remove the guesswork.

- **We don't solve for W, but impose it based on the approximate measurement errors. We are trying to assess whether an observationally-derived quantity with a large uncertainty agrees (in direction of the control space) with an observation of a quantity with a small uncertainty. This is the simplest way to do it. The measurement uncertainties of other variables are used to determine their W so there's nothing different about this approach in ECCO. There is no "forecast" here.**

- 245-250: "Short of assimilating ... we assess whether the assimilation of a particular dataset *could lead* to a more ..." - !!! - This should be The first line in 225 - you are just confusing the reader - all of that text to say that you are just looking at the innovations and not performing it (apart from the Oxygen experiment I guess). I'm puzzled here about what is being done - you need to clarify the whole section 2. What you are optimizing here - W is usually the solution to the problem but you are imposing it now?

- **We have moved this sentence to the beginning of the subsection. We are**

not optimizing anything with our simulations. We are simply comparing how $\kappa_\rho$ is being directed to change with new observational information included in the model's cost function, which could motivate a new optimization run with one of the observational data products included.

- 240-245: All that worry about Kp from different observations having different origins to just set the values like this? Also, the uncertainty here is related to the model to obs, not to obs to obs.

- **In the ECCO framework, the uncertainty inversely scales with the weight for each observation and the uncertainty corresponds to observational uncertainty.**

- This part doesn't bode well for a robust setup in the sensitivity task. Also, there is no discussion about these settings and the sensitivity impact - you just let it for later I assume? (But this never happens down the results...)

- **We discuss this in the results, where we mention the Monte Carlo simulations. We allow for uncertainties in the observed quantity and in the weights when we sample values of the adjoint sensitivities to get correlations between the sensitivities from different experiments. We now have a paragraph in this section that discusses this.**

- 245-250: "Because the observations of Kp are not direct measurements...": Again nothing is a direct measurement (maybe Temperature is the closest thing), so this is not the reason to seek how the model Kp differ from observations...You just need to understand how the model errors are distributed in space/time. You just need

to know how (y-Sxtilde) looks like - just said it.

- **We have changed the language here to no longer refer to *in-situ* observations as "directly measured" quantities.**

- 250-255: "However we dont want to assimilate ... because of their uncertainties and still limited spatial coverage relatively to oxygen". Why not? - because your constraint is only to Kp - you will be overfitting? Or because the equations you are using are not up to the Kp statistical log distribution? Why not try to solve the problem by assimilating Kp with all the other ECCOv34 parameters? This phrase here is probably locked in with the methods you are using so better to describe these insights with good information. Again some important concepts and insights are not being fully described here. I'm surprised by how this entire section confuses the reader.

- **We are describing the analyses that need to be performed to determine whether it's a good idea to include observationally-derived $\kappa_\rho$ products in the model's misfits and if it's unclear, then whether it's a good idea to include oxygen concentrations in the model's misfits. One of our conclusions is that either the uncertainties will be too large to place any constraint on the model's $\kappa_\rho$ or the resulting model-estimated $\kappa_\rho$ won't be any more accurate than the observationally-derived products. It has been implemented in the model to use the log-transformed $\kappa_\rho$ because of their distribution so that isn't a concern. We have changed the wording of this to: "We devote the first portion of our study to determining whether $|\kappa_{\rho,W15} - \kappa_{\rho,micro}| < |\kappa_{\rho,ECCO} - \kappa_{\rho,micro}|$ (and, by extension, $\kappa_{\rho,K17}$**

in place of $\kappa_{\rho,W15}$) is true. We do this because $\kappa_{\rho,micro}$ is limited in its spatial coverage compared to $\kappa_{\rho,W15}$, $\kappa_{\rho,K17}$, and $\kappa_{\rho,deL20}$. Also, $\kappa_{\rho,W15}$ and $\kappa_{\rho,K17}$ are still limited spatial coverage relative to dissolved oxygen concentrations. While $\kappa_{\rho,deL20}$ has global spatial coverage, its measurement plus structural uncertainties are not well-known compared to dissolved oxygen concentrations..."

- 256: I would assimilate both since they probably provide both information - but I'm not sure because you don't say what you are fitting here (boundary conditions? initial conditions? fluxes? model parametrizations?

- One of our suggestions in the conclusions is that an observationally-derived $\kappa_\rho$ should be used for $\kappa_{\rho,ECCO}$ instead of including it in the model's misfits. This would reduce the number of parameters that need to be estimated in an under-determined estimation procedure and these observationally-derived products are closer to $\kappa_{\rho,micro}$ than $\kappa_{\rho,ECCO}$. It would still be possible to allow for some adjustments in $\kappa_\rho$ if the observationally-derived products were assumed as a first guess within some uncertainty. Then oxygen concentrations could potentially help constrain $\kappa_\rho$, if wanted.

- 260-265: "is more than a factor of 3/ above". How this choice relates to the specified errors in the fitting? No explanation and insight provided - seems a match-fixing kind of a problem when reading at first.

- There are reasons to do this. We need to be clear about this. Disagreements between the model and observational products in $\kappa_\rho$ that are

greater than a factor of 3 are outside the approximate uncertainty of the observational products; i.e., the differences are statistically significant. Also, regions where disagreements between the model's initial conditions and observations in oxygen concentrations that are within 2% are statistically insignificant. We are interested in where these both occur because these are regions where changes in $\kappa_\rho$ are needed and the errors in oxygen are due to errors in the physics (e.g., $\kappa_\rho$), not initial conditions. We added this sentence: "We are interested in regions where $\kappa_\rho$ is significantly erroneous and where the errors in oxygen are due to errors in the physics (e.g., $\kappa_\rho$), not initial conditions; hence, these choices."

- 265- 270: I'm quite surprised by the lack of a discussion on how fitting for Kp will improve the model run since this is a parametrisation - the impact/practicalities and impact on dynamics are not discussed at all. The author refers that Kp is fixed in ECCO, but how is the model/analysis will perform after Kp is improved jointly or alone is left to the imagination. I understand now that the author is not looking for the analysis but just for the impact, but given the exoteric parameters, a mention of how this will flow down in the model run is important.

- **Our paragraph regarding the motivation for using oxygen contains some of this discussion, but we now include some discussion of how an improved $\kappa_\rho$ will impact the state estimate. We added the following: "If $\kappa_{\rho,ECCO,0}$ is in closer agreement with $\kappa_\rho$ from observational products than $\kappa_{\rho,ECCO}$, then errors in $\kappa_{\rho,ECCO}$ are likely being compensated by errors in other control variables beyond the first iteration of the model's optimization**

run. **Inaccuracies in control variables can make physical inference using ECCO less grounded in reality and could make the state estimate itself less accurate–e.g., errors in $\kappa_\rho$ will influence vertical tracer transport and mixed layer depths."**

- 270-280: More methods in results? "A geometric average is taken ...". New information about kp "log-nomal". these need to be properly defined beforehand.

- **We moved the methods-related sentence to the end of the paragraph describing the microstructure data in the methods section: "A geometric average is taken for each profile because this is more representative than an arithmetic average for a small sample size and when the data are not normally distributed (*Manikandan*, 2011), like the log-normal distribution of $\kappa_\rho$ (*Whalen*, 2021)."**

- 284: not shown here? Really? The first result of the paper that can show some guess of spatial variability of Kp and you skip it!?

- **We now show the spatial standard deviation of the $\kappa_\rho$ profiles averaged over microstructure locations. We also show comparisons of $\kappa_{\rho,ECCO,0}$, $\kappa_{\rho,ECCO}$, and $\kappa_{\rho,micro}$ for 16 example campaigns. As mentioned earlier, we additionally show spatial variability of depth-averaged $\kappa_{\rho,ECCO}$ in a new figure. This is all in addition to the spatial variability shown in $\kappa_{\rho,W15}$ and how it compares with $\kappa_{\rho,ECCO}$ and likewise for a new figure with $\kappa_{\rho,deL20}$.**

- 285: IMO, showing $Kp_{ecco}$ vs Kp, micro (Fig3) is better done after $Kp_{ecco}$ KP, W15. This is so because you show in Fig4 the locations of the microstructure witch avoid

the user to be distracted by Fig4 when reading Fig3.

- **I had this order at one point, but other people who have seen this manuscript (including my co-authors) thought the order you saw made the most sense because the first result shown in this section should be a comparison with the gold standard measurement for $\kappa_\rho$ and justification for trusting the *Whalen et al.* (2015) product (shown with magneta X's). However, now that we have a new, depth-averaged $\kappa_{\rho,ECCO}$ figure, we have moved the microstructure locations to this figure.**

- Figure4. The right figures are misleading in regards to the other figures in the paper since white is not where data is missing, but where $kp_{ecco} \sim= kp_{argo}$! I would recommend adding dot points where this situation was found to distinguish from the lack of data.

- **Per another reviewer's comments, we have changed the colorbar to show a non-white color where the disagreements are small.**

- $kp_{argo}$ is better than $kp_{w15}$ reference, just as well $kp_{ctd}$ is better than $kp_{k17}$. Finally, I would help the reader here and say "that red(blue) areas are where argo is smaller (higher) than ECCO" since the log10 ratio.

- **We have added the sentence, "Red (Blue) areas in Figs. 6b,d,f indicate locations where Argo-derived $\kappa_{\rho,W15}$ is smaller (larger) than $\kappa_{\rho,ECCO}$." We have also changed the notation for the different $\kappa_\rho$ products derived from observations. Because we also include the *de Lavergne et al.* (2020) product, we refer to this product as $\kappa_{\rho,tides}$.**

- 295-300: I don't agree with how you define/use constraint here. Lack of agreement is not a lack of constraint. A realistic constraint is what? perfect match? the same order of magnitude? Define what you consider a good constraint or not. You can't in the paper because no one knows what is the initial Kp at all and how it was improved from the base case.

- **We added the following text to the introduction: "Here, by 'constrain,' we were referring to using new data to change the level of agreement between the model and an observational product–not necessarily to achieve a perfect match."**

- 300-305: This is just blaming without a proper discussion which should have happened beforehand to explain the limitations of the state estimate. Although I agree that compressing all the information required in a short text is a challenge, this ending appears a shoot in the foot - you can't give the reader proper insights of why kp is not better constrained. You forgot to mention the numeric nature of the parametrisation and its limitations (plus the other parameters). Wouldn't be a leakage towards fixing the other parameters instead of kp? I miss some discussion around that.

- **Per other reviewers' comments, we have added a discussion of how information from each observed quantity that is included in the misfit can change most–if not all–parameters. Oxygen concentrations would not be an exception. We focus on $\kappa_\rho$ here because there appears to be some correlation between the sensitivities of the misfits in oxygen and those in $\kappa_\rho$. We investigated the sensitivities for another parameter–**

the Redi coefficient–and found virtually no correlation between the two types of simulations–50/50 agreement in sign. This doesn't rule out the possibility that the Redi coefficients–or other model parameters–would compensate for some errors in the way that $\kappa_\rho$ would change, though. Including information from more data sources could make this type of error compensation less prominent, though. We do not explicitly address this issue in our manuscript because we are are not running optimization runs to examine this possibility. We are simply motivating the possibility of doing optimization runs that include oxygen concentrations.

- 310-315: "Because $kp_{ecco}$ tends to be very large inside mixed layers" - another sprinkle of missing information in the middle. Couldn't you provide your readers with how $kp_{ecco}$ is beforehand?

- We have added a figure that shows $\kappa_{\rho,ECCO}$ depth-averaged below the average model mixed layer. We exclude the mixed layer because $\kappa_\rho$ effectively doesn't do anything within a bulk mixed layer–it already being homogeneous in density. So $\kappa_\rho$ can be very large and the mixed layer will still be homogeneous in density.

- 311-312: Another of saying it is "A positive adjoint sensitivity implies that the model overestimated Kp". But isn't the kp less than an order of magnitude compared to micro in Fig3? Globally, the tendency of your "adjoint" here is to increase Kp (more red than blue in Fig4), which is akin to what your GMAO results (appendix) are doing and overshooting the Kp beyond the microstructure. There is no discussion around this.

- A positive adjoint sensitivity implies that the model will reduce $\kappa_\rho$ because the objective is to reduce the misfit ($dJ < 0$) and the model determined that $d\kappa_\rho < 0$. One takeaway from the combination of the microstructure-inferred and Argo-derived $\kappa_\rho$ comparisons with $\kappa_{\rho,ECCO}$ is that $\kappa_{\rho,ECCO}$ is too small compared with $\kappa_{\rho,micro}$ and often too large compared with $\kappa_{\rho,Argo}$. This is because of the regions where microstructure measurements were taken tend to be in regions where $\kappa_{\rho,ECCO}$ is too small compared with $\kappa_{\rho,Argo}$. So the microstructure samples are not very representative of the entire ocean. We edited the last sentence of this paragraph to say: "Microstructure measurements tend to be regions where there are prominent topographic features and where the centers of subtropical gyres are found, which–judging from the predominant signs of disagreement in Figs. 4-5 versus Figs. 6b,d,f–aren't representative of the ocean where Argo measurements were taken."

- 310-320: Why not a figure with adjoint sign profiles similar to Fig3? This would be more helpful than the whole description by region.

- The new figure we have with 16 example microstructure campaigns compared with $\kappa_{\rho,ECCO,0}$ and $\kappa_{\rho,ECCO}$ should help with this. Where $\kappa_{\rho,ECCO}$ is too small (large), the adjoint sensitivities are negative (positive).

- Figure 5. Why not include a comparison of signals here? better than asking the reader to do that with these small figures and very gappy coverage. I'm also now presented that the calculations are done only for one year - 1992 - which is another surprise since this is not mentioned or discussed anywhere.

- **Actually, this was mentioned on lines 208-209 of the version you read. The table quantifying the volume of the ocean over which the signs of the adjoint sensitivities agree from each experiment and the following figure get across the essential points we're trying to make. The purpose of the figure you're referring to here is to present a visual comparison of how the adjoint sensitivities agree from each experiment.**

- 325-335: A global metric would be better here. Better metrics to compute %s would be beneficial in the paper since this is rather arbitrary. Maybe one or two regions of focus (one with a lot of obs - Kuroshio/North pacific and South Indian ocean?

- **We quantify this in a table–both globally and regionally.**

- I found a bit daunting all the %s without a laser focus on the process at hand and where it will impact the most. The figures definitely need to be centred shifted towards the pacific.

- **We expect that oxygen will provide information about $\kappa_\rho$ away from regions with large air-sea flux of oxygen and away from the intensified jets. The subtropical North Atlantic Ocean is one region that previous studies have used oxygen concentrations to determine water mass erosion rates and residence times, and we find that this ocean basin has the highest percent agreement in the signs of the adjoint sensitivities across each experiment. The west Pacific is very different from the east Pacific in the quantities we show so we're not sure if centering the figures on the Pacific would help. We added the following to the first paragraph of**

this subsection: "We expect that the signs of sensitivities agree most in regions away from where air-sea fluxes and transport of oxygen–e.g., by intensified jets–are large. One of these regions is the subtropical North Atlantic Ocean, away from the Gulf Stream Extension."

- 335-340: I think this deserves a new paragraph and more explanation since it is an important result.

- **We have added the following sentence to this paragraph: "Thus, the regions with the largest disagreements in oxygen concentrations can always decrease their oxygen misfits by changing $\kappa_{\rho,ECCO}$ with a sign consistent with decreasing its disagreement with observationally-derived $\kappa_\rho$."**

- 340-345: Asking readers to calculate the white regions in one figure that are not white in the other !? tip: Making the life of the reader easier is the best way to make them happy. This is the most problematic aspect of using Fig5/Fig6 together since the white parts are misleading. You already set the reader mind that white is missing data, and now some figures perturb this notion and are used to reason about the results. I would refit Fig5/Fig6 to match the text and what you want to say more directly.

- **We didn't ask the readers to calculate the amount of extra white area in one figure compared to another. We did the calculation ourselves and reported it. The rationale for the extra white area is as follows: we need to subset the data in order to control for the possibility that the bias in initial conditions is the reason for the errors in oxygen concentrations.**

We also are not interested in regions where $\kappa_{\rho,ECCO}$ is already consistent with observationally-derived estimates to within their approximate uncertainties. After we remove these possibilities that would confound our inferences, we are left with a little less than half of the comparable grid points.

- 350:355: I would also wrap this in another paragraph since the insight here is important and related to the next paragraph at $\sim$360.

- **We edited the first sentence of the paragraph near the line (we think) you're referring to: "Lastly, given that the general agreement in signs of sensitivities between $E_\kappa$ and $E_O$ are likely underpinned by physical reasons unrelated to stratification, we pursue whether there is a statistically significant relationship between the adjoint sensitivities from $\mathbf{E}_\kappa$ and $\mathbf{E}_O$."**

- 362: Finally a good insight, but not without trouble. You don't describe anywhere how N2 is distributed, statistics, or where it will likely dominate against Kp and how it is related to it. Also, how this would be fit together with Kp (and other parameters) is a missing point in the paper discussion.

- **We now describe how $N^2$ is distributed, but already wrote several times how $\kappa_\rho$ is related to $\epsilon$ through $N^2$–because of the *Osborn* (1980) relation. We have added the following sentence: "$N^2$ is generally about $10^{-7} - 10^{-5}$ s$^{-2}$, with lower values in high-latitude and deeper regions and higher values in the thermocline and in shallow water areas–which skew its global average (standard deviation) below the mixed layer to about $1.2 \times 10^{-4}$**

$(3 \times 10^{-3})$ s$^{-2}$."

- 365-375: This end is badly written and looks like a last-minute addition. I would rewrite it since it is the concluding remarks.

- **We have heavily revised this paragraph and split it into two.**

- 375-400:. I think it is too shallow here. There is no broad discussion of alternatives, cause/effects.

- **We have heavily revised this whole section to discuss alternative approaches– i.e., not improving the agreement between $\kappa_{\rho,ECCO}$ and observationally-derived products but just using observationally-derived products to reduce the number of parameters the model estimates–and how to approach a new optimization that makes use of informatino from oxygen concentrations.**

- 390: Again the uncertainties are not discussed in terms of model error/background covariances/ inflation or decorrelations.

- **Some of these issues are related to sequential data assimilation systems. The others we hope have clarified your concerns about uncertainties we did not discuss before.**

- 400-435: yes, yes, several things can affect but you don't discuss the real deal: fencing your results so people can locate themselves of what needs to be done next or how to relate this paper to their problem. IMO, at this stage, the reader is just tired of unlinked/big scoped caveats/problems instead of pin-pointed smaller scope

discussion.

- **We discuss the caveats, which are many-fold, but we also discuss what can be done to potentially improve the ECCO ocean state estimate, at least through improvement of $\kappa_\rho$ now.**

- 440-eof : I will leave it to other reviewers/editors to see if this is important to be kept in the paper. Certainly, there are not enough references in the text to this section although some results are interesting from the point of view of assimilation. Also, important references in the appendix are not mentioned in the text and even some discussions related to the results presented are better than in the paper itself. Puzzling to understand why the author didn't include some of that in the actual sections!

- **We now reference the Appendix in several locations in the main text. We didn't focus on the sequential data assimilation system results because of a computer crash where we lost much of the data and because we couldn't do justice to the cause of the large errors in $\kappa_{\rho,GMAO}$, as I was never granted access to running the GMAO S2S data assimilation system (only what I could figure out on my own with the GEOS-5 model).**

---

## Author Response (AR2)

Dear editor,

We are resubmitting the manuscript, "Tracer and observationally-derived constraints on diapycnal diffusivities in an ocean state estimate," to *Ocean Science*. We have changed some of the text to clarify how we separate the background diapycnal diffusivity (now denoted by $\kappa_{\rho,bg}$), Gaspar et al. (1990) scheme, and (3,3) entry of the along-isopycnal tensor. We also note that convective adjustment does not act through a diapycnal diffusivity in the MITgcm. We address each of the reviewer's points below.

**1 Reviewer #2 (Comments to Author (shown to authors):**

- In this round of revision, the authors added more materials and rewrote some of the text. Those efforts clearly improved the readability of this difficult paper. The authors also tried to address the few questions I had about the previous version of the paper. I appreciate their hard work and effort. However, I am confused about their reply to one of my major concerns that if the "ECCO diapycnal diffusivity" the authors used is actually comparable to the directly observed or inferred diapycnal diffusivity. I consider this a key issue that should be examined and clarified before I recommend acceptance.

- **Thanks for your additional feedback.**

- In their reply and new text, the authors added "Vertical mixing–diapycnal plus the vertical component of the along-isopycnal tensor–is determined according to the Gaspar et al. (1990) mixed layer turbulence closure, simple convective adjustment,

and estimated background $\kappa_\rho$ for internal wave-induced mixing." Does this mean the vertical component of the Redi parameter is affected by the other processes, like the convective adjustment? I don't think that is the case.

- **Each component of the along-isopycnal diffusivity tensor is time-invariant, as is the background diapycnal diffusivity. However, the Gaspar et al. (1990) mixed layer turbulence closure is not time-invariant. While simple convective adjustment is not time-invariant either, convective adjustment does not act through a diapycnal diffusivity in the MITgcm. Thus, the vertical diffusivity is affected by the (3,3) entry of the along-isopycnal tensor, but not the other way around (other than indirectly due to how the Gaspar et al. (1990) scheme affects the ocean state estimate). We have edited the above quote to read: "Vertical mixing is the sum of diapycnal mixing and the vertical component of the along-isopycnal tensor, where diapycnal mixing is determined according to the _Gaspar et al._ (1990) mixed layer turbulence closure and estimated $\kappa_{\rho,bg}$. Convective adjustment does not act through $\kappa_\rho$ in the MITgcm. Here, $\kappa_\rho$ represents a combination of processes, including–but potentially not limited to–internal wave-induced mixing. $\kappa_{\rho,bg}$, the Redi coefficient, and the Gent-McWilliams coefficient are time-independent..."**

- The authors also mentioned that ".. to transform this value into a diapycnal diffusivity that's equivalent to the observational product values, we need to subtract out the (3,3) entry of the along-isopycnal diffusivity tensor, which is not perpendicular to the isopycnal contours but parallel to them. After subtracting this from

the vertical diffusivity, we are left with the diapycnal diffusivity. As for what this coefficient physically represents, it is the adjusted background diffusivity via the adjoint estimation process." Unless I misunderstood these sentences, subtracting the vertical component of the redi parameter from the vertical diffusivity won't give us the background diapycnal diffusivity $\kappa_\rho$, but a term including background diapycnal diffusivity, parameterized mixing based on Gaspar et al. (1990) and convective adjustment.

- **This quote was from a previous version of the manuscript. The vertical diffusivity includes the (3,3) entry of the along-isopycnal tensor, the parameterized mixing based on Gaspar et al. (1990), and background diapycnal diffusivity. Thus, we need to subtract the (3,3) entry of the along-isopycnal tensor to get the full diapycnal diffusivity. We explain why we said that what's left physically represents the adjusted background diffusivity below, but you're right that we should rephrase the sentence in the quote, which is what we did with the version you just read.**

- My understanding is that in this study the authors just compared the directly available background diapycnal diffusivity $\kappa_\rho$ with the observed or inferred diapycnal diffusivity. I don't think they are directly comparable. A more sensible choice would be the full diapycnal diffusivity including the other parts rather than just the background diapycnal diffusivity. I understand that a lot of diffusivity terms used in this paper and it is possible that the confusion is simply a presentation issue. If that is the case, this should be easily addressed by rewriting a few sentences. If that

is not the case, this will be a critical issue and require a lot of extra work.

- We used the full diapycnal diffusivity but in the regions where we're comparing the simulated diapycnal diffusivities from observations, the full diapycnal diffusivity field is approximately the same as the background diapycnal diffusivity field. Microstructure-inferred diapycnal diffusivities include the full sum of process, but are not reliably measured near the boundaries. On the other hand, Argo/CTD-derived diapycnal diffusivities are only valid away from the surface because they represent the internal wave-related background mixing. The *de Lavergne et al.* (2020) product represents internal tide-related background mixing. Note that the regions with the highest level of agreement between the adjoint sensitivities from $J_O$ and $J_\kappa$ are away from the boundaries. This is either because there are no observations near the boundaries, as is the case with microstructure and Argo, or because there are other factors not included in the observational product impacting the diapycnal diffusivities near the boundaries, as is the case with the *de Lavergne et al.* (2020) product. Thus, the background diapycnal diffusivity is an appropriate description for the model parameter we compare with each of the observational products away from the surface, but we now clarify that we use the full diapycnal diffusivity field. We have edited the text to read: "The resulting $\kappa_{\rho,bg}$ field in the ECCOv4r3 solution–plus the *Gaspar et al.* (1990) contribution–will be referred to as $\kappa_{\rho,ECCO}$ hereafter and is shown in Fig. 2–depth-averaged below the model's average mixed layer

depth. Note that the initial guess for $\kappa_{\rho,bg}$ in ECCO is $10^{-5}$ m$^2$ s$^{-1}$ and in the absence of observation-driven adjustments, $\kappa_{\rho,bg}$ in ECCO remains at or is close to its initial value in the ECCOv4r3 solution, at least in its depth-average. Also note that in regions away from ocean boundary layers, $\kappa_{\rho,ECCO}$ is approximately the $\kappa_{\rho,bg}$ in ECCO."

- In addition, it would be very helpful if the authors can provide references or documents indicating diffKr is indeed a combination of diapycnal diffusivity and the vertical component of the redi mixing rate. This is kind of technical but essential for the whole story.

- **The variable diffKr is just the background diapycnal diffusivity but the sensitivities we used for this study are for the full diapcynal diffusivity (background plus Gaspar et al. (1990)), not including the (3,3) entry of the along-isopycnal tensor. The *Forget et al.* (2015) study in Geoscientific Model Development we cite at least partially describes this (hence their notation with perpendicular et cetera).**